# Nanoscale segregation of channel and barrier claudins enables paracellular ion flux

Hannes Gonschior [1], Christopher Schmied [1], Rozemarijn Eva Van der Veen [1], Jenny Eichhorst[1], Nina Himmerkus[2], Jörg Piontek[3], Dorothee Günzel [3], Markus Bleich [2], Mikio Furuse[4,5], Volker Haucke [1,6] & Martin Lehmann [1] ✉

The paracellular passage of ions and small molecules across epithelia is controlled by tight junctions, complex meshworks of claudin polymers that form tight seals between neighboring cells. How the nanoscale architecture of tight junction meshworks enables paracellular passage of specific ions or small molecules without compromising barrier function is unknown. Here we combine super-resolution stimulated emission depletion microscopy in live and fixed cells and tissues, multivariate classification of super-resolution images and fluorescence resonance energy transfer to reveal the nanoscale organization of tight junctions formed by mammalian claudins. We show that only a subset of claudins can assemble into characteristic homotypic meshworks, whereas tight junctions formed by multiple claudins display nanoscale organization principles of intermixing, integration, induction, segregation, and exclusion of strand assemblies. Interestingly, channel-forming claudins are spatially segregated from barrier-forming claudins via determinants mainly encoded in their extracellular domains also known to harbor mutations leading to human diseases. Electrophysiological analysis of claudins in epithelial cells suggests that nanoscale segregation of distinct channel-forming claudins enables barrier function combined with specific paracellular ion flux across tight junctions.

During development and homeostasis, tissues not only have to tightly control passage of small molecules and ions through transcellular transport mechanisms but also via paracellularly located adhesion complexes including tight junctions (TJs)[1]. TJs form apical cell–cell contacts in epithelia[2] and restrict the passage of pathogens, small molecules, ions, and water by very close paracellular membrane contacts[1]. Claudins are transmembrane proteins that form ~10 nm thick strands interwoven into TJ meshworks[3] and associate intracellularly with numerous scaffolding proteins and the cytoskeleton[4]. These strands can act as paracellular diffusion barriers against small and large

solutes, e.g., when composed of barrier claudins like claudin-1 (Cldn1)[5,6] or Cldn3[7]. In addition, size and charge selective ion/water channels are formed within strands by Cldn2[8–10], Cldn10a/b[11,12], Cldn15[13–15], and Cldn16[16,17].

Early electrophysiological measurements, mathematical modeling[18], as well as structural and functional data on channel and barrier claudins[19,20] led to a model in which paracellular pores formed by selected claudins integrate into TJ strands. How TJ meshworks composed of single or multiple claudins are organized at the nanoscale to integrate paracellular barrier and channel functions is, however,

[1]Leibniz-Forschungsinstitut für Molekulare Pharmakologie (FMP), 13125 Berlin, Germany. [2]Institute of Physiology, Christian-Albrechts-Universität zu Kiel, 24118 Kiel, Germany. [3]Clinical Physiology/Nutritional Medicine, Medical Department, Division of Gastroenterology, Infectiology, Rheumatology, Charité – Universitätsmedizin Berlin, 12203 Berlin, Germany. [4]Division of Cell Structure, National Institute for Physiological Sciences, Okazaki, Aichi 444-8787, Japan. [5]Department of Physiological Sciences, School of Life Science, SOKENDAI (Graduate University for Advanced Studies), Okazaki, Aichi 444-8585, Japan. [6]Faculty of Biology, Chemistry and Pharmacy, Freie Universität Berlin, 14195 Berlin, Germany. ✉e-mail: mlehmann@fmp-berlin.de

unknown. Besides amino acids that enable paracellular charge selectivity, comparison of the primary protein sequences of barrier- *vs.* channel-forming claudins has not uncovered distinctive channel features that could explain their distinct organization[21]. Moreover, freeze fracture electron microscopy (FFEM) images of endogenous or reconstituted claudin strands failed to reveal specific ultrastructural features of barrier and channel claudins[14], potentially due to structural effects of strong chemical fixation required for this analysis[2]. Super-resolution fluorescence microscopy now enables molecular imaging of living cells and fixed tissues with nanometer resolution and has been applied to visualize selected claudins[22–24].

Here we use super-resolution stimulated emission depletion (STED) microscopy in live and fixed cells and tissues to reveal the nanoscale TJ organization of all 26 mammalian claudins. We find that only half of the individually expressed claudins form TJ strands or meshworks on their own, whereas TJs formed by multiple claudins display nanoscale organization principles of intermixing, integration, induction, segregation, and exclusion of strand assemblies. The notable nanoscale segregation of channel- and barrier-forming claudins within individual strands is mediated, at least in part, by claudin extracellular domains, while claudin association with ZO-1 adaptor proteins is dispensable. We hypothesize that segregation of channel and barrier claudins is a key mechanism for proper claudin channel function, that is compromised in certain patients with claudin missense mutations (e.g., N48K)[25]. Functional ion permeability

measurements of claudins re-expressed in a TJ-strand free epithelial cell line indicated that segregation enables the parallel flow of oppositely charged ions and increases the ion specificity. Our data show that paracellular ion transport is mediated by segregated TJ meshworks. These findings may have wide implications for cell physiology and our understanding of tissue barrier function.

## Results

### STED microscopy reveals the nanoscale organization of mammalian claudins in living cells and tissue

Claudins, the key constituent proteins of TJs, are tetraspan transmembrane proteins with two extracellular loops, encoded by 26 genes in mice and 25 genes in humans. They determine the paracellular barrier and flux properties of TJs by forming up to one micrometer-sized meshworks at the most apical part of the lateral membrane of epithelial cell-to-cell contacts (Fig. 1a). Super-resolution STED microscopy with ~50 nm lateral (XY) resolution enabled us to visualize endogenous Cldn3 within TJs of mouse duodenum labeled with specific antibodies (Fig. 1c). While single strands and close strand assemblies below 50 nm distance could not be resolved within the intestinal TJ meshwork, large individual meshes and the overall TJ thickness of ~500 nm became apparent, consistent with earlier observations from FFEM[26]. To characterize the nanoscale organization of individual claudins within TJs, and to overcome limitations of antibody labeling, we reconstituted TJ meshworks formed by individual claudins (and

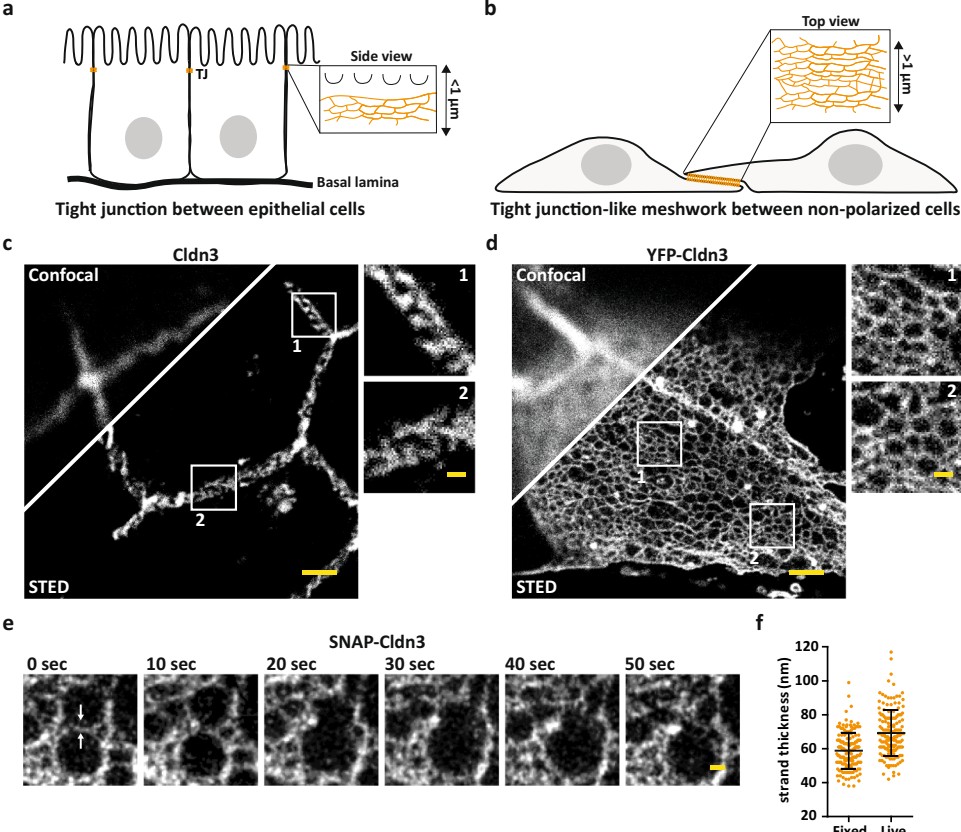

**Fig. 1 | STED microscopy reveals the nanoscale organization of TJ meshworks.**
**a** Scheme illustrating the endogenous TJ at the most apical cell-to-cell contact in epithelial cells. **b** Scheme illustrating TJ-like meshwork formed in flat overlapping areas of claudin transfected non-polarized cells. **c**, **d** Representative confocal and STED image of an endogenous formed TJ labeled for Cldn3 (2nd-Atto647N) between epithelial cells from tissue of mouse duodenum (**c**) and TJ-like meshwork formed by overexpressed YFP-Cldn3 (α-GFP-NB-Atto647N) between two COS-7 cells (**d**).
**e** Representative single-color STED time series (1 frame/10 s) of a TJ-like meshwork in an overlapping region of living COS-7 cells expressing SNAP-Cldn3 (BG-JF646).

White arrows indicate the initial strand break followed by the fusion of two smaller meshes into a larger mesh. A Gaussian blur with a sigma of 20 nm was applied. **f** Full-wide-half-maximum (FWHM) measurement of TJ strands of SNAP-Cldn3 (BG-JF646) in fixed and living COS-7 cells. Data represent the mean ± SD. Every data point represents one line profile of total 160 line profiles from 8 independent TJ-like meshworks ($n = 160$). The overall FWHM resulted in $59 ± 11$ nm for fixed and in $69 ± 14$ nm for living samples. All representative images derive from 3 independent experiments. Scale bars, 1 μm (**c**, **d**) and 200 nm (magnifications in **c**, **d**, and **e**). Source data are provided as a Source data file.

their isoforms) in COS-7 cells, e.i., non-polarized fibroblast-like cells isolated from African green monkey kidney (Fig. 1b). These non-polarized cells lack endogenous tight junctional transmembrane proteins including claudins, but can still form TJ meshworks[8] that colocalize with ZO-1[23], an important scaffolding TJ protein[27], when claudins are exogenously expressed. YFP-tagged Cldn3 visualized in the flat overlapping areas of neighboring COS-7 cells formed large meshworks, comprised of multiple individual claudin strands and small meshes (Fig. 1d), greatly resembling the organization of endogenous Cldn3 in intestinal tissue (Fig. 1c). Similarly organized meshworks were formed by untagged Cldn3 labeled with anti-Cldn3 antibodies and by mouse Cldn3 tagged at either its N- or C-terminal end with an YFP (Supplementary Fig. 1d). Moreover, TJ-like meshworks formed by Cldn3 or related claudins (i.e., Cldn1, Cldn2, Cldn10a) colocalized generally with ZO-1 endogenously expressed in COS-7 cells but only peripherally with actin (Supplementary Fig. 1a–c). Since meshwork morphology might be affected by chemical fixation (Supplementary Fig. 1e) we analyzed the nanoscale organization of SNAP-Cldn3 in living COS-7 cells. Live STED imaging revealed strand dynamics, the formation of large Cldn3 meshes in living cells and a similar Cldn3 meshwork organization as in cells fixed with 4% PFA (Fig. 1e; Supplementary Fig. 1d; Supplementary Movie 1).

Individual claudin strands of ~10 nm thickness can form intricate meshworks of various morphology[8,28]. To characterize individual TJ strands quantitatively, we measured isolated SNAP-Cldn3 strands in fixed and living COS-7 cells and found characteristic widths of $59 \pm 11$ nm and $69 \pm 14$ nm, respectively (Fig. 1f). The slightly larger value in living cells could be due to strand mobility and/or a lower signal-to-noise ratio in cell culture medium compared to fixed cell mounting medium. Of note STED images generally show a lower signal-to-noise ratio than imaging techniques that were previously used to image claudin strands including wide field[29], confocal[29], and SIM[23]. We note that single-molecule localization microscopy of YFP-Cldn3 in fixed HEK cells[24] yielded similar strand widths. Collectively, these results show that STED microscopy can resolve meshes of single strands in living or fixed cells. Therefore, STED is capable of revealing the nanoscale molecular architecture of TJ assemblies in cells and tissue.

## Mammalian claudins show differences in strand formation and meshwork organization

Based on these results and taking advantage of the excellent structural preservation of chemically fixed strand morphologies we analyzed all 26 mammalian claudins, tagged at their N-termini with YFP and stained with Atto647N-labeled anti-GFP nanobodies in fixed COS-7 cells (Supplementary Fig. 2). Since half of the analyzed claudins formed no clear meshworks we wanted to objectively classify TJ strand formations based on STED intensity texture features[30–32]; that are spatial intensity variations across a scale of 20–200 nm, as explained in the method section. Indeed, hierarchical clustering of basic intensity texture features largely confirmed our separation between meshwork-forming from non-meshwork forming claudins (Supplementary Figs. 2 and 3). Most claudins previously classified as classic claudins[21] were able to polymerize into TJ strands; exceptions were Cldn4, Cldn8 and Cldn17. We confirmed the mostly uniform and unstructured distribution of SNAP-Cldn4 or SNAP-Cldn8 by STED imaging of living COS-7 cells (Supplementary Fig. 5i). In contrast, all claudins previously attributed as non-classic[21,33], except for Cldn11 and Cldn20, did not form TJ strands on their own. Instead, they appeared either as uniformly distributed, small punctae, irregular clusters or small strands. Cldn22 and Cldn27 localized partly to ER (Supplementary Fig. 2 and 3b, c). Non-meshwork forming claudins might require co-assembly with classic claudins or other TJ proteins to polymerize into TJ assemblies. We did not observe any correlation between the ability to form strands with claudins either acting as a barrier or a channel

(Supplementary Fig. 2 and 3). Claudins previously implicated in bivalent cation transport such as Cldn12[34,35] or Cldn16[16] were unable to form detectable strands and meshworks under these conditions (Supplementary Fig. 2 and 3).

To further classify TJ meshwork formation by individual claudins, STED images of meshwork-forming claudins were segmented and features of meshwork morphology such as average mesh size, branch length, and the number of branches were extracted and combined with intensity texture features (Fig. 2a). Using hierarchical clustering and principal component analysis, we identified three classes of meshworks: Class A claudins, comprising Cldn7, Cldn10a (anion channel), Cldn19a/b, and Cldn20, formed large mesh sizes. Claudins of class B included cation channel-forming Cldn2, Cldn10b, Cldn15, as well as the barrier-forming Cldn3, Cldn5, and Cldn14, and formed meshes of much smaller sizes. Finally, class C claudins, such as Cldn1, Cldn6, Cldn9, and Cldn11 assembled into very dense meshwork structures with predominantly parallel strands (Fig. 2b; Supplementary Fig. 4). On rare occasions, single strands of Cldn11 were observed and measured to have a similar width as Cldn3 strands (Supplementary Fig. 3d; Fig. 1f). Interestingly class C claudins are known to form particularly tight seals in epithelia[36], a feature that appears to be recapitulated upon exogenous expression in non-polarized COS-7 cells.

Taken together, we found that only about half of all mammalian claudins are able to form meshworks and this ability seems to correlate with protein sequence homology[21,33] but not with so far-known barrier vs. channel function[14]. This suggests that meshwork formation may be an intrinsic feature of claudin 3D-structure and polymer assembly and barrier as well as channel claudins can assemble into similarly organized meshworks that may facilitate their co-assembly.

## Claudin combinations display nanoscale organization principles of intermixing, integration, induction, segregation and exclusion of strand assemblies

TJ meshworks are mostly composed of multiple barrier- and/or channel-forming claudins[36,37]. Copolymers of these claudins may form at different length scales ranging from mixed strands to small separated clusters or larger segments within meshworks thereby creating different local ion permeabilities as suggested previously[1,18,38] (Fig. 3a). To investigate the nanoscale organization of claudin copolymers we co-expressed Cldn3 or Cldn4, two typical barrier-forming claudins found in the respiratory, urinary and gastrointestinal tracts[36], with other claudins in COS-7 cells and analyzed overlapping areas by two-color STED microscopy (Fig. 3b; Supplementary Fig. 5a; Supplementary Fig. 6a). Strikingly, five different types of behaviors of co-expressed claudins were observed: (i) Intermixing of barrier-forming Cldn1, Cldn5, Cldn6, Cldn7, Cldn9, Cldn14, and Cldn19b with Cldn3. (ii) Integration of Cldn4, a claudin incapable of forming TJ strands on its own, into Cldn3 strands. (iii) Induction of co-assembled strands and meshworks from two non-meshwork forming claudins, Cldn4 and Cldn8. (iv) Segregation of channel-forming Cldn2, Cldn10a, Cldn10b and Cldn15 from Cldn3 within individual TJ strands and meshworks resulting in alternating areas of channel-forming and barrier-forming TJs. Finally, (v) exclusion of Cldn3 and Cldn11 from each other by formation of independent TJ meshworks. These five different types of nanoscale organization of claudin copolymers were also observed in the mouse fibroblast cell line 3T3 and in COS-7 cells using a GFP-nanobody labeled with a more hydrophilic fluorophore (Supplementary Fig. 7e, f).

The intermixing, segregation, and exclusion behavior of the various claudins with respect to Cldn3 was confirmed by quantitative colocalisation analysis based on Pearson correlation of pixel intensity, where positive values indicate intermixing and negative values indicate segregation or exclusion (Fig. 3c; Supplementary Fig. 5b–d). This analysis revealed that Cldn3 intermixing with other barrier-forming claudins was indistinguishable from colocalization of Cldn3 with itself.

**a**

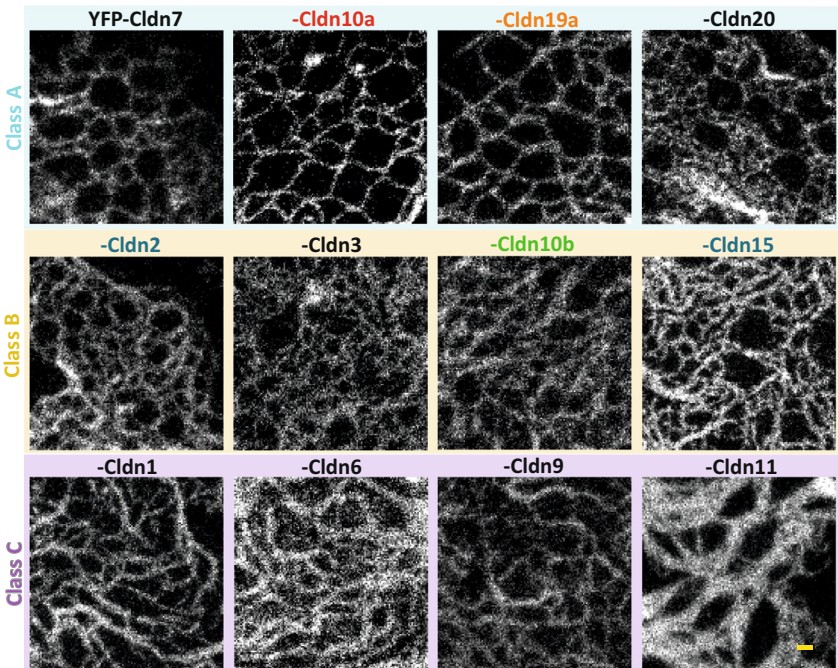

**Fig. 2 | Automated analysis of TJ meshwork enables the overall classification of all meshwork-forming claudins. a** Hierachical clustering of all meshwork forming mammalian claudins expressed in COS-7 cells into three different classes using an automated analysis based on Haralick texture features and image segmentation. All claudins were N-terminally tagged with YFP or GFP and boosted with α-GFP-NB-Atto647N. Columns are centered and unit variance scaling was applied to columns. Correlation distance with average linkage was used for clustering of rows and columns. For the Haralick texture features one pixel (px) equals 20 nm. Three classes are labeled in cyan (class A), yellow (class B), and magenta (class C).

Claudins were color-coded by function, as barrier (black), cation channel (green), anion channel (red), cation and water channel (blue), and heteromeric ion channel formed by two different claudins (orange). Color code of heatmap represents unit variance scaling and represents number of standard deviations. Based on a dataset of 15 claudins with 202 total images. Source data are provided as a Source data file. **b** Representative STED images of TJ-like meshworks from three identified mesh-work classes (A, B, C) in (**a**). ROIs were taken from cell–cell overlaps of two transfected cells. All claudins were N-terminally tagged with YFP and boosted with α-GFP-NB-Atto647N. Scale bar, 200 nm (**b**).

In contrast, channel-forming claudins such as Cldn2, Cldn10a, Cldn10b, and Cldn15 were nearly completely segregated from Cldn3 within the TJ-like meshwork, as indicated by a significantly lower and sometimes even negative colocalization index (Fig. 3c). A similarly negative colocalization index was seen for Cldn3 and Cldn11 that formed mutually exclusive TJ meshworks (Supplementary Fig. 5d). To test whether these behaviors reflect a general organizational principle of TJ formation by claudins we analyzed the barrier-forming Cldn1. We found Cldn1 to intermix with Cldn3, but segregate from channel-forming Cldn2, Cldn10a, Cldn10b, and Cldn15 (Supplementary Fig. 5f).

To challenge these findings by an alternative approach, we analyzed claudin interactions in living cells on a molecular scale of ~10 nm

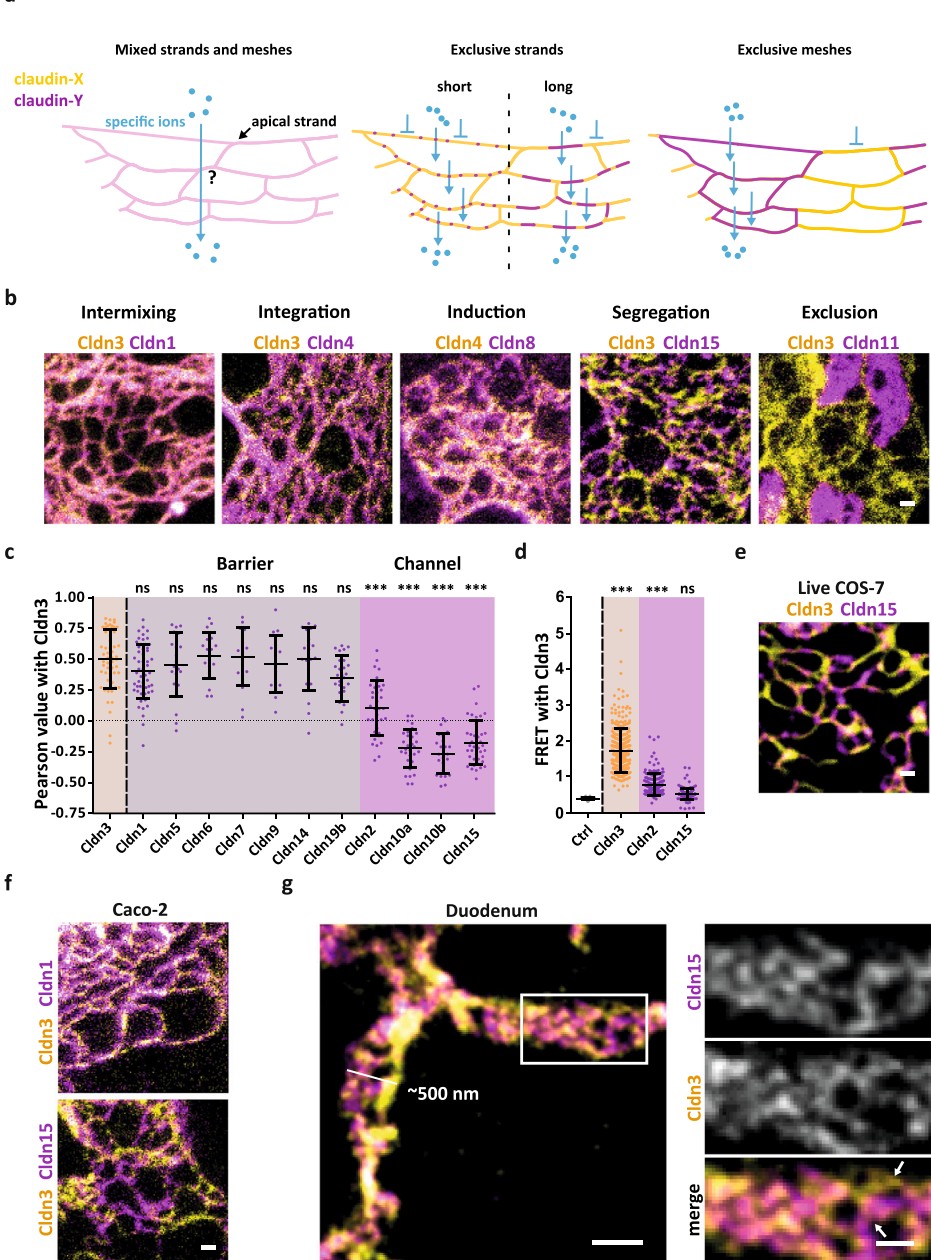

**Fig. 3 | Claudins form characteristic copolymers including segregation of channel-forming claudins from Cldn3. a** Scheme is illustrating predicted claudin-claudin organization patterns. Two claudins (yellow and magenta) that are compatible with each other tightly colocalize in the TJ (pink). Incompatible claudins separate into different strands or even larger claudin-specific parts. This separation might facilitate permeability of specific ions (cyan) over the TJ meshwork. **b** Representative STED images of TJ-like meshwork of the observed five organization patterns formed by the indicated co-expressed claudins (SNAP-tagged (yellow; BG-Atto590) and YFP-tagged (magenta; α-GFP-NB-Atto647N)) in fixed COS-7 cells are shown. Tight colocalization: intermixing of mesh-forming claudins, integration of non-mesh-forming with a mesh-forming claudin or induction (de novo mesh-forming of co-expressed non-mesh-forming claudins). Separated claudins: segregation and exclusion. **c** Pearson correlation analysis of Cldn3 co-expressed with barrier-forming (gray) or channel-forming (magenta) claudins. Co-expression of Cldn3 with Cldn3 (yellow) served as positive control. Data represent the mean ± SD. Every $n$ represents the Pearson of one TJ-like meshwork. $n$(Cldn3 + Cldn3) = 55; $n$(Cldn3 + Cldn1) = 54; $n$(Cldn3 + Cldn5) = 20; $n$(Cldn3 + Cldn6) = 20; $n$(Cldn3 + Cldn7) = 17; $n$(Cldn3 + Cldn9) = 15; $n$(Cldn3 + Cldn14) = 19; $n$(Cldn3 + Cldn19b) = 25; $n$(Cldn3 + Cldn2) = 36; $n$(Cldn3 + Cldn10a) = 30; $n$(Cldn3 + Cldn10b) = 24; $n$(Cldn3 + Cldn15) = 42; from 3–6 independent experiments; one-way ANOVA with

Dunnett's multiple comparison test; ***$P \leq 0.001$, ns (not significant). **d** Spectral FRET analysis of Trq2-Cldn3 alone (blank; negative control) or co-expressed with indicated YFP-tagged barrier Cldn3 (yellow; positive control) and channel-forming Cldn2 or Cldn15 (magenta) in HEK293 cells. Data represent the mean ± SD. Every $n$ represents one cell–cell contact. $n$(negative control) = 57, from one experiment; $n$(Cldn3 + Cldn3) = 312; $n$(Cldn3 + Cldn2) = 162; $n$(Cldn3 + Cldn15) = 154; all from 4 to 5 independent experiments; one-way ANOVA with Dunnett's multiple comparison test; ***$P \leq 0.001$, ns (not significant). **e** Representative STED image of segregating SNAP-Cldn3 (yellow; BG-JF646) and Halo-Cldn15 (magenta; CA-Atto590) in living COS-7 cells. Image noise was removed for visualization using noise2Void[70]. **f** Representative STED images of intermixing and segregation of SNAP-Cldn3 (yellow; BG-Atto590) with YFP-tagged Cldn1 and Cldn15 (both magenta; α-GFP-NB-Atto647N), respectively expressed in Caco-2 cells. **g** Representative STED images of the endogenous formed TJ in cryo-sections from mouse duodenum immunostained for Cldn3 (yellow; 2nd-Atto647N) and Cldn15 (magenta; 2nd-AF594). White arrows indicate regions in the TJ that shows the segregation of Cldn3 and Cldn15 strands. A Gaussian blur with a sigma of 20 nm was applied. All representative images derive from 3 independent experiments. Scale bars, 500 nm (**g**), 200 nm (**b**, **e**, **f**, magnification in **g**). Source data are provided as a Source data file.

by spectral fluorescence resonance energy transfer (FRET) measurements. This also allowed us to cope with the potential caveat that the observed intermediate colocalization of Cldn3 with some channel-forming Cldn2, a paracellular sodium[39] and water channel[40], might reflect the limited resolution of STED microscopy of ~50 nm. Selected Turquoise- and YFP-tagged claudins were visualized in lateral contacts in human epithelial cells (HEK293) (Fig. 3d; Supplementary Fig. 5g). We found much lower FRET for heterotypical Cldn3-Cldn2 interaction compared to homotypic Cldn3-Cldn3 assemblies, verifying earlier studies[41]. No FRET between barrier-forming Cldn3 and channel-forming Cldn15 was observed in HEK293 and COS-7 cells (Fig. 3d; Supplementary Fig. 7c).

Integration, induction and segregation could further be observed by live cell STED imaging in COS-7 cells (Fig. 3e; Supplementary Fig. 5j), excluding the possibility that these nanoscale organization principles represent artefacts of chemical fixation and nanobody labeling. Importantly, endogenous Cldn3 and Cldn15, which are highly expressed in the intestine where Cldn15 forms a paracellular Na+ channel required to drive nutrient uptake[42], also segregated, upon exogenous expression of Cldn3 and Cldn15 in human intestinal Caco-2 cells (Fig. 3f; Supplementary Fig. 5e) and in mouse duodenum tissue (Fig. 3g; Supplementary Fig. 5h). We speculate that the apparent differences in claudin meshwork architecture between native tissue and cell lines (Fig. 3b, e, f, g) might be attributable to differences in sample preparation and TJ orientation (cryosectioned tissue vs. intact living or fixed cells), post-fixation by ethanol, staining procedures (indirect immunofluorescence vs. genetically encoded markers), and/or the presence of other TJ proteins and additional cell adhesion structures in epithelial cells or native tissue. Overall, the segregation between barrier- and channel-forming claudins appears to be a protein-intrinsic property rather than a reflection of cell type or tissue.

These data suggest that claudins co-polymerize according to distinct organizational principles that drive the segregation of barrier- and channel-forming claudins in cells and tissues.

## Channel-forming claudins form segregated meshworks in cells and tissue

Channel-forming claudins such as Cldn2, Cldn10a, Cldn10b, and Cldn16 in combination with Cldn19 are highly abundant in the kidney and facilitate Na+, Cl−, and Mg2+ resorption[10,16,43]. We thus wanted to know whether channel claudins would not only segregate from barrier Cldn3 but also from each other possibly to preserve their unique ion channel properties or create localized ion permeabilities. When we co-expressed Cldn2, i.e., a paracellular Na+ channel, with other channel-forming claudins such as the Cl−-selective Cldn10a or the Na+-selective claudins Cldn10b and Cldn15, or with barrier Cldn1 in COS-7 cells, we observed segregated TJ-like meshworks as evidenced by their low colocalization index in STED analyses (Fig. 4a, b; Supplementary Fig. 6b). FRET measurements in HEK293 and COS-7 cells confirmed the observed lack of interaction between Cldn2 and Cldn10a on a scale below 10 nm in living cells (Fig. 4c; Supplementary Fig. 7c). Segregation of channel-forming Cldn2 and Cldn10a was also found in living COS-7 cells visualized by live STED imaging (Fig. 4d) and in chemically fixed not fully-polarized kidney epithelial cells (Fig. 4e, f).

In the thick ascending limb of Henle's loop of the nephron paracellular Na+ and Mg2+ flux is mediated by three claudins Cldn10b/16/19 that were found to localize in a cell−cell contact specific mosaic pattern[43]. Interestingly, a combination of integration, segregation and exclusion was observed upon exogenous coexpression of Cldn10b, Cldn16, and Cldn19a in COS-7 cells (Supplementary Fig. 7a, b). Pairwise coexpression of Cldn19a and Cldn16 produced integrating meshworks. Cldn19a and Cldn10b formed segregated meshworks, whereas meshworks formed by Cldn16 and Clnd10b were excluded from each other. When all three claudins were co-expressed together, Cldn16 became integrated into Cldn19a strands that remained segregated from

strands composed of Cldn10b (Supplementary Fig. 7a, b). Therefore, three of the basic nanoscale organization principles of claudins, i.e., integration, segregation, and exclusion, are preserved in a reconstituted system and could lead to spatial separation of paracellular Na+ and Mg2+ transport found in TAL[43].

To determine whether the organizational principles indeed govern the nanoscale distribution of claudins in the nephron, we stained the proximal tubule of isolated mouse nephrons. As predicted from our analysis of claudins reconstituted in COS-7 cells, we found alternating immunoreactivity for Cldn2 and Cldn10a in primary mouse nephrons, that displayed a low degree of colocalisation (Fig. 4g, h; Supplementary Fig. 7d). Apart from their segregation behavior in proximal tubules, Cldn2 and Cldn10a displayed considerable heterogeneity with respect to strand length between different tubules and even within the same tubule without any indication of a claudin or location-specific strand length (Fig. 4i).

We conclude that TJ meshworks formed by different claudins not only are segregated between barrier- and channel-forming claudins but also between distinct channel-forming claudin isoforms, possibly to spatially restrict ion flux on the nanoscale within cells and tissues.

## Claudin protein levels and extracellular loop sequences regulate claudin segregation

Next, we set out to determine the molecular basis of the observed segregation of distinct channel claudins. As the ability to form segregated claudin meshworks may depend on relative protein copy numbers, we first tested the effect of different protein expression ratios. Channel-forming Cldn2 and Cldn10a displayed segregation with low colocalisation at all expression ratios tested, whereas strand lengths positively correlated to expression levels (Fig. 5a).

While these data demonstrate that TJ strand length is determined by claudin copy numbers and polymerization properties, they fail to provide insight into the mechanism underlying channel claudin segregation. In principle, two main mechanisms could mediate the segregation of distinct channel claudins: (i) claudin segregation might be a consequence of their differential association with other factors, e.g., membrane lipids such as cholesterol, which was shown to be required for TJ formation[44] and to associate with claudins in distinct membrane domains[45], or TJ-associated intracellular scaffold proteins, most notably ZO-1 that has been proposed to promote TJ assembly via multiple mechanisms[46–48] including liquid-liquid phase separation[49]. (ii) Alternatively, claudin segregation might reflect an intrinsic property of claudin proteins themselves that could either relate to their channel activity or the ability to assemble via their extracellular and transmembrane domains.

We tested these mechanisms by combined chemical and genetic approaches. Treatment of COS-7 cells with the HMG-CoA reductase blocker mevastatin or with the cholesterol-sequestering drug Methyl-β-cyclodextrin (MβCD) markedly reduced plasma membrane cholesterol levels (Supplementary Fig. 8a) but did not impair claudin segregation (Fig. 5b). Deletion of the C-terminal PDZ binding motif (required for binding to ZO-1) or even almost the entire C-terminal cytoplasmic domain (Fig. 5c) in either Cldn2 and Cldn10a or Cldn3 and Cldn15 resulted in smaller and less frequent meshworks, but did not affect the ability of claudins to segregate along TJ strands (Fig. 5d, e; Supplementary Fig. 8d, e). Hence, claudin segregation does not appear to be a consequence of their association with extrinsic factors such as cholesterol, proteins that bind to phosphorylated residues within the C-terminal tail, or PDZ domain proteins such as ZO-1.

We therefore pursued the alternative hypothesis that claudin segregation is a protein-intrinsic property built into their 3D architecture. The fact that distinct channel-forming claudins segregated from each other as well as from barrier-forming claudins indicates that segregation might be a consequence of ion flux. Potentially, this stabilizes claudin channel polymers. We tested this possibility by

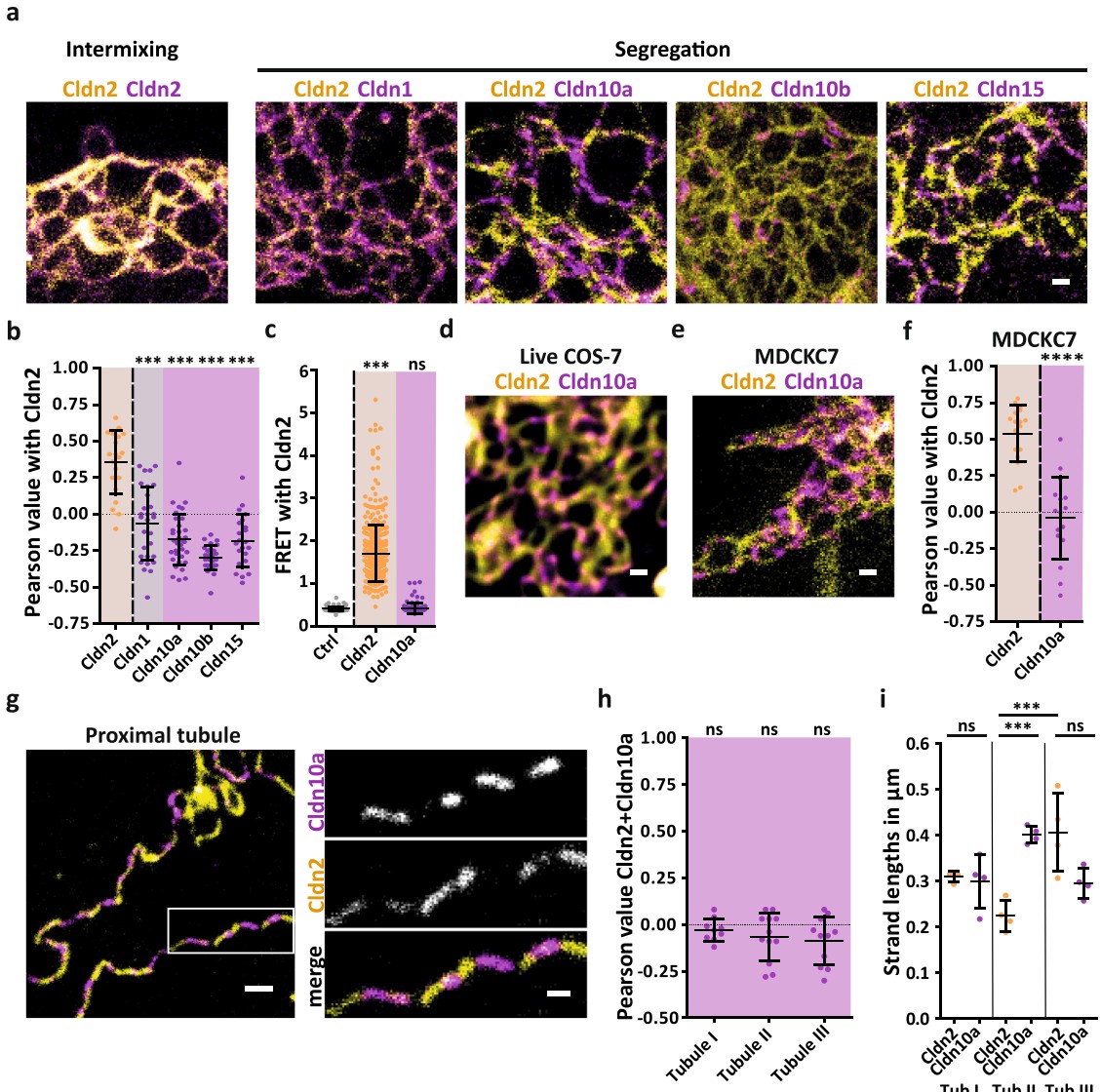

**Fig. 4 | Channel-forming claudins form segregated meshworks in non-polarized cells, epithelial cells and the murine proximal tubule.**
**a** Representative STED image of TJ-like meshwork formed by SNAP-Cldn2 (yellow; BG-Atto590) co-expressed with YFP-tagged Cldn2, Cldn1, Cldn10a, Cldn10b, and Cldn15 (all magenta; α-GFP-NB-Atto647N) in COS-7 cells.
**b** Pearson correlation analysis of Cldn2 co-expressed with Cldn2 (yellow), Cldn1 (gray) and Cldn10a, Cldn10b, Cldn15 (magenta). Data represent the mean ± SD. Every *n* represents the Pearson of one TJ-like meshwork. *n*(Cldn2 + Cldn2) = 21; *n*(Cldn2 + Cldn1) = 29; *n*(Cldn2 + Cldn10a) = 35; *n*(Cldn2 + Cldn10b) = 28; *n*(Cldn2 + Cldn15) = 22; from 3 to 5 independent experiments; one-way ANOVA with Dunnett's multiple comparison test; ***$P \leq 0.001$, ns (not significant).
**c** Spectral FRET analysis of Trq2-Cldn2 (blank; negative control) or co-expressed with YFP-tagged Cldn2 (yellow; positive control) and Cldn10a (magenta) in HEK cells. Data represent the mean ± SD. Every *n* represents one cell–cell contact. *n*(negative control) = 126, from one experiment; *n*(Cldn2 + Cldn2) = 405; *n*(Cldn2 + Cldn10a) = 115; from 4 to 10 independent experiments; one-way ANOVA with Dunnett's multiple comparison test; ***$P \leq 0.001$, ns (not significant). **d** Representative STED image of segregating SNAP-Cldn2 (yellow; BG-JF646) and Halo-Cldn10a (magenta; CA-Atto590) in living COS-7 cells. Image noise was removed for visualization using noise2Void[70]. **e** Representative STED image of segregation of SNAP-Cldn2 (yellow; BG-Atto590) and YFP-Cldn10a

(magenta; α-GFP-NB-Atto647N) localized in an overlap region of two transfected MDCKC7 cells. **f** Pearson correlation analysis of Cldn2 co-expressed with Cldn2 (yellow) and Cldn10a (magenta) in MDCKC7. Data represent the mean ± SD. Every *n* represents the Pearson of one TJ-like meshwork. *n*(Cldn2 + Cldn2) = 17; *n*(Cldn2 + Cldn10a) = 16; from 3 independent experiments; Mann-Whitney test, two-tailed; ****$P \leq 0.0001$. **g** Representative STED image of the TJ in mouse proximal tubule immunostained for Cldn2 (yellow; 2nd-Atto647N) and Cldn10a (magenta; 2nd-AF594). The white rectangle indicates the area of the magnification. **h** Pearson correlation analysis of Cldn2 and Cldn10a in three mouse proximal tubules. Data represent the mean ± SD. Every *n* represents the Pearson of one STED image of the tubular TJ. *n*(Tubule I) = 9; *n*(Tubule II) = 12; *n*(Tubule III) = 11; one-way ANOVA Tukey's multiple comparison test; ns (non-significant). **i** Strand length analysis of Cldn2 and Cldn10a strands in three proximal tubules (Tub I-III). Data represent the mean ± SD; *n* = 4, mean of the measured strands for each tubule; strands Tubule I: 541 (Cldn2)/664 (Cldn10a), strands Tubule II: 747 (Cldn2)/791 (Cldn10a), strands Tubule III: 693 (Cldn2)/828 (Cldn10a); one-way ANOVA with Tukey's multiple comparison test; ***$P \leq 0.001$, ns (non-significant). All representative images derive from 3 independent experiments. Scale bars, 500 nm (**g**), 200 nm (**a**, **d**, **e**, magnification in **g**). Source data are provided as a Source data file.

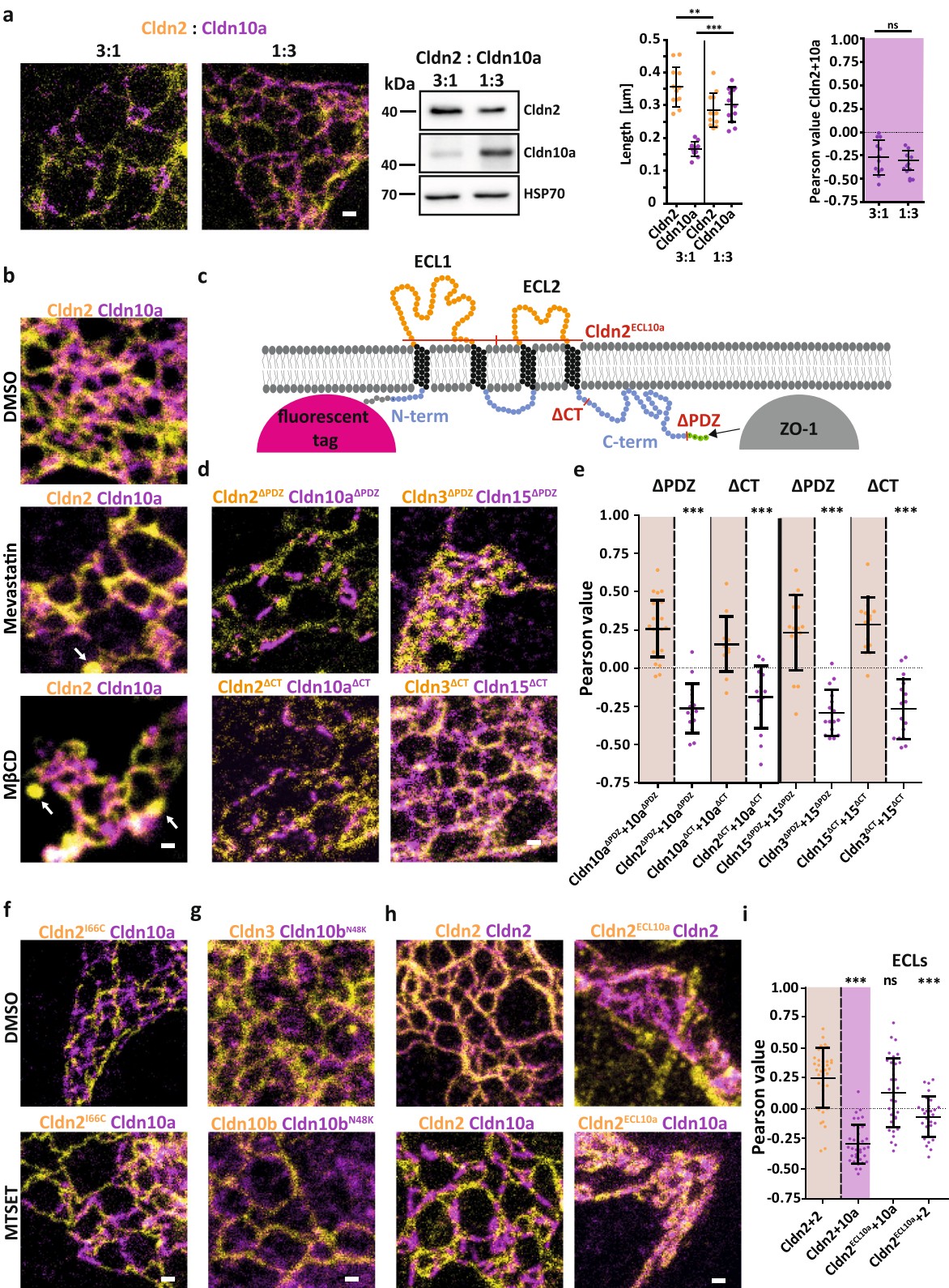

mutating I66 in the pore center of Cldn2 to Cysteine to allow pharmacological blockade of ion permeation by application of the cysteine-reactive reagent MTSET[20,50]. MTSET treatment of mutant Cldn2[I66C] expressed in COS-7 cells did not affect its ability to form a TJ-like meshwork. When we expressed Cldn2[I66C] with Cldn10a we observed segregation in both control (DMSO) and MTSET conditions (Fig. 5f). Hence, channel activity is dispensable for channel-forming claudin segregation.

A pathogenic mutation (N48K) in the first extracellular loop of Cldn10b causes HELIX (hypohidrosis, electrolyte imbalance, lacrimal gland dysfunction, ichthyosis, and xerostomia) syndrome. Cldn10b dysfunction alters the paracellular cation permeability in HELIX syndrome patients, resulting in anhidrosis and kidney damage[25,51]. Consistently, we found that Cldn10b[N48K] fails to form Cldn10b WT-like meshworks (Supplementary Fig. 8b) and is not integrated into TJ meshworks formed by Cldn3 and Cldn10b (Fig. 5g; Supplementary

**Fig. 5 | Claudin segregation is conserved in the extracellular loops and does not depend on cholesterol or the binding to ZO-1. a** Representative STED images, immunoblot analysis, strand length, and Pearson correlation analysis of TJ-like meshwork formed by SNAP-Cldn2 (yellow; BG-Atto590) and YFP-Cldn10a (magenta; α-GFP-NB-Atto647N) with different expression ratios (3:1, 1:3) in COS-7 cells. Blotting was performed against SNAP tag and YFP tag. HSP70 (70 kDa) served as loading control. For the strand and Pearson analysis the same TJ-meshworks were used. Data represent the mean ± SD. For the strand length analysis every *n* represents the mean of 40 strands of Cldn2 and Cldn10a per meshwork from 12 TJ-like meshworks (*n* = 12); one-way ANOVA Tukey's multiple comparison test; \*\*\*$P \leq 0.0001$, \*\*$P \leq 0.01$, ns (not significant); from 3 independent experiments; for the Pearson analysis every *n* represents one TJ-like meshwork (*n* = 12); Mann-Whitney test, two-tailed; ns (not significant) ($P = 0.8623$) from 3 independent experiments. **b** Representative STED images of control and cholesterol-depleted COS-7 cells expressing SNAP-Cldn2 (yellow; BG-Atto590) and YFP-Cldn10a (magenta; α-GFP-NB-Atto647N). White arrows point at claudin-containing vesicles that were increasingly formed under cholesterol depletion conditions. **c** Schematic representation of a claudin in the plasma membrane. The sites for the ECL exchange of Cldn2$^{ECL10a}$ and the PDZ- and C-term deletion mutants (ΔPDZ and ΔCT) are pointed out with red lines. **d** Representative STED images of COS-7 cells co-expressing SNAP-Cldn2$^{ΔPDZ}$/-Cldn3$^{ΔPDZ}$ (yellow; BG-JF646) with YFP-Cldn10a$^{ΔPDZ}$/-Cldn15$^{ΔPDZ}$ (magenta; α-GFP 2$^{nd}$-AF594) or SNAP-Cldn2$^{ΔCT}$/-Cldn3$^{ΔCT}$ (yellow; BG-

JF646) with YFP-Cldn10a$^{ΔCT}$/-Cldn15$^{ΔCT}$ (magenta; α-GFP 2$^{nd}$-AF594). **e** Pearson analysis of co-expressed mutants from (**d**). Data represent the mean ± SD. Every *n* represents the Pearson of one TJ-like meshwork. *n*(Cldn10a$^{ΔPDZ}$ + Cldn10a$^{ΔPDZ}$) = 27; *n*(Cldn2$^{ΔPDZ}$ + Cldn10a$^{ΔPDZ}$) = 15; *n*(Cldn10a$^{ΔCT}$ + Cldn10a$^{ΔCT}$) = 15; *n*(Cldn2$^{ΔCT}$ + Cldn10a$^{ΔCT}$) = 15; *n*(Cldn15$^{ΔPDZ}$ + Cldn15$^{ΔPDZ}$) = 16; *n*(Cldn3$^{ΔPDZ}$ + Cldn15$^{ΔPDZ}$) = 16; *n*(Cldn15$^{ΔCT}$ + Cldn15$^{ΔCT}$) = 15; *n*(Cldn3$^{ΔCT}$ + Cldn15$^{ΔCT}$) = 16; from 3 independent experiments; one-way ANOVA Tukey's multiple comparison test; \*\*\*$P \leq 0.001$. **f** Representative STED images of COS-7 cells co-expressing cysteine mutant SNAP-Cldn2$^{I66C}$ (yellow; BG-Atto590) and YFP-Cldn10a (magenta; α-GFP-NB-Atto647N) in presence of channel blocking agent MTSET or DMSO as control. **g** Representative STED images of COS-7 cells co-expressing SNAP-Cldn3 or SNAP-Cldn10b (both yellow; BG-Atto590) with YFP-Cldn10b$^{N48K}$ (magenta; α-GFP-NB-Atto647N). **h** Representative STED images of co-expressed SNAP-Cldn2 (yellow; BG-JF646) with YFP-Cldn2 or YFP-Cldn10a (magenta; α-GFP 2$^{nd}$-AF594) and SNAP-Cldn2$^{ECL10a}$ (yellow; BG-JF646) with YFP-Cldn2 or YFP-Cldn10a (magenta; α-GFP 2$^{nd}$-AF594) in COS-7 cells. **i** Pearson analysis of co-expressed combinations from (**h**). Data represent the mean ± SD. Every *n* represents the Pearson of one TJ-like meshwork. *n*(Cldn2 + Cldn2) = 37; *n*(Cldn2 + Cldn10a) = 35; *n*(Cldn2$^{ECL10a}$ + Cldn10a) = 34; *n*(Cldn2$^{ECL10a}$ + Cldn2) = 29; from 4 independent experiments; one-way ANOVA with Dunnett's multiple comparison test; \*\*\*$P \leq 0.001$, ns (not significant). All representative images derive from 3 independent experiments. Scale bars, 200 nm (**a**, **b**, **d**, **f**–**h**). Source data are provided as a Source data file.

---

Fig. 8c). This finding indicates a crucial role for the extracellular loops (ECLs) in strand incorporation. To test this hypothesis further, we re-engineered Cldn2 by replacing its ECLs with those of Cldn10a (Cldn2$^{ECL10a}$) (Fig. 5c). Cldn2$^{ECL10a}$ showed significantly less segregation from Cldn10a, while it efficiently segregated from Cldn2 (Fig. 5h, i). FRET and co-culture experiments confirmed these results. They further revealed that although the exchange of the extracellular loops that dominate claudin interactions reversed the segregation behavior, it had only had minor effects on the lateral association of Cldn2$^{ECL10a}$ and normal Cldn2 in cis (Supplementary Fig. 8f, g). These results are consistent with the fact that claudin-claudin interactions are not only based on the ECLs but also on the association of their transmembrane domains[52,53]. In summary, our findings indicate that channel-forming claudins spatially segregate from each other via determinants encoded in their extracellular domains.

## Claudin segregation enables parallel paracellular ion flux and increases ion specificity of TJ

The segregation of channel- from barrier-forming claudins and of channel-forming claudins from each other suggests that it may be of physiological importance to enable paracellular ion flux while maintaining the function of the TJ barrier. Ion flux could potentially occur at the interfaces between segregating claudins, which may form unstable polymer break points. To determine the ion permeabilities of single claudins and segregated claudin pairs we capitalized on genome-edited claudin-deficient TJ-strand depleted quintuple claudin knock-out MDCKII cells (MDCKII QKO)[54] in which the expression of major endogenous claudins found by Shukla et al.[55] namely Cldn1, 2, 3, 4, and 7 is eliminated. While we were able to detect expression of Cldn12 and Cldn16 in MDCKII WT and MDCK QKO cells by immunoblot analysis, both claudins largely failed to colocalize with occludin or ZO-1 at the plasma membrane (Supplementary Fig. 9). Furthermore neither Cldn12 nor Cldn16 appeared to be capable of forming TJ strands when expressed on their own in the absence of other claudins (Supplementary Fig. 2) and do not copolymerize with strands and meshworks formed by Cldn2, 3, 10a, or Cldn15 in COS-7 cells when analyzed by STED microscopy (Supplementary Fig. 10).

Consistent with previous publications we found parental MDCKII cells to display a trans-epithelial resistance (TER) of 37 ohm\*cm² and a Na+ selective TJ barrier[56] (Fig. 6d; Supplementary Fig. 12). In contrast, MDCKII QKO cells suffered from a defective barrier function characterized by low TER, high macromolecule permeation[54], and high and non-selective ion flux similar to TJ-free ZO-1/ZO-2 dKO cells[54] (Fig. 6d;

Supplementary Fig. 12). Importantly, TJ barrier properties and selective ion fluxes could be restored by retrovirus-mediated re-expression of selected FLAG-tagged claudins in MDCKII QKO cells (Fig. 6a–e; Supplementary Fig. 11 and 12). Reconstitution of barrier-forming Cldn3 pronouncedly increased TER, lowered fluorescein flux, and decreased absolute Na+ and Cl− permeabilities compared to MDCKII QKO cells indicating the formation of a tight TJ barrier to all ions. Conversely, re-expression of individual channel-forming Cldn2, Cldn10a, or Cldn15 slightly increased the TER, lowered fluorescein flux and restored the relative (Fig. 6d; Supplementary Fig. 12) and absolute permeabilities of TJs for Na+ (Cldn2, Cldn15) or Cl− (Cldn10a) compared to MDCKII QKO cells (Fig. 6e). Of note prior structural modeling of the Cldn15 ion pore[57] predicted a ~4-fold Na+/Cl− selectivity, in close agreement with our experimental data (Fig. 6d; Supplementary Fig. 12c, d). Furthermore, these results are consistent with multiple studies reporting channel claudins expressed within a complex background of other claudins[9,11,13,39].

Given that single claudin reconstitution enables specific ion permeability, we conclude that ions do not pass through segregation interfaces but through pores formed within claudin strands. When the segregating channel-forming claudin pair Cldn2 (i.e., a cation channel) and Cldn10a (i.e., an anion channel) was co-expressed, we observed Cldn2/Cldn10a segregation (Fig. 6b) and the absolute permeabilities for Na+ and Cl− were similar to those of MDCKII QKO cells expressing either Cldn2 or Cldn10a alone (Fig. 6e; Supplementary Fig. 12c, d). These values indicate that TJ formed by segregating channels claudins Cldn2 and 10 enable parallel Na+ and Cl− flux consistent with permeabilities measured in tight MDCK cells co-overexpressing Cldn2 and Cldn10a[58]. Co-expression of the Na+ channel-forming Cldn15 together with barrier-forming Cldn3 produced high and more selective Na+ permeability compared to single Cldn15 expression (Fig. 6e; Supplementary Fig. 12c, d).

Based on these data together with the observed claudin segregation in cells and mouse tissue we propose an extension of the paracellular ion flux model by Claude et al.[18] and Weber et al.[38] that integrates highly selective ion permeability through stretches of channel claudins with a general barrier function of the TJ (Supplementary Fig. 13).

## Discussion

The TJ meshwork is a functional and structural highly complex multi-protein assembly, on the one hand important for controlling paracellular ion flux, and on the other hand for sealing tissues against

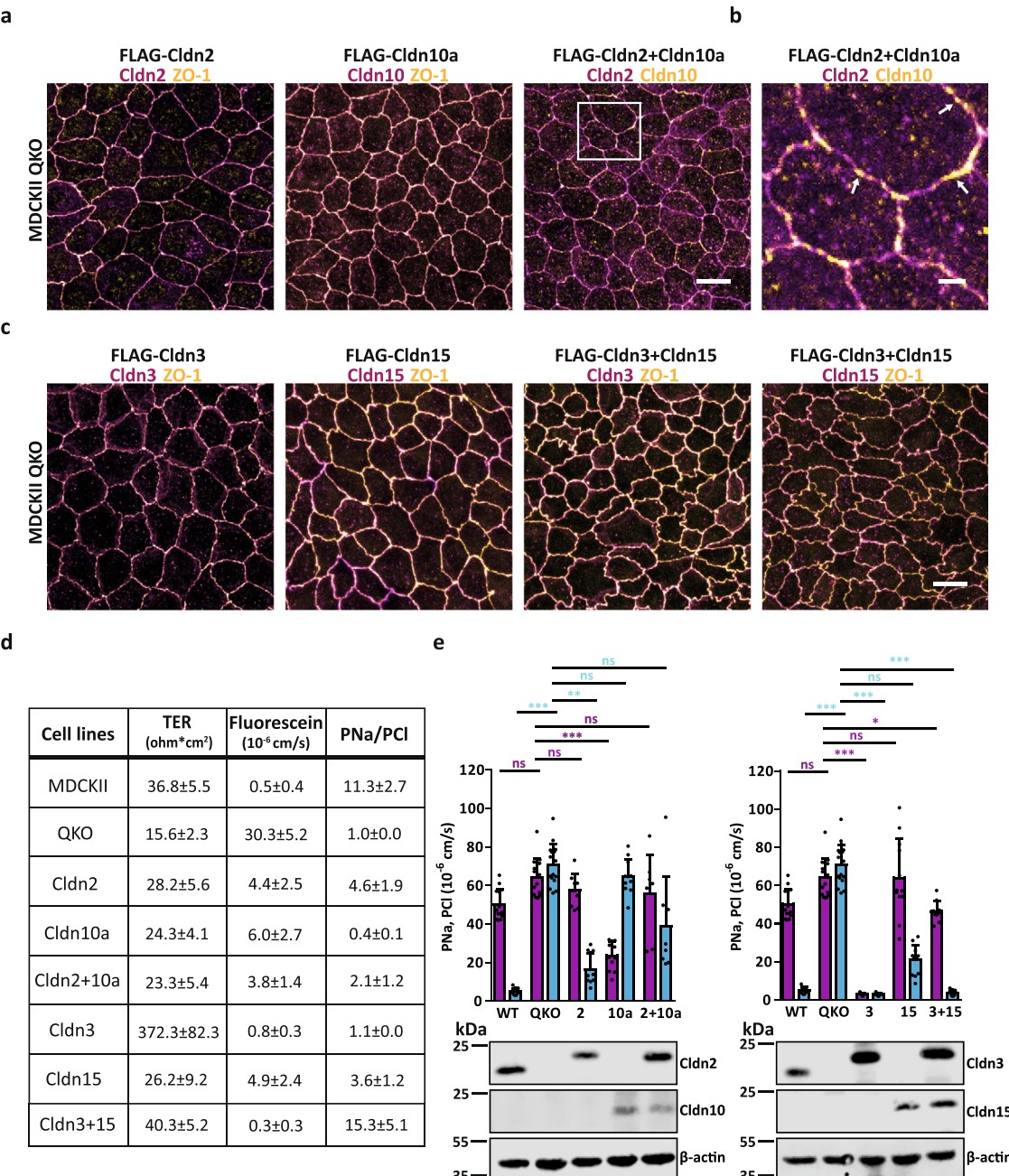

| Cell lines | TER (ohm*cm²) | Fluorescein (10⁻⁶ cm/s) | PNa/PCl |
|---|---|---|---|
| MDCKII | 36.8±5.5 | 0.5±0.4 | 11.3±2.7 |
| QKO | 15.6±2.3 | 30.3±5.2 | 1.0±0.0 |
| Cldn2 | 28.2±5.6 | 4.4±2.5 | 4.6±1.9 |
| Cldn10a | 24.3±4.1 | 6.0±2.7 | 0.4±0.1 |
| Cldn2+10a | 23.3±5.4 | 3.8±1.4 | 2.1±1.2 |
| Cldn3 | 372.3±82.3 | 0.8±0.3 | 1.1±0.0 |
| Cldn15 | 26.2±9.2 | 4.9±2.4 | 3.6±1.2 |
| Cldn3+15 | 40.3±5.2 | 0.3±0.3 | 15.3±5.1 |

**Fig. 6 | Segregation enables paracellular ion flux over the TJ meshwork.**
**a** Representative confocal images of MDCKII QKO cells stably expressing (sT) FLAG-Cldn2, FLAG-Cldn10a, and FLAG-Cldn2+FLAG-Cldn10a. Single claudin expressing cells were immunostained with anti-Cldn2 (magenta; 2nd-Atto647N) or anti-Cldn10 (magenta; 2nd-Atto647N) and anti-ZO-1 (yellow; 2nd-AF594) antibody. Double claudin expressing cells were immunostained with anti-Cldn2 (magenta; 2nd-Atto647N) and anti-Cldn10 (yellow; 2nd-AF594). The white rectangle indicates the region of interest for the magnification in (**b**). **b** Magnification of FLAG-Cldn2+FLAG-Cldn10a expressing cells from (**a**). White arrows point out differently sized TJ parts that contain only Cldn10a. **c** Representative confocal images of MDCKII QKO sT FLAG-Cldn3, FLAG-Cldn15 and FLAG-Cldn3+FLAG-Cldn15. Single and double claudin expressing cells were stained with anti-Cldn3 (magenta; 2nd-Atto647N) or anti-Cldn15 (magenta; 2nd-Atto647N) and anti-ZO-1 (yellow; 2nd-AF594) antibody. **d** Summary of the TER (ohm*cm²), fluorescein flux (10⁻⁶ cm/s) and PNa/PCl ratio from the measured cell lines: MDCKII, MDCKII QKO and MDCKII QKO sT FLAG-tagged Cldn2, Cldn3, Cldn10a, Cldn15 and MDCKII QKO sT FLAG-tagged

Cldn2 + Cldn10a and FLAG-tagged Cldn3 + Cldn15. Data represent the mean ± SD from 3 to 5 independent experiments. Single data points are shown in Supplementary Fig. 12. **e** Absolute permeability for sodium (PNa; magenta bars) and chloride (PCl; cyan bars) of MDCKII, MDCKII QKO and MDCKII QKO sT FLAG-tagged Cldn2, Cldn3, Cldn10a, Cldn15, and MDCKII QKO sT FLAG-tagged Cldn2 + Cldn10a and FLAG-tagged Cldn3 + Cldn15. Data represent the mean ± SD. For the electrophysiological measurements every *n* represents one transwell filter; *n*(MDCKII) = 14; *n*(MDCKII QKO) = 14; *n*(FLAG-Cldn2) = 9; *n*(FLAG-Cldn10a) = 11; *n*(FLAG-Cldn2+FLAG-Cldn10a) = 8; *n*(FLAG-Cldn3) = 9; *n*(FLAG-Cldn15) = 12; *n*(FLAG-Cldn3+FLAG-Cldn15) = 9; from 3–5 independent experiments; one-way ANOVA with Kruskal−Wallis test; ***$P \le 0.001$, **$P \le 0.01$, ns (not significant). The total claudin expression level was determined by immunoblotting for Cldn2, Cldn3, Cldn10, and Cldn15 (20−25 kDa). β-actin (40 kDa) served as control. All representative images derive from 3 independent experiments. Scale bars, 10 μm (**a**, **c**), 2 μm (**b**). Source data are provided as a Source data file.

pathogens and their toxins. How TJs and their proteins coordinate this interplay at the nanoscale level has remained unknown since their initial discovery[23,24,59,60]. Using super-resolution STED microscopy, we overcame the diffraction limit of fluorescence microscopy and the limitations of electron microscopy for the visualization of the molecular nanoscale composition of TJ in living or fixed cells and in different tissues (Figs. 1–5)[22].

Here we systematically show which of the mammalian claudins are able to form TJ-like meshworks (Supplementary Fig. 2, 3). The observed ability to form TJ strands grossly (but not strictly) parallels the previous classification into classic versus non-classic claudins based on sequence homology[21]. TJ-like meshworks were formed by 11 out of 14 classic claudins but only two (Cldn11 and Cldn20) of the non-classic claudins formed a TJ-like meshwork. Notably, multiple claudins previously proposed to act as paracellular ion channels, e.g., Cldn4[61], 8[61], 16[16,17], and 17[62] were unable to form strands and may require co-assembly with meshwork- or other non-meshwork-forming claudins, as observed for Cldn16 and Cldn19[17] or Cldn4 and Cldn8[61]. Albeit a uniform distribution of Cldn4 and Cldn8 was observed here in fixed and living cells further analyses are required to determine the precise nanoscale architecture and organization of non-meshwork forming claudins in living cells. In the future, classification of nanoscale intensity features of non-meshwork forming claudins in living cells could reveal new polymerization principles, the interplay of cis/trans interactions and general structure-function relationships for these incompletely characterized claudin family members.

A different clustering analysis approach based on clustering of intensity texture as well as meshwork morphology features identified three distinct groups of meshwork forming claudins (Fig. 2). Class A and B showed distinct mesh sizes and contained both channel and barrier claudins. Class C comprised exclusively strongly sealing claudins (Cldn1, Cldn6, Cldn9, and Cldn11) and leads us to propose that dense TJ meshworks with parallel strands, as found solely in this class, perform sealing functions. Overall, our analysis of claudin meshworks suggests that there is no strict correlation between meshwork organization and channel or barrier function of individual claudins, confirming earlier work using FFEM on a limited set of claudins[14]. However, it is important to note that unsupervised machine learning can only produce classification hypotheses that then need to be tested by further experimental analyses, e.g., by super-resolution imaging, genetic manipulations or paracellular flux analysis.

Since TJs are formed by multiple claudins, we combined multi-color STED microscopy with colocalisation analysis and FRET to investigate claudin interactions on a nanoscale. We found that barrier-forming claudins can intermix and that channel-forming claudins segregate from barrier-forming claudins and other channel-forming claudins when overexpressed in non-polarized cells, epithelial cell lines, and at endogenous level in resorptive tissues (Figs. 3, 4). Individually and co-expressed barrier claudins Cldn1 and Cldn3 appeared as homogenous strands in FFEM[63], produce high colocalisation signals in STED images and high FRET values consistent with a highly intermixed organization of this and maybe other barrier claudin combinations. Conversely, very small clusters were found in FFEM images of reconstituted Cldn2 combined with Cldn1 or Cldn3 and endogenous Cldn2[8,10] but their molecular identity was so far unknown. Cluster from Cldn2 FEEM images could reflect differential association of Cldn2 with the protoplasmic and extracellular FFEM surfaces[63] or alternatively a segregation pattern of Cldn2 from Cldn1 or Cldn3. Trans interactions reported between Cldn2 and Cldn3, but not between Cldn2 and Cldn1[63] are mirrored by higher STED colocalisation of Cldn2/3 compared to Cldn2/1. We speculate that stable trans interactions between Cldn2 and Cldn3[63] might lead to very short Cldn2 strands that cannot be resolved by STED microscopy. Such short strands may underlie the high degree of colocalisation between Cldn3 and Cldn2 compared to that of Cldn3 with other channel-forming claudins, e.i., Cldn10a, Cldn10b, or Cldn15.

Therefore the balance of cis and trans interactions can finetune the segregation properties of claudins as also observed by exchanging extracellular loops of Cldn2 and Cldn10a (Fig. 5h, i).

Neither Cldn2 channel inactivation by MTSET, cholesterol depletion, nor the deletion of the entire claudin C-terminus or the PDZ motif required for phosphorylation or ZO-1 binding respectively affected the nanoscale segregation. Unexpectedly, we observed smaller, unstructured, and less frequent meshworks formed by combinations of C-terminal or PDZ deletion mutants of Cldn2/10a in COS-7 cells. Conversely, FFEM and live imaging data of C-terminal or PDZ deletion mutants of Cldn1, Cldn2, or Cldn3 expressed in non-polarized fibroblasts were reported to lead to unperturbed strand and meshwork morphologies, albeit showing higher meshwork dynamics[8,23,63]. We therefore speculate that segregated meshworks could be more dynamic and unstable, especially at segregation break points with possibly imperfect interaction interfaces. Here ZO-1 interactions could be required to stabilize segregated meshworks. Future work based on live-cell super-resolution microscopy could characterize nanoscale strand dynamics, strand breaks, and molecular factors affecting the barrier properties of claudin homo- and hetero-polymers in TJs.

The extracellular loops of channel claudins form not only the paracellular ion pore through structural organization, but together with transmembrane helices[64] and other interacting surfaces[65] contribute to strand formation and segregation. Molecular modeling based on the crystal structure of Cldn15[19,66] and experimental data[67] indicate that multiple claudin molecules interacting in cis and trans are implicated in pore formation[19,57,64–66]. The segregation mechanism described here could ensure that these pores are not disturbed by residues of ECLs from nearby barrier- or channel-forming claudins. The maximum length of a segregated strand on the other hand must be smaller than the total length of a typical mesh to increase the probability of integration of a selective ion pore that spans all meshwork strands (Supplementary Fig. 13). Finally, discontinuities in the barrier that lead to non-specific leaks must be prevented when TJ strands consisting of claudins with different permeability properties are formed. These non-specific leaks could be sealed by close membrane contacts formed at strand boundaries or by a so far undefined transmembrane protein enriched at TJs.

The integration of non-meshwork forming Cldn16 into Cldn19 strands or the induction of strands by copolymers of the two non-meshwork forming Cldn4 and Cldn8 were not distinguishable from colocalization values of intermixed barrier strands by our STED microscope (Fig. 3; Supplementary Figs. 5 and 7), so could enable the formation of ion pores at so far unresolved interfaces that are different from the paracellular pore structures proposed for Cldn15 polymers[57,66]. The combination of integration, segregation, and exclusion observed for Cldn10b, Cldn16, and Cldn19 is in accordance with differential interactions of these claudins[43] and could lead to the observed mosaic spatial separation of paracellular $Na^+$ (Cldn10b) and $Mg^{2+}$ (Cldn3/16/19) transport found in cell–cell-contact areas of the thick ascending limb of Henle's loop of the nephron[43].

Because nanoscale segregation of barrier and channel claudins could ensure specific claudin polymerization and meshwork incorporation, as described above, and was observed in epithelial cells and in two resorbing tissues, it may be important for paracellular transport processes. To measure the barrier and channel properties of the segregated claudin combinations (Cldn2/Cldn10a and Cldn3/Cldn15) and their individual claudin components (Cldn2, Cldn3, Cldn10a, and Cldn15), claudins were reconstituted in MDCKII QKO cells, an epithelial cell line lacking five major claudins and thus the TJ meshwork[54]. Of note, very low levels of non-meshwork forming Cldn12 and Cldn16 were expressed in MDCKII QKO (Supplementary Fig. 9a), but these failed to colocalize with occludin or ZO-1 at the plasma membrane (Supplementary Fig. 9b) and did not copolymerize with Cldn2, Cldn3,

Cldn10a, and Cldn15 (Supplementary Fig. 10). The apparent absence of both claudins from the TJ could be explained by their inability to form strands and/or the absence of copolymerizing claudins such as Cldn19, as previously observed for Cldn16[17].

Reconstitution of Cldn3 in QKO showed that a single barrier claudin can form an efficient paracellular seal. When Cldn2 or Cldn15 were reconstituted in QKO cells a paracellular barrier containing Na[+] selective channels was formed with a high Na[+]/Cl[-] permeability that closely match predictions form molecular models of Cldn15 homopolymers[57,66]. Conversely, reconstitution of Cldn10a created barrier containing a Cl[-] selective channels consistent with the results from former studies[11,58,68]. Albeit based on a limited set of claudins we noted that barrier claudins show significantly lower permeability to the small molecule fluorescein than all channel-forming claudins tested. This could be a consequence of more complex meshworks and/or fewer strand breaks occurring in restituted barrier claudins and will be subject of further investigation.

Since single barrier or channel claudins can form tight or ion selective TJ strands in QKO cells, we conclude that segregation interfaces between different claudin segments do not form specific ion pores (Fig. 6; Supplementary Fig. 13). Coexpression of segregating Cldn2 and Cldn10a produces parallel Na[+] and Cl[-] permeability consistent with the results of an established Cldn2/Cldn10a Tet On/Off system in MDCKI cells from Curry et al. in 2020[58] and ion absorption in the proximal tubules of the nephron[10,68]. When segregating barrier Cldn3 and channel-forming Cldn15 are co-expressed we observe more specific ion fluxes (Na[+]/Cl[-] ~ 15) compared to Cldn15 alone (Na[+]/Cl[-] ~ 4), consistent with the predicted Na[+]/Cl[-] permeability of Cldn15 by Samantha et al.[57] and relevant for a selective Na[+] and nutrient uptake in the intestine[42].

We propose that segregation enables balanced, interspaced, and alternating incorporation of claudins with different permeability properties into common TJ meshworks. This can produce parallel fluxes of different ions or increase the charge selectivity of paracellular ion channels at the TJ. Thereby, the segregation/intermixing pattern of the claudins expressed critically contributes to regulation of paracellular permeability in a given tissue. Our model (Supplementary Fig. 13) that is based on predictions from Claude et al.[18] and Weber et al.[38] assures that sufficiently large pore stretches enable a balanced flux of oppositely charged ions, by having minimal occurrence of dead ends for ions, while ensuring paracellular barrier integrity.

## Methods

### Plasmid cloning

All used constructs are listed in Supplementary Dataset 1. For all standard cloning and sub-cloning approaches self-made chemically competent TOP10 were used. Commercial NEB 5-alpha chemically competent *E. coli* (New England BioLabs Inc., #C2987H) were used for site-directed mutagenesis and HIFI-assembly cloning. HB101 (*recA*-) (Promega GmbH, #L2011) bacteria were used for cloning and amplifying pLIB plasmid DNA. Standard PCR for amplification was performed by using the Phusion High-Fidelity DNA Polymerase (Thermo Fisher Scientific, #F530S) according to the manufacturer's instructions. All used primers in this study were synthesized by BioTeZ Berlin Buch GmbH and are listed in Supplementary Dataset 1. For restriction digest, fast digest enzymes from Thermo Scientific were used (Supplementary Dataset 1). All human claudin (Cldn) constructs are labeled with "hu" (huCldn). All murine Cldn constructs are labeled with an "m" (mCldn). For the following constructs pYFP-Cldns, pTrq2-Cldns and pGFP-Cldns an identical linker ("SLVPSSDP" (8 AA)) between fluorescent tag and Cldn sequence was used. The linker for pSNAP-Cldns and pHalo-Cldns contains three additional AA ("**LYK**SLVPSSDP" (11 AA)).

The pYFP-Cldn constructs (pYFP-huCldn1,-2,-3,-4,-8,-10a,-10b,-11,-11b,-12,-15,-16,-19a,-19b,-20,-22,-23), pmCldn3-YFP, phuCldn3-YFP,

phuCldn1-YFP as well as pTOPO-huCldn6 and pTOPO-huCldn9 were provided by Susanne Krug, Dorothee Günzel and Jörg Piontek (Clinical Physiology/Nutritional Medicine, Medical Department, Division of Gastroenterology, Infectiology, Rheumatology, Charité – Universitätsmedizin Berlin).

The pYFP-Cldn constructs (pYFP-huCldn5,-7,-12,-20,-24,-25 and pYFP-mCldn13,-14,-18.1,-18.2) and pCFP-huCldn17 were provided by Lorena Suarez from the Mertins Lab (Max-Delbrück-Center for Molecular Medicine (MDC), Berlin).

phuCldn3-C1 was generated by sub-cloning Cldn3 from pYFP-huCldn3 via PCR and restriction enzymes HindIII and SalI into p-C1 (generated from a pGFP-C1).

pYFP-huCldn17 was generated by sub-cloning Cldn17 from pCFP-huCldn17 via restriction enzymes BsrGI and SalI into pYFP-huCldn15.

pYFP-huCldn6 and pYFP-huCldn9 were generated by sub-cloning Cldn6 and Cldn9 from pTOPO-huCldn6 and pTOPO-huCldn9 via PCR and restriction enzymes BsrGI and EcoRV into pYFP-huCldn15.

pGFP-Cldn-C1 constructs (pGFP-huCldn2[I66C] and pGFP-mCldn26,−27) were de novo synthesized and obtained from General Biosystems (Chuzhou, Anhui, China). pGFP-huCldn2[I66C] construct is based on Weber et al. (2015).

pSNAP-C1 and pHalo-C1 were generated by sub-cloning SNAP from pSNAP$_f$ (New England BioLabs Inc., #N9183S) and Halo from pTUBB5-Halo (Addgene, #64691) via PCR and the restriction enzymes BshTI and BsrGI into pmCherry-C1.

pSNAP-Cldn constructs (pSNAP-huCldn1,-2,-3,-4,-8,-10a,-10b,-11,-15,-16,-19a,-22) were generated by sub-cloning Cldns from pYFP-Cldn constructs (pYFP-huCldn1,-2,-3,-4,-8,-10a,-10b,-11,-15,-16,-19a,-22) via PCR and the restriction enzymes BsrGI and SalI into SNAP-C1.

pSNAP-huCldn12 construct was de novo synthesized and obtained from General Biosystems (Chuzhou, Anhui, China).

pHaloCldn constructs (pHalo-huCldn10a,-15,16) were generated by sub-cloning Cldns from pYFP-Cldn constructs (pYFP-huCldn-10a,-15,-16) via PCR and the restriction enzymes BsrGI and SalI into pHalo-C1.

pTrq2-Cldn constructs (pTrq2-huCldn2,-3) were generated by sub-cloning huCldn2 and huCldn3 from pSNAP-huCldn2 and pSNAP-huCldn3 via restriction enzymes BsrGI and SalI into Trq2-C1.

pSNAP-Cldn PDZ binding motif deletion mutants (pSNAP-huCldn2[ΔPDZ],-3[ΔPDZ],-10a[ΔPDZ],-15[ΔPDZ]) were generated by site-directed mutagenesis PCR and following phosphorylation of the PCR product with a T4 Polynucleotide Kinase (Thermo Scientific, #EK0032) using pSNAP-huCldn2,-3,-10a,-15. For all constructs the last three amino acids of the Cldn sequence were deleted: pSNAP-huCldn2[ΔPDZ] "GYV" (amino acids 228–230), pSNAP-huCldn3[ΔPDZ] "DYV" (amino acids 218–220), pSNAP-huCldn10a[ΔPDZ] "AYV" (amino acids 224–226), and pSNAP-huCldn15[ΔPDZ] "AYV" (amino acids 226–228).

pSNAP-Cldn C-terminus deletion mutants (pSNAP-huCldn2[ΔCT],-3[ΔCT],-10a[ΔCT],-15[ΔCT]) were generated by PCR of huCldn2,-3,-10a,-15 from pSNAP-huCldn2,-3,-10a,-15, restriction digest with BsrGI and EcoRV and sub-cloning into pSNAP-huCldn15. For all constructs the c-terminal part except for a three amino acids overhang was deleted: pSNAP-huCldn2[ΔCT] "SQRNRSNYYDAYQAQPLATRSSPRPGQPPKVKSEF NSYSLTGYV" (amino acids 187–230), pSNAP-huCldn3[ΔCT] "CPPREKKY-TATKVVYSAPRSTGPGASLGTGYDRKDYV" (amino acids 184–220), pSNAP-huCldn10a[ΔCT] "DNNKTPRYTYNGATSVMSSRTKYHGGEDFKTTN PSKQFDKNAYV" (amino acids 183–226), and pSNAP-huCldn15[ΔCT] "GSDEDPAASARRPYQAPVSVMPVATSDQEGDSSFGKYGRNAYV" (amino acids 186–228).

pYFP-Cldn PDZ binding motif deletion and C-terminus deletion mutants (pYFP-huCldn10a[ΔPDZ/ΔCT],-15[ΔPDZ/ΔCT]) were generated by sub-cloning huCldn10a[ΔPDZ/ΔCT],-15[ΔPDZ/ΔCT] from pSNAP-huCldn10a[ΔPDZ/ΔCT],-15[ΔPDZ/ΔCT] via restriction enzymes HindIII and EcoRV into pYFP-Cldn15.

pSNAP-huCldn2[ECL10a] was generated with the NEBuilder HiFi DNA Assembly Cloning Kit (New England BioLabs Inc., #E5520S) according to the manufacturer's instructions in two independent approaches.

In the first approach pSNAP-huCldn2^(ECL1C10a) was generated by using pSNAP-huCldn2 as backbone and pSNAP-Cldn10a for generating ECL1C10a with Cldn2 homology arms. The ECL1 "MLLPSWKTSSYVGA SIVTAVGFSKGLWMECATHSTGI TQCDIYSTLLGLPADIQAAQA" (amino acids 25–82) of huCldn2 was substituted by the ECL1 "TTSNEWKVT-TRASSVITATWVYQGLWMNCAGNALGSFHCRPHFTI FKVAGYIQACRG" (amino acids 23–79) of huCldn10a. In the second approach pSNAP-huCldn2^(ECL1C10a) was used as backbone plasmid and pSNAP-huCldn10a for generating ECL2C10a with huCldn2 homology arms to sub-clone pSNAP-huCldn2^(ECL10a). The ECL2 "WNLHGILRDFYSPLVPDSMKFEIGE" (amino acids 138–162) of huCldn2 was substituted by the ECL2 "LYANKITTEFFDPLFVEQKYELGA" (amino acids 135–158) of Cldn10a.

pTrq2-huCldn2^(ECL10a) was generated via sub-cloning of huCldn2^(E-CL10a) from pSNAP-huCldn2^(ECL10a) using restriction enzymes BsrGI and SalI into Trq2-C1.

pLIB-CMV-FLAG-huCldn2,-10a,-15-Puro constructs were generated by sub-cloning Cldns from pSNAP-Cldn constructs (pSNAP-huCldn2,-10,-15) via PCR and the restriction enzymes BshTI and NotI into pLIB-CMV-GFP-N1-Puro.

pLIB-CMV-MCS2-Neo construct was generated by sub-cloning CMV-MCS2 from pLIB-CMV-MCS2-Puro via restriction enzymes BsrGI and EcoRI into pLIB-MCS2-Neo.

pLIB-CMV-GFP-N1-Neo construct was generated by sub-cloning pGFP-N1 from pLIB-CMV-GFP-N1-Puro via restriction enzymes EcoRI and NotI into pLIB-CMV-MCS2-Neo.

pLIB-CMV-FLAG-huCldn2,-10a-Neo constructs were generated by sub-cloning Cldns from pLIB-CMV-FLAG-Cldn-Puro constructs (pLIB-CMV-FLAG-huCldn2,-10a-Puro) via restriction enzymes BshTI and NotI into pLIB-CMV-GFP-N1-Neo and via restriction enzymes EcoRI and NotI into pLIB-CMV-MCS2-Neo.

pLIB-CMV-FLAG-huCldn3-Puro constructs were generated by sub-cloning Cldns from pSNAP-Cldn constructs (pSNAP-huCldn3) via PCR and the restriction enzymes EcoRI and SalI into pLIB-CMV-MCS2-Puro.

pLIB-CMV-FLAG-huCldn3-Neo constructs were generated by sub-cloning Cldns from pLIB-CMV-FLAG-Cldn-Puro constructs (pLIB-CMV-FLAG-huCldn3-Puro) via restriction enzymes EcoRI and SalI into pLIB-CMV-MCS2-Neo.

phuCldn15-C1 was generated by sub-cloning Cldn15 from pYFP-huCldn15 via PCR and restriction enzymes HindIII and SalI into p-C1 (generated from a pGFP-C1).

phuCldn6-C1 was generated by sub-cloning Cldn6 from pSNAP-huCldn6 via restriction enzymes HindIII and EcoRV into p-C1 (generated from a p phuCldn15-C1).

phuCldn12-N1 was generated by sub-cloning Cldn12 from pSNAP-huCldn12 via PCR and restriction enzymes HindIII and NotI into p-N1 (generated from a pSNAP-N1).

phuCldn16-C1 was generated by sub-cloning Cldn16 from pSNAP-huCldn16 and pYFP-huCldn16 via restriction enzymes HindIII and EcoRV into p-C1 (generated from a phuCldn15-C1).

All constructs and sub-constructs were verified by restriction digest and by Sanger Sequencing (LGC Genomics GmbH, Berlin).

### Fluorophore coupling to SNAP- and Halo-tag ligands
Fluorophore-labeled SNAP-tag ligands (BG-Atto590/BG-JF646) and Halo-tag ligands (CA-Atto590/CA-JF646) were chemically synthesized as described in Bottanelli et al.[69]. Atto590-NHS was obtained from Sigma-Aldrich (#79636-5MG-F) and JF646-NHS from Tocris (#6148). The SNAP-tag ligand BG-NH₂ was obtained from New England Biolabs Inc. (#S9148S) and the Halo-tag ligand HALOTag Amine (O2) ligand was obtained from Promega (#P6711). All ligands are listed in Supplementary Dataset 1.

### Mammalian cell culture
The following cell lines were used in the study. COS-7 (ATCC CRL 1651), HEK293 (ATCC CRL 1573), HEK293T (ATC CRL 3216), 3T3 fibroblasts

(DSMZ, no.: ACC 173), MDCKII (ECACC 00062107), and Caco-2 cells (ATCC HTB 37) were taken from own lab cell culture stocks. MDCKC7 cells were kindly provided by Lorena Suarez Artiles from the Mertins Lab (Max-Delbrück-Center for Molecular Medicine (MDC), Berlin). MDCKII quintuple claudin knock-out (QKO) cells were kindly provided by the lab of Mikio Furuse. MDCKII QKO cells stably expressing claudins were generated in this study. COS-7, HEK293, HEK293T, 3T3 fibroblasts, MDCKII, MDCKII QKO, and MDCKC7 were cultured in DMEM with high glucose (Gibco, Thermo Fisher Scientific, #11965084) supplemented with 10% FBS (Gibco, Thermo Fisher Scientific, #10082147) and 50 µg/ml penicillin-streptomycin (Pen-Strep) (Gibco, Thermo Fisher Scientific, #15140122) at 37 °C and 5% CO₂. MDCKII QKO cells stably expressing claudins were cultured as described above and under additional selection pressure of 2–5 µg/ml Puromycin (Invivogen, #ant-pr-1) and 300–500 µg/ml G418 (ant-gn-1, Invitrogen). Caco-2 were cultured in MEM (Gibco, Thermo Fisher Scientific, #11095080) containing 15% FBS and 50 µg/ml Pen-Strep at 37 °C and 5% CO₂. PBS (Gibco, Thermo Fisher Scientific, #14190144) was used for all washing steps. For transfection, cells were seeded on 18 mm or 25 mm glass coverslips (#1.5H) (Thermo Fisher Scientific, #CB00250RAC33MNT0) or in µ-Slide 8-well glass bottom dishes (Ibidi, #80827) coated with 2% matrigel in DMEM in a confluency of 70%. After 24 h the transfection was performed with Lipofectamine 2000 Transfection Reagent (Thermo Fisher Scientific, #11668019) according to the manufacturer's instructions. SNAP-tag or/and Halo-tag labeling was performed 24 h after transfection. The transfected cells were incubated for 1 h at 37 °C and 5% CO₂ with 2 µM BG-Atto590 (SNAP-tag ligand) or BG-JF646 (SNAP-tag ligand) and/or 1 µM CA-Atto590 (Halo-tag ligand) or CA-JF646 (Halo-tag ligand) in DMEM, intensively washed and post-incubated in DMEM for another 30 mins at 37 °C and 5% CO₂. For electrophysiological measurements, MDCKII, MDCKII QKO, and MDCKII QKO cells stably expressing claudins were seeded in a cell density of $1.5 \times 10^5$ cells on 0.6 cm² PCF transwell filter with a pore size of 0.4 µm (Fisher Scientific, Millipore Millicell insets, #10126240) and cultivated for 5–7 days at 37 °C and 5% CO₂.

### Retrovirus production via calcium phosphate transfection of HEK293T cells
HEK293T cells were seeded in 6-well plates or in 10 cm petri dishes and were transfected with plasmid mix of packaging plasmid DNA pCIG3.NB, lentiviral envelope plasmid pMD2.G, and genomic plasmid DNA (pLIB-CMV-FLAG-Cldn2/3/10a/15-Puro/-Neo) by using calcium phosphate. A plasmid coding for GFP-N1 served in every transfection round as a transfection efficiency control. After 24 h, the cells were checked with a basic fluorescent microscope for transfection efficiency. The medium was changed, and the medium volume was reduced by 20%. Every 48 h for 6 days, the supernatant was collected, and 8 ml of fresh medium was added. The supernatant was centrifuged for 5 min at $1000 \times g$ to remove cell debris, filtered (0.45 µm pore size) and transferred in a fresh 50 ml tube. All supernatants were stored at 4 °C for up to 2 weeks or at −80 °C for long-term storage. To concentrate the virus, all collected supernatants were pooled in Amicon Ultra-15 (100 kDa) tubes and spun down for 20 min at $4696 \times g$. The concentrated supernatants were collected and stored at −80 °C.

### Retroviral generation of stable cell lines
For the generation of stably expressing cell lines, MDCKII QKO were seeded at a confluency of 60–70% in 6-well plates or 10 cm petri dishes containing DMEM supplemented with 10% FBS and 1% Pen/Strep. After 24 h, the medium was reduced to 1.5 ml for 6-well plates or to 6 ml for 10 cm petri dishes. 0.5–1 ml of non-concentrated or 20–80 µl of concentrated virus was used to infect the cells. GFP-N1 transducing virus served as infection control. After 48–72 h, the control cells were checked with a fluorescence microscope for transduction efficiency indicated by the number of cells expressing free GFP. For the initial

selection of successfully infected cells and non-infected cells, up to 10 µg/ml Puromycin and 800 µg/ml G418 were used. The growth medium was changed to DMEM with 10% FBS but without Pen/Strep when G418 was used. The cells were routinely checked for viability and expression level for about one week. After one week, the selection pressure was reduced to 2–5 µg/ml Puromycin and 300–500 µg/ml G418. The selection process was continuously checked and the expression of the claudins was controlled by immunofluorescent staining and immunoblotting.

## Antibodies

All used antibodies and their dilutions are listed in Supplementary Dataset 1.

## Antibody fluorophore conjugation

Donkey anti-rabbit Atto542, donkey anti-mouse Atto542, and rabbit anti-Cldn3 Atto590 were produced by incubating 100 µl donkey anti-rabbit (AffiniPure Donkey Anti-Mouse IgG (H + L) from Jackson Immuno Research Ltd., #711-005-152) and 100 µl donkey anti-mouse (AffiniPure Donkey Anti-Mouse IgG (H + L) from Jackson Immuno Research Ltd., #715-005-151) with a 5–10x excess of Atto542-NHS (AttoTEC, #AD542-31) and rabbit anti-Cldn3 (Thermo Fisher Scientific, #34-1700) with a 5–10x excess of Atto590-NHS (Sigma-Aldrich, #79636-5MG-F) for 1 h at RT under constant agitation. The labeled antibody was purified by using Zeba™ Spin Desalting Columns (7 K MWCO, 0.5 ml) (Thermo Fisher Scientific, #89883) according to the manufacturer's instructions. The degree of antibody labeling was determined by absorbance measurements via NanoDrop (Nanodrop ND-1000, Thermo Fisher Scientific).

## Immunocytochemistry

On glass coverslips (#1.5) seeded and transfected COS-7, 3T3 fibroblasts, MDCKC7 and Caco-2 were washed with PBS supplemented with 0.5 mM magnesium and 1 mM calcium (PBS⁺), fixed with 37 °C prewarmed 4% PFA/sucrose for 10 min, permeabilized with 0.2% Triton X-100 in PBS⁺ for 5 min and blocked in blocking solution containing 10% NGS, 1% BSA, 0.05% Tween-20 dissolved in PBS⁺. For YFP-boosting, cells were incubated with anti-GFP Atto647N nanobody (α-GFP-NB-Atto647N; Chromotek, #gba647n-100, 1:200), anti-GFP Atto594 nanobody (α-GFP-NB-Atto594; Chromotek, #gba594-100, 1:200) or anti-GFP Atto488 nanobody (α-GFP-NB-Atto488; Chromotek, #gba488-100, 1:200) for 1 h at room temperature or with a mouse anti-GFP antibody (Thermo Fisher Scientific, #A-11120, 1:500) in blocking solution overnight at 4 °C. MDCKII, MDCKII QKO and MDCKII QKO stably expressing FLAG-tagged claudins were seeded on transwell filters were cultivated for 5–7 days, washed with PBS⁺, fixed with ice-cold ethanol at −20 °C for 20 mins and blocked in blocking solution containing 10% NGS, 1% BSA, 0.05% Tween-20 dissolved in PBS⁺. All used primary antibodies, rabbit anti-ZO-1 (Thermo Fisher Scientific, #61-7300), mouse anti-ZO-1 (Thermo Fisher Scientific, #33-9100), mouse anti-Occludin (Thermo Fisher Scientific, #33-1500), rabbit anti-Calreticulin (Abcam, #ab92516), mouse anti-Cldn2 (Thermo Fisher Scientific, #32-5600), rabbit anti-Cldn3 (Thermo Fisher Scientific, #34-1700), rabbit anti-Cldn10 (antibodies-online.de, ABIN3183935), rabbit anti-Cldn12 (IBL America, #18801), rabbit anti-Cldn15 (Thermo Fisher Scientific, #38-9200), mouse anti-Cldn16 (gift from Prof. Henrik Dimke) were incubated in blocking solution overnight at 4 °C. For secondary antibody labeling the following antibodies were incubated in blocking solution for 1 h at room temperature: donkey anti-mouse Atto542 (Atto542-NHS from AttoTEC, #AD542-31 and AffiniPure Donkey Anti-Mouse IgG (H + L) from Jackson Immuno Research Ltd., #715-005-151, 1:200), goat anti-mouse Alexa Fluor Plus 594 (Thermo Fisher Scientific, #A32744, 1:200), goat anti-mouse Atto647N (Active Motif, #15058, 1:200), donkey anti-rabbit Atto542 (Atto542-NHS from AttoTEC, #AD542-31 and AffiniPure Donkey Anti-Rabbit IgG (H + L) from Jackson Immuno Research Ltd., #711-005-151, 1:200), goat anti-rabbit

Alexa Fluor Plus 594 (Thermo Fisher Scientific, #A32740, 1:200) or goat anti-rabbit Atto647N (ActiveMotif, #15048, 1:200). For actin staining Phalloidin Alexa Fluor 488 (Thermo Scientific Fisher, #A12379, 1:1000) was used. For nuclei staining DAPI (Thermo Scientific Fisher, #62248, 1:1000) was used. Cells were washed with PBS⁺ and mounted in Pro Long Gold Antifade (Thermo Fisher Scientific, #P36934) on glass slides (Thermo Fisher Scientific, VWR, #630-1985). The slides were cured for 24–48 h at room temperature and stored at 4 °C.

## Immunohistochemistry of murine duodenum

C57BL/6J mice were sacrificed by cervical dislocation. The Landesamt für Gesundheit und Soziales (LAGeSo) Berlin with their permission under the license T-025/16 approved the use of animals for this study. The whole duodenum was taken out, washed with a Ringer solution (7.2 g NaCl, 0.17 g CaCl₂, 0.37 g KCl) and dissolved in reagent-grade H₂O, cut in lateral direction into multiple 1–2 cm parts, embedded in Tissue-Tek (Sakura Finetek, Fisher Scientific, #12351753) and immediately frozen in liquid nitrogen. 10 µm thick longitudinal slices were cut with a cryostat (Microm HM 560 Cryostat, Thermo Fisher Scientific) and placed on acid cleaned and organosilan coated glass coverslips (#1.5H). The tissue sections were fixed with −20 °C cold ethanol for 20 min, washed with PBS⁺, blocked with PBS⁺ containing 6% NGS, 1% BSA and 0.05% Tween-20 for 1 h at room temperature and stained with rabbit anti-Cldn3 (Thermo Fisher Scientific, #34-1700, 1:100) overnight at 4 °C. After 24 h the slices were incubated with goat anti-rabbit Alexa Fluor Plus 594 (Thermo Fisher Scientific, #A32740, 1:200) for 1 h at room temperature followed by intensively washing and an over blocking step using donkey anti-rabbit (Jackson Immuno Research Ltd., #711-005-152, 1:200) in a 10x excess for 2 h at room temperature. The samples were washed intensively with PBS⁺. As control the over blocked slices were incubated with a goat anti-rabbit Alexa Fluor 488 (Thermo Fisher Scientific, #A-11008, 1:200) for 1 h at room temperature. The samples were washed intensively with PBS⁺ and incubated with rabbit anti-Cldn15 (Thermo Fisher Scientific, #38-9200, 1:100) overnight at 4 °C. For secondary labeling goat anti-rabbit Atto647N (Active Motif, #15048, 1:200) for 1 h at room temperature was used. The tissue was washed and mounted using Pro Long Gold Antifade (Thermo Fisher Scientific, #P36934) on object slides and incubated for 24 h at room temperature. For every sample, the control staining with Alexa Fluor 488 was imaged to verify the quality of the over blocking step.

## Immunohistochemistry of single murine tubules from proximal tubule

C57BL/6J mice were sacrificed by decapitation and kidneys were removed immediately. The "Ministerium für Landwirtschaft, Umwelt und ländliche Räume" in Schleswig-Holstein (MELUND SH) with their permission under the animal ethics protocol number V312-72241.121-2 approved the use of animals for this study. After de-capsulation, the middle section of each kidney was sliced in transversal direction in fine (0.2–0.4 mm) section and these sections were transferred into prewarmed incubation solution (in mM: 140 NaCl, 0.4 KH₂PO₄, 1.6 K₂HPO₄, 1 MgSO₄, 10 sodium acetate, 1 α-ketoglutarate, 1.3 calcium gluconate, 5 glycine, supplemented with 48 mg/l trypsin inhibitor and 25 mg/l DNase I, pH 7.4) containing 2 mg/ml collagenase II (pan biotech). Enzymatic digestion was performed at 37 °C in a thermo-shaker for 15 min. Free-floating tubular segments were transferred into ice-cold sorting solution (incubation solution supplemented with albumin 0.5 mg/ml) for washing. Tubular segments were allowed to settle, and the supernatant was replaced by fresh ice-cold sorting solution for at least two times to remove erythrocytes and cellular debris. Washed tubular segments were transferred to a dissection microscope and proximal tubules were identified and sorted. After transfer to poly-lysine coated slides (superfrost, Thermo Fisher, #11976299) and short settlement, tubules were fixed with 4% PFA for 7 min and PFA

subsequently vigorously removed from the slide under visual control and washing with PBS containing 0.3% Triton X-100 (PBS-T100). Primary antibodies (either mouse anti-Cldn2 (Thermo Fisher Scientific, #32-560), rabbit anti-Cldn10 (Antibodies-online, #ABIN3183935), or rabbit anti-Cldn2 (Thermo Fisher Scientific, #51-6100) and mouse anti-Cldn10 (Thermo Fisher Scientific, #41-5100), 50 µl/slide) were incubated at 4 °C overnight. After extensive washing under visual control (PBS-T100), secondary antibodies goat anti-mouse Alexa Fluor Plus 594 (Thermo Fisher Scientific, #A32744, 1:200) and goat anti-rabbit Atto647N (AttoTEC, #15048, 1:200) were incubated for 1–2 h at room temperature. After final vigorous washing with PBS-T100, embedding was performed with Pro Long Gold Antifade (Thermo Fisher Scientific, #P36934) together with #1.5H coverslips.

## Cell lysates

COS-7 were seeded in 6-well plates or 10 cm petri dishes and after 24 h incubation scraped with 100–500 µl lysis buffer (1% Triton X-100, 20 mM HEPES, pH 7.4, 130 mM NaCl, 10 mM NaF, 0.03% PIC) on ice. MDCKC7, Caco-2, MDCKII, MDCKII QKO, and MDCKII QKO FLAG-tagged or untagged Cldns expressing cells were seeded in 6-well plates and after 5–7 days scraped with 100 µl lysis buffer (1% Triton X-100, 20 mM HEPES, pH 7.4, 130 mM NaCl, 10 mM NaF, 0.03% PIC) on ice. The solutions were transferred into pre-cooled 1.5 ml vials and incubated for 30 min on ice. The protein-containing supernatant was isolated via centrifugation at $17,000 \times g$ for 20 min at 4 °C. 1–10 µl of the supernatant was incubated with 490–499 µl $H_2O$ and 500 µl 2x Bradford reagent for 5 min and the protein concentration was determined by measuring the OD595 with a photometer (BioPhotometer plus, Eppendorf). The protein lysates were denaturized with 6x Laemmli buffer (0.375 M Tris pH 6.8, 12% SDS, 60% glycerol, 0.6 M DTT, 0.06% bromophenol blue) for 5 min at 95 °C and stored at −20 °C.

## Immunoblot-based analysis

For protein separation via SDS-PAGE (sodium dodecylsulfate polyacrylamide gel electrophoresis) 15–30 µg of lysate was loaded on a 10–12% polyacrylamide gel or 4–15% MINI-PROTEAN TGX Precast Gels (Bio-Rad, #4561083/86). The PageRule Plus Prestained (Thermo Fisher Scientific, #26620) or PageRuler Prestained (Thermo Fisher Scientific, #26616) were used as protein ladder. The gel was run in SDS-running buffer (25 mM Tris base, Glycine 0.192 M, SDS 0.1%) at 120 Volt for 60–90 min using the Mini-PROTEAN Tetra Vertical Electrophoresis Cell (Bio-Rad, #1658004). The protein transfer on a nitrocellulose membrane was done by Wet Blot in a 20% methanol containing transfer buffer (25 mM Tris-HCl (pH 7.6), 192 mM glycine, 20% methanol, 0.03% SDS) for 90 min at 110 V at 4 °C. Ponceau staining was performed to check the quality of the protein transfer. It was removed with $H_2O$ and 0.1% acidic acid. The membranes were blocked with PBS containing 0.05% Tween-20 (PBS-T) supplemented with 3% BSA, TBS containing 0.05% Tween-20 (TBS-T) supplemented with 5% milk or Intercept (PBS) Blocking Buffer (LI-COR, #927-70001) for 1 h at room temperature. Primary antibodies were incubated overnight in blocking solution under constant agitation at 4 °C. After washing with PBS-T or TBS-T, HRP conjugated goat anti-mouse (Jackson Immuno Research Ltd., #115-035-003) and HRP conjugated goat anti-rabbit (Jackson Immuno Research Ltd., #111-035-003) were used in a dilution of 1:2000 in PBS-T with 3% BSA or TBS-T with 5% milk for 1 h at room temperature or IRDye 800CW conjugated donkey anti-mouse (LI-COR, #926-32212), IRDye 800CW conjugated donkey anti-rabbit (LI-COR, #926-32213), were used in a dilution of 1:15000 in 50% LI-COR blocking solution and 50% PBS-T for 1 h at room temperature. Membranes with HRP conjugated secondary antibodies were washed with PBS-T or TBS-T and incubated with Pierce™ ECL Western Blotting-Substrate (Thermo Fisher Scientific, #32209) for 5 min at room temperature. For imaging, the ChemiDoc XRS+ (Bio-Rad) controlled by the Image Lab software (version 6.0.1) was used. For quantification, the proteins of interest

were normalized to their loading controls. Membranes with LI-COR secondary antibodies were washed with PBS-T and once more with PBS without Tween-20 and imaged with LI-COR Odyssey Fc imaging system controlled by the Image Studio software (version 5.2). Colorimetric analysis of the protein bands was performed with Fiji ImageJ or with Image Studio Lite (LI-COR Biosciences) on raw Western Blot images. The loading control was used for the comparison of different samples. Overexposed bands were excluded from this analysis because of the loss of linearity. The brightness and contrast of Western Blot images were only changed for presentational reasons. Uncropped Western Blot are shown in the Source data.

## Confocal imaging

Confocal images of fixed cells and tissue were acquired with an LSM780 from Carl Zeiss Microscopy and with the Leica SP8 TCS STED microscope (Leica Microsystems).

For the detection in the LSM780, a photomultiplier was used. The LSM780 was controlled by the Zeiss ZEN2010 software (Carl Zeiss Microscopy). Single- and multi-color confocal imaging of fixed samples was performed in sequential mode with the following fluorophore-specific excitation (Ex.) and emission filter (EmF.) settings: Alexa Fluor Plus 594 (Ex.: 561 nm; EmF.: 566–630 nm), Atto647N (Ex.: 633 nm; EmF.: 636–740 nm). Images were acquired with a PL APO DIC M27 63×/1.40 NA oil objective (Carl Zeiss Microscopy).

For the detection in the Leica SP8 TCS STED microscope, two hybrid detectors (HyDs) were used. The system was controlled by the Leica LAS X software. Single- and multi-color STED imaging of fixed samples was performed in sequential mode with the following fluorophore-specific excitation (Ex.) and emission filter (EmF.) settings: Alexa Fluor 488 (Ex.: 488 nm; EmF.: 498-560 nm), YFP (Ex.: 514 nm; EmF.: 524–568 nm), Alexa Fluor Plus 594 (Ex.: 590 nm; EmF.: 600–640 nm), Atto647N (Ex.: 640/650 nm; EmF.: 650/660–700 nm). Images were acquired with a HC PL APO CS2 100×/1.40 NA oil objective (Leica Microsystems).

## Time-gated single- and multi-color STED imaging

STED images were taken with a Leica SP8 TCS STED microscope (Leica Microsystems) equipped with a pulsed white-light excitation laser (WLL; ~80 ps pulse width, 80 MHz repetition rate (NKT Photonics)) and two STED laser for depletion at 592 nm and 775 nm. The microscope was housed in a heatable incubation chamber (LIS Life Imaging Services). The system was controlled by the Leica LAS X software. Single- and multi-color STED imaging of fixed samples was performed in sequential mode with the following fluorophore-specific excitation (Ex.) and emission filter (EmF.) settings: YFP (Ex.: 514 nm; EmF.: 524–568 nm), Atto542 (Ex.: 540 nm; EmF.: 550–580 nm), Atto590 (Ex.: 590 nm; EmF.: 600–640 nm), Alexa Fluor Plus 594 (Ex.: 590 nm; EmF.: 600–640 nm), JF646 (Ex.: 640 nm; EmF.: 650–700 nm), Atto647N (Ex.: 640 nm; EmF.: 650–700 nm). For all emissions, the 775 nm STED laser was used only YFP was depleted with the 592 nm depletion laser. Time-gated detection was set from 0.3–6 ns. The fluorescence emission signal was collected by two HyDs. Images were acquired with a HC PL APO CS2 100×/1.40 NA oil objective (Leica Microsystems), a scanning format of 1024×1024 pixels, 8-bit sampling, 16x line averaging and 6x optical zoom, yielding in a pixel size of 18.9 × 18.9 nm. In addition, to every STED image a confocal image with the same settings but 1x line averaging was acquired.

## One- and two-color live STED imaging

Live STED imaging was performed with COS-7 seeded on 25 mm glass coverslips (#1.5H) using an Attofluor cell chamber (Thermo Fisher Scientific, #A7816) or in µ-Slide 8-well glass bottom dishes (Ibidi, #80827). The glass surface was coated with 2% matrigel. The cells were transfected with SNAP-tag or Halo-tag constructs. A HEPES buffered live imaging solution (Thermo Fisher Scientific,

#A14291DJ) was used. 24 h prior imaging, the heatable incubation chamber was set to 37 °C to provide focus stable imaging. For single-color live STED imaging JF646 (Ex.: 640 nm; EmF.: 650–700 nm) was used. The imaging settings were the same as described above (see the "Time-gated single- and multi-color STED imaging" section) except for the reduced scanning format of 512 × 512, increased 16–32x line averaging, and a 12x optical zoom yielding in a pixel size of 18.9 × 18.9 nm. The acquisition time per frame for serial imaging was set to 10 sec/frame.

Two-color Live STED imaging was performed with the JF646 (Ex.: 640 nm; Em.: 650–700 nm) as SNAP ligand, Atto590 (Ex.: 590 nm; Em.: 600–640 nm) as Halo ligand or YFP. For the combination of pSNAP-Cldn3 with pYFP-Cldn4 and pSNAP-Cldn8 with pYFP-Cldn4 the imaging settings were the same as described above (see the "Time-gated single- and multi-color STED imaging" section). For the combination pSNAP-Cldn2 and pHalo-Cldn10a a reduced scanning format of 512 × 512, 16x line averaging and a 12x optical zoom yielding in a pixel size of 18.9 × 18.9 nm was used. Imaging was performed in an acquisition speed of 10 sec/frame. For imaging of the combination of pSNAP-Cldn3 and pHalo-Cldn15 a resonance scanner at 8000 Hz was used. Single live images were taken with a scanning format 512 × 512, 32x line averaging and a 12x optical zoom yielding in a pixel size of 18.9 × 18.9 nm. Imaging was performed in an acquisition speed of 19.5 sec/frame. For further imaging processing we used noise2Void[70] a deep learning-based image restoration method to remove noise from images for visualization as specified in the figure legends. We trained the noise model in Fiji on a GPU with 150 epochs and 200 steps per epoch on a large set of training data with a batch size of 100 and a dimension of 180 × 60 px. The neighborhood radius was set to 5. After training the best model was chosen for predicting the image with filtered noise.

## FWHM measurements
Full-width half-maximum (FWHM) measurements were performed with pSNAP-huCldn3 or pSNAP-huCldn11 transfected COS-7 cells labeled with BG-JF646. The same imaging settings, except for temperature and imaging solution differences (fixed: PBS at room temperature; live: live imaging solution at 37 °C), were used for live and fixed cell imaging. For Cldn3 line profiles (straight line 0.3 μm length, 10 px = 189 nm width) of 20 single strands per TJ-like meshwork (8 TJ-like meshworks in total) were analyzed for FWHM and average fluorescence intensity by Gaussian Fitting using Fiji ImageJ. For Cldn11 only line profiles of 20 strands in one TJ-like meshwork were measured. The FWHM was determined by multiplication of sigma with the factor 2.35.

## Automated TJ-like meshwork analysis
**Cluster analysis of claudins.** A dataset of 29 claudins with in total 168 images with an average number of 5 images per claudin was used for this analysis. Analysis for texture and meshwork analysis was performed within a single 200 × 200 px crop per image. Crops were defined by an expert annotator when possible in the center of a structure, avoiding bright potential artefacts and outside edges.

Images were filtered for the texture analysis using a 3 px Gaussian filter kernel and a sigma of 1. Gray level co-occurrence matrices were computed at distances 1, 3, 5, and 10 px and Haralick texture features[32] were extracted from 4 angles and averaged using the python mahotas library[71]. Three texture features (Sum average, sum variance and sum of squares: variance) were selected based on literature[31].

For analyzing all claudins, the extracted texture features from each ROI were averaged. For clustering we then used pheatmap implemented in R, unit variance scaling was applied. Clustering was computed using euclidean distance and average linkage. This clustering was then visually checked and meshwork former and non-meshwork former classified based on visual confirmation. Code

and example images provided at (https://doi.org/10.5281/zenodo.7009994).

**Cluster analysis of meshwork former claudins.** Claudins forming meshworks were selected from the dataset described in the section "Cluster analysis of claudins" and a further dataset was added where only meshwork forming claudins were imaged. This new dataset consisted of 15 claudins with in total 202 images. $n$(Cldn10a) = 14; $n$(Cldn7) = 12; $n$(Cldn19a) = 14; $n$(Cldn19b) = 14; $n$(Cldn20) = 3; $n$(Cldn2) = 13; n(Cldn14) = 17; $n$(Cldn5) = 10; $n$(Cldn15) = 20; $n$(Cldn10b) = 12; $n$(Cldn3) = 7; $n$(Cldn1) = 21; $n$(Cldn6) = 9; $n$(Cldn11) = 6; $n$(Cldn9) = 7. The clustering was performed on image features extracted by the same Haralick texture feature approach with additional image features extracted by image segmentation and analysis. A 200 × 200 px ImageJ Region of interest (ROI) was defined per image by an expert annotator using the previously use criteria (Cluster analysis of claudins) and both Haralick texture feature extraction and image analysis was performed within this ROI.

Images were filtered for the texture analysis using a 3 px Gaussian filter kernel and a sigma of 1[72]. Gray level co-occurrence matrices were computed at distances 1, 3, 5, and 10 px and Haralick texture features[32] were extracted from 4 angles and averaged using the python mahotas library[71]. Three texture features (sum average, sum variance, and sum of squares: variance) were selected based on literature[31].

For segmenting the meshwork, the images were filtered using Multiscale Oriented-Flux Tubularity filter[73] implemented in the Fiji plugin[74] simple neurite tracer[75]. A fixed threshold value was then applied to the filtered image and the largest connected region[76] comprising the meshwork was kept. Within the ROI the percent of segmented area was measured. A skeleton analysis[77] was performed to measure within the ROI the number of branches, average and maximum branch length, number of total junctions as well as triple and quadruple junctions. The binary mask was inverted to measure the mesh size within the ROI, excluding objects touching the edge. The number of meshes, average mesh size, and variance of the mesh size was measured.

For clustering the meshwork forming claudins, both the meshwork analysis and texture analysis features were used. ROIs with a segmented area of less than 10% or more than 90% were excluded as well as meshworks with less than 10 branches. The features were then averaged for each ROI. ClustVis server[78] was used for visualizing the clustering with ln(x)-transformation applied to the values, the rows were centered, and unit variance scaling was applied. Clustering was computed using correlation distance and average linkage. Code and example images provided at (https://doi.org/10.5281/zenodo.7009994).

## Pearson correlation analysis
COS-7, Caco-2, or MDCK-C7 cells were seeded on with 2% matrigel-covered coverslips and transfected with equal amounts of two or three plasmids coding YFP-, SNAP- or Halo-tagged claudins. For a reproducible Pearson analysis over several different experiments, the imaging parameter were set with the following positive controls set as reference: pSNAP-Cldn2/pYFP-Cldn2, pSNAP-Cldn3/pYFP-Cldn3, pSNAP-Cldn19a/pYFP-Cldn19a and then kept constant for the rest of an experiment. In general, the cells were labeled with the SNAP-ligand BG-Atto590 and GFP-booster α-GFP-NB-Atto647N. For the pSNAP-Cldn2^ECL10a, pSNAP-Cldn10a^ΔPDZ/ΔCT/pYFP-Cldn10a^ΔPDZ/ΔCT and pSNAP-Cldn15^ΔPDZ/ΔCT/pYFP-Cldn15^ΔPDZ/ΔCT Pearson analysis the combination of SNAP-ligand BG-JF646 and mouse anti-GFP with anti-mouse Alexa Fluor Plus 594 was used. For the triple claudin expression experiments the combination of Halo-ligand CA-JF646, SNAP-ligand BG-Atto590, and mouse anti-GFP with anti-mouse 2^nd-Atto542 was used. For every condition at least 5 meshworks over 3 independent experiments with the same settings (1024 × 1024, 16x line averaging, pixel size of

18.9 × 18.9 nm) were imaged. From every meshwork a ROI with a representative signal of the transfected claudins was picked and the Pearson above threshold was measured with a Coloc2-based script in Fiji ImageJ with the PSF set to 2 px = 38.8 nm.

At least >30 STED images from three different isolated proximal tubules stained with anti-Cldn2 mouse, anti-mouse Alexa Fluor Plus 594 and anti-Cldn10 rabbit, anti-rabbit Atto647N (see: Immunofluorescence of single murine tubules from proximal tubule) were taken and imaged with the same settings (1024 × 1024, 16x line averaging, pixel size of 18.9 × 18.9 nm). Out of every meshwork, a ROI with a representative signal of the stained claudins was picked and the Pearson above threshold was measured with a Coloc2-based script in Fiji ImageJ with the PSF set to 2 px = 38.8 nm.

### Fluorescence resonance energy transfer (FRET) measurement

HEK293 cells were transfected with pTrq2- and pYFP-Cldn constructs. A plasmid DNA ratio of 1:1 was used for all pTrq2-Cldn3 approaches. For pTrq2-Cldn2 or pTrq2-Cldn2$^{ECl10a}$ the ratio was changed in all approaches to 1:5. Transfected HEK293 cells were visualized on a LSM510-NLO inverted microscope (Carl Zeiss Microscopy GmbH, Jena, Germany) using a 40x/1.3 oil objective. Trq2 fluorescence signals were recorded (IR laser, λexc = 810 nm, two-photon technique, META detector, spectral range 436–650 nm) and split using a MBS KP 700. Channel pictures were taken prior to the recording of spectra in order to estimate expression of the Trq2 and YFP-tagged constructs (Trq2: IR laser λexc 810 nm, two-photon technique, λem = 430–505 nm, META detector; YFP: argon laser, λexc = 514 nm, λem = 560 nm long pass filter). FRET data analysis was performed using the software ZEN2010 (Carl Zeiss Microscopy, Jena, Germany), and Excel2013 (Microsoft Office), respectively. For the analysis, ROIs were set on the contact side of two neighboring HEK293 cells that were expressing both claudins. The fluorescence spectra of Trq2-tagged constructs in presence and absence of the YFP-tagged constructs receptor were measured. To calculate the FRET-based fluorescence, a λ-stack with a linear spectral unmixing mode was used to correct any YFP fluorescence crosstalk into the FRET channel (523–532 nm). The λ-stack is an integral part of the confocal laser system software. YFP correction was carried out to correct for direct excitation of the acceptor during donor excitation. The effect was expressed by changes in the FRET ratio, which is calculated by dividing the acceptor emission (YFP, 532 nm) by the donor emission (Trq2, 468 nm).

COS-7 cells were transfected with pTrq2- and pYFP-Cldn constructs. A plasmid DNA ratio of 1:5 was used for all pTrq2-Cldn3 and pTrq2-Cldn2 approaches. Transfected COS-7 cells were visualized on a LSM780 inverted microscope (Carl Zeiss Microscopy GmbH, Jena, Germany) using a 63x/1.3 oil objective. Trq2 fluorescence signals were recorded (405 diode, λexc = 405 nm, META detector, spectral range 438–639 nm) and split using a MBS-405. Channel pictures were taken prior to the recording of spectra in order to estimate expression of the Trq2 and YFP-tagged constructs (Trq2: 405 diode λexc 405 nm, λem = 414–502 nm, META detector; YFP: argon laser, λexc = 514 nm, λem = 516–652 nm). FRET data analysis was performed using the software ZEN2010 (Carl Zeiss Microscopy, Jena, Germany), and Excel2013 (Microsoft Office), respectively. For the analysis, ROIs were set on the contact side of two neighboring HEK293 cells that were expressing both claudins. The fluorescence spectra of Trq2-tagged constructs in the presence and absence of the YFP-tagged constructs receptor were measured. To calculate the FRET-based fluorescence, a λ-stack with a linear spectral unmixing mode was used to correct any YFP fluorescence crosstalk into the FRET channel (516–652 nm). The λ-stack is an integral part of the confocal laser system software. YFP correction was carried out to correct for direct excitation of the acceptor during donor excitation. The effect was expressed by changes in the FRET ratio, which is calculated by dividing the acceptor emission (YFP, 530 nm) by the donor emission (Trq2, 478 nm).

### Strand lengths measurement

COS-7 cells were transfected with pSNAP-Cldn2 and pYFP-Cldn10a in three different ratios (Cldn2/Cldn10a: 3:1, 1:3) with a total plasmid conc. of 1.6 µg/ml. In addition, pHalo-C1 was used to normalize the plasmid concentration over the different ratios. For the detection and protein amount determination via Western Blot (see: Immunoblot-based analysis) the antibodies rabbit anti-SNAP (New England BioLabs Inc., #P9310S) (1:1000) and mouse anti-GFP (Thermo Fisher Scientific, #A-11120) (1:2000) as well as a mouse anti-HSP70 (Thermo Scientific Fisher, #MA3006) (1:5000) were used. For the strand length analysis via IF (see: Immunofluorescence of overexpressed claudins), the cells were labeled with BG-Atto590 and immunofluorescent stained with α-GFP-NB-Atto647N (1:200) and imaged in two-color STED (see the "Time-gated single- and multi-color STED imaging" section). For the measurement of each condition, 40 individual strands of 3–4 different meshworks were analyzed using the segmented line function of Fiji ImageJ.

Four STED images of three different isolated proximal tubules stained with mouse anti-Cldn2, donkey anti-mouse Atto647N and rabbit anti-Cldn10 rabbit, donkey anti-rabbit AF594 (see: Immunofluorescence of single murine tubules from proximal tubule) were taken and imaged with the same settings (1024 × 1024, 16x line averaging, pixel size of 18.9 × 18.9 nm). For the measurement ≥132 individual claudin strands per image were analyzed using the segmented line function of Fiji ImageJ.

### Cholesterol depletion assay

COS-7 cells were transfected with pSNAP-Cldn2 and pYFP-Cldn10a with a total plasmid conc. of 1.6 µg/ml and treated with 10 µM Mevastatin (Sigma-Aldrich, #567022-5MG) for 24 h in DMEM with FBS or 10 mM Methyl-β-cyclo-dextrin (MβCD) (Sigma-Aldrich, #C4555-1G) for 1 h in DMEM without FBS. As control in an additional approach the equal amount of $H_2O$ was used. The transfected cells were labeled with BG-Atto590 and immunofluorescent stained with α-GFP-NB-Atto647N (1:200) and imaged in two-color STED.

For Filipin III staining, the cells were fixed with 4% PFA/Sucrose for 15 min, quenched with 0.1 M glycine in PBS$^+$ for 30 min and incubated for 2 h with a freshly prepared Filipin III solution (final concentration: 0.05 µg/ml) (Sigma-Aldrich, #F4767-1MG) under light protection. The cells were washed with PBS$^+$ and imaged with a Leica SP8 TCS microscope (Leica Microsystems). Filipin III staining was imaged with the following settings: for excitation, a UV-laser was used (405 nm) and the emission filter was set to 415–470 nm. The fluorescence signal was detected by a photomultiplier tube (PMT). Images were acquired with a HC PL APO CS2 100 × /1.40 NA oil objective (Leica Microsystems), a scanning format of 1024 × 1024 pixels, 8-bit sampling, 6x line averaging and 1x optical zoom, yielding a pixel dimension of 113 × 113 nm.

### Pore blocking assay

COS-7 cells were transfected with pSNAP-Cldn2$^{I66C}$ and pYFP-Cldn10a with a total plasmid conc. of 1.6 µg/ml. Cells were either treated with 1 mM MTSET (Biotium, #91021) or the equal volume of DMSO (Roth, #A994.1) for 2 h in serum-free DMEM. Cells were labeled with BG-Atto590 and immunofluorescent stained with α-GFP-NB-Atto647N (1:200) and imaged in two-color STED (see the "Time-gated single- and multi-color STED imaging" section).

### Co-culture assay

COS-7 cells were seeded in 6-well plates and transfected with pSNAP-Cldn2$^{ECL10a}$, pYFP-Cldn10a, or pYFP-Cldn2. After 24 h the cells were detached and intensively washed with PBS. After resuspension in DMEM pSNAP-Cldn2$^{ECL10a}$ transfected cells were mixed with pYFP-Cldn2 or pYFP-Cldn10a transfected cells in a ratio of 1:1 and seeded on 25 mm glass coverslips (#1.5H) and incubated for 24 h at 37 °C and 5% $CO_2$. For SNAP-Tag labeling the BG-Atto590 was used (see: Cell culture). And the YFP-signal was boosted with the α-GFP-NB-Atto647N

(see: Immunofluorescence of overexpressed claudins). Confocal images were taken with the Leica SP8 TCS STED microscope. The excitation and emission filter settings were the same as described above (see the "Time-gated single- and multi-color STED imaging" section). The images were acquired with a HC PL APO CS2 $100 \times /1.40$ NA oil objective (Leica Microsystems), a scanning format of $1024 \times 1024$ pixels, 8-bit sampling, 8x line averaging and 1x optical zoom, yielding in a pixel size $113 \times 113$ nm.

### Electrophysiological of monovalent ion permeabilities and fluorescein flux measurements

For the electrophysiological and flux measurement cells (MDCKII, MDCKII QKO, MDCKII QKO FLAG-Cldn2/3/10a/15, MDCKII QKO FLAG-Cldn2 + FLAG-Cldn10a, MDCKII QKO FLAG-Cldn3 + FLAG-Cldn15) were seeded on $0.6$ cm$^2$ and $0.4$ μm pore sized transwell filter (see: cell culture). In addition, for all measurements, cells were seeded on filters or in 6-well plates for an additional immunofluorescence labeling and immunoblotting approach to control the claudin expression. The measurements for the ion permeabilities were performed with an Ussing-Chamber as described in detail in Günzel et al. 2009[16]. The measurements were performed in circulating Ringer's solution (21 mM NaHCO$_3$, 119 mM NaCl, 5.4 mM KCl, 1 mM MgSO$_4$, 1.2 mM CaCl$_2$, 3 mM HEPES, 10 mM glucose; pH of 7.4) with a total concentration of 140 mM Na$^+$ and 128.8 mM Cl$^-$. The solution was gassed and mixed using a bubble lift (95% O$_2$ and 5% CO$_2$) and warmed up to 37 °C. For dilution potential measurements, each side of the Ussing-Chamber was filled with 5 ml Ringer's solution. After acclimatization of the cells, 5 ml of a modified Ringer's solution containing 238 mM mannitol and only 80.5 mM Na$^+$ and 69.3 mM Cl$^-$ were added to the apical bathing solution. At the same time an equal amount of Ringer's solution was added to the basolateral bathing solution. The trans-epithelial resistance (TER) and voltage were recorded (every 10 s) during the whole experiment. The relative permeability ratios for Na$^+$ and Cl$^-$ (PNa/PCl) were calculated according to the Goldman-Hodgkin-Katz equation. Details are described in Günzel et al. 2009[12]. The absolute permeabilities (PNa, PCl) were calculated from the relative permeabilities and trans-epithelial resistances as described in Günzel et al. 2009[12]. For the flux measurement of fluorescein (332.31 Da), the bathing solution was changed back on both sides of the chamber to a Ringer's solution. A voltage clamp was applied to the system and fluorescein was added to the apical bathing solution (end conc.: 100 μM). Samples were taken every 5 mins over for a period of 15 min. Fluorescein concentrations were measured with a plate reader at 525 nm (Tecan Infinite 200 Pro, Tecan Trading AG, Männedorf, Switzerland).

### Statistics and reproducibility

All data were derived from at least three independent experiments and are presented as means ± standard deviation (SD) or median and interquartile range unless differently noted in the figure legend. Comparisons among groups were performed using a one-way ANOVA and additional Dunnett's multiple comparison test or Tukey's multiple comparison test or Kruskal–Wallis test. Comparisons between two independent groups, statistical significance was analyzed with a two-tailed non-parametric Mann-Whitney test in GraphPad Prism Version 5.04. The level of significance is indicated in the figures by asterisks (*$P \le 0.05$, **$P \le 0.01$, ***$P \le 0.001$, ****$P \le 0.0001$). No statistical method was used to predetermine sample sizes as sample sizes were not chosen based on a prespecified effect size. Instead, multiple independent experiments were carried out using several sample replicates as detailed in the figure legends.

### Reporting summary

Further information on research design is available in the Nature Research Reporting Summary linked to this article.

## Data availability

All relevant data supporting the key findings of this study are available within the article and its Supplementary Information files or from the corresponding author upon reasonable request. Source data are provided with this paper.

## Code availability

Code and example data to reproduce data from Fig. 2 are available here: https://doi.org/10.5281/zenodo.7009994.

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

## Acknowledgements
We thank Marc-Andre Kasper and Christian Hackenberger from the Chemical Biology department of the FMP in Berlin for their help with custom synthesis of fluorescent SNAP and Halo ligands and Henrik Dimke (University of Southern Denmark, Odense) for the gift of mouse-anti-Cldn16 antibody. pMD2.G was a gift from Didier Trono (Addgene plasmid #12259). This work was supported by the Sonnenfeld Foundation with a stipend to Hannes Gonschior and grants from the Deutsche Forschungsgemeinschaft (DFG, German Research Foundation) to Dorothee Günzel (GRK2318/TJ-Train A1; GU 447/14-2), to Jörg Piontek (GRK2318/TJ-Train A2, PI 837/4-2) and Martin Lehmann and Volker Haucke (GRK2318/TJ-Train A4).

## Author contributions
H.G. performed experiments and analyzed data, C.S. wrote code and performed Image and data analysis with help from R.E.v.d.V., J.E. and H.G. acquired and analyzed FRET data, H.G. and D.G. acquired and analyzed ion flux data, R.E.v.d.V. acquired & quantified STED images of selected claudins with ER, N.H. and M.B. prepared & stained mouse kidney tubules and found initial segregation phenotype in kidney; D.G., J.P., M.B. and MF provided reagents and analyzed data, H.G., V.H. and M.L. designed the study, analyzed data, and wrote the manuscript with input from all authors.

## Funding

## Competing interests
The authors declare no competing interests.
