## [Peer Review File · Nature Communications]

REVIEWER COMMENTS

Reviewer #1 (Remarks to the Author):

In this manuscript, the authors performed a systematic analysis of the assembly patterns of 26 different mammalian claudins including major splicing isoforms using STED superresolution microscopy to analyze them expressed by COS7 cells as a claudin null background. When individually expressed, half of them formed what they referred to as a tight junction like meshwork, the other half did not. They also expressed several combinations of claudins pairwise, predominantly with *cldn3*, to show that in many cases, one claudin influenced the assembly of another, they classified these effects as intermixing, integration, induction, segregation, and exclusion, which provides a valuable framework for understanding the fine structure of claudin organization in tight junctions. One notable finding was that channel forming claudins tended to segregate away from barrier forming claudins, which can influence their function. On the other hand, barrier forming claudins tended to fully intermix. This was validated using claudins expressed in MDCK quintuple knockout cells. Determinants of strand architecture were attributed to the extracellular claudin domains as opposed to interactions with cholesterol or ZO1.

Strengths of the study include the scope of the claudins tested, validation of the COS7 model, unbiased analysis of meshwork architecture (Fig. 2), enabling further classification by principal component analysis and validation of results using bona fide epithelial cell models (duodenum, proximal tubule) and complementary techniques (e.g. FRET).

Although the basic principles of claudin segregation seemed to be conserved in the different models tested, there were some differences in strand architecture when comparing native tissue with cell lines (e.g. Fig 3 g vs 3 f and 3c). It would be useful for this to be further discussed in the main text.

Also, as shown in Extended Data Fig 2, the claudins that are classified as not forming a meshwork seem to fall into multiple categories, such as those that are uniformly distributed (such as *cldn4*, *cldn8*, *cldn12*, etc.) some that form puncta (*cldn13*, *cldn25*, *cldn26*) and those that form another type of structure (*cldn11b*, *cldn16*, *cldn27*). Is there any significance to these differences in non-meshwork distributions?

The criteria for determining misclassification (Extended Figure 3) needs further explanation. For instance, was an independent stain for the endoplasmic reticulum used to determine cases for ER staining? How frequently did this occur?

Δ PDZ and Δ CT mutants affect more than just ZO1 interactions, this should be emphasized in the text. How do these results reconcile with previous studies showing that CT domain mutations and tail swaps alter strand number and architecture as measured by FFEM?

Why was FRET done using HEK cells? Would similar results be obtained with COS7 cells?

Reviewer #2 (Remarks to the Author):

In this manuscript, Gonschior and colleagues addressed the nanoscale organization of Claudin proteins at the tight junctions using super-resolution STED imaging. They showed that only a certain pool of claudins can form meshwork when alone, and multiple claudins forms various nanoscale behavior when together. For instance, some claudins are randomly

distributed alone but forms various nanoscale structures with other claudins and these structural principles can be integration, intermixing, segregation, exclusion and induction. Overall, the study has been done very carefully, it is full of interesting results and, I believe, it will be quite important for the field. Especially the nanoscale behavior of multiple Claudin complexes (integration, intermixing, segregation, exclusion and induction) is really nicely shown and will bring more interesting questions in the future. Therefore, I recommend publication of the manuscript. I only have a few points which needs clarification with easy experiments or textual amendments.

- Most of the imaging was done by nanobodies. Is accessibility of FP to nanobody the same for all Claudins? In my opinion, authors should unequivocally show that nanobody labelling does not bias the results on nano-organization (as there is growing evidence that nanobodies can select certain pools of the same protein PMID: 30466346; PMID: 33364580). This is especially for non-meshwork claudins. For instance, a live cell image of Cldn4 or Cldn8 (alone and together) would be great to prove that nanobody labelling does not interfere with nanoscale organization.

- Moreover, Atto647N is quite a hydrophobic dye which might interact with membranes easily. At least to show that the dye does not influence the results, nanobodies tagged with another dye (preferably not hydrophobic, such as Abberior Star 635P) should be used (only for selected claudins to prove the point).

Minor:

- Authors use many different ways of labelling, and it is hard to follow what the label is in given figures (FP vs SNAP vs nanobody). Therefore, it would be great to specify in the figures what fluorescent molecules they use. For instance, in figure 3f, instead of Cldn3 and Cldn1, one can write SNAP-Cldn3 and EYFP-Cldn1. This would make it easier to read the figures without going to legends for details.

- Size-dependent organization of Claudins (and how they affect other proteins' organization based on their size) was shown before, authors can discuss these results as well (PMID: 30209136). I believe it is in line with size dependence authors observe here.

- Fig6, green/red combination is not ideal for color blind readers.

Reviewer #3 (Remarks to the Author):

Summary:

This is a remarkable study with an incredible amount of data. The decision to study all claudins is ambitious and provides a global view compared to previous studies focusing on fewer claudins. This is a great thing about this work.

The authors use superresolution microscopy (STED) and FRET to assess the nature of strands formed by claudins expressed in non-polarized epithelial cells (COS-7). They find that the networks formed by individual claudins can be subclassified based on differences between structural organization. The data also shows multiple claudins cannot forming strands. This is new.

Pairwise expression showed that 1. some strand-forming claudins colocalize, or intermix, within strands, 2. some claudins that could not form strands were incorporated, or integrated, into strands formed by other claudins, 3. some combinations of two claudins that could not form strands independently led to induction of co-polymerization, 4. some pairs of claudins concentrated within exclusive subregions of a strand mesh, or segregated, and 5. some claudins pairs formed completely independent strand networks and excluded one another (figure 3). Authors then examined claudins-3, 2, and 15 by FRET. There are some significant concerns regarding the cell type used, technical quality of the images, automated methods of analysis, and statistical interpretations. But these data are mostly solid and make a significant contribution to the field.

The authors first continue by asking whether some of these properties, particularly

segregation can be observed in polarized epithelia using either Caco2 cells, mouse duodenum (figure 3), or mouse renal proximal tubules (figure 4). These data is not convincing and the numbers of images scored seem to be insufficient based on statistical results.

Authors deleted PDZ binding motifs from C-terminal claudin tails and seemed surprised to see that they could form strands even though this was originally shown by one of the authors (Furuse) in 1998 when he expressed claudins 1 and 2 that were FLAG-tagged at the C terminus (which would block PDZ binding). Since that time, at least two other groups have reproduced those data and one reported that PDZ domain is important for trafficking to the tight junction but not anchoring at the tight junction of in epithelial cells. This caused authors to conclude that the extracellular loops determine claudin polymerization. This may be incorrect since transmembrane regions of claudins differ and can have large differences in confirmation caused by single amino acid changes (Nakamura et al, Nat Comm, 2019). Nakamura also showed that the helix bending effects of a single amino acid change profoundly alter morphologies of tight junction strands when expressed in epithelial cell lines.

Finally, authors try to understand the implications for tight junction permeability and barrier functions. This work is not equal in quality to the other aspects of this work and might best be deleted from the manuscript entirely. It could serve as a starting point for a separate study. The work depends on MDCK cells in which 5 claudin isoforms have been knocked out. These may not have complete deletion of all claudins since previous work shows that MDCK cells express other claudins, including claudin-15 (Bagnat et al, Nat Cell Biol, 2007). Also, the study in which these quintuple claudin KO cells were reported (Otani et al, J Cell Biol, 2019) retained junctional organization by immunostains for ZO-1, ZO-2, ZO-3, and JAM-A and TEMs showed preservation of sites of close membrane apposition. So authors claim that these are TJ-free is not supported. Second the data are not consistent with known functions of claudins 2 and 10a and just don't make sense.

Specifics:

1. COS-7 cells are African green monkey kidney-derived epithelial cells (Jensen et al, Proc Natl Acad Sci U S A 1964). They are not fibroblasts. They express keratins but it is not known if they express claudins or other transmembrane tight junction proteins. Key studies should be repeated in true nonepithelial cells as others have done (such as rat-1 or NIH3T3)
2. Most of the images are inferior to recently published pictures using SIM and Airyscan (Van Itallie et al, Mol Biol Cell, 2017) or even widefield with deconvolution (Sasaki et al, Proc Natl Acad Sci USA, 2003). Does this relate to use of an undifferentiated epithelial cell type for the studies. Single to noise also seems poor.
3. Images of claudin-3 in mouse duodenum (figure 1) are striking. I have to ask if these are tight junction strands or tangential imaging artifacts -- Claudin-3 is present at lateral membranes. The differences between these and claudin 3 networks in COS-7 cells AND the thin lines seen by many other groups staining claudins in epithelial cells suggests that the results may be artifactual. STED-EM correlation microscopy is needed to validate the conclusions.
4. The timelapse in figure 1e seems to be of very quickly moving cells or have some sort of movement artifact and are not helpful in shining light on claudin strand dynamics. Older Sasaki and Van Itallie articles above are much better.
5. The classifications based on network features shown in figure 2 are great but are also confusing. The data makes it look like claudin-2 is more closely related to claudin-15 than claudin-5 and claudin-14 but this is not how they are grouped. How many times was the analysis repeated? Were probabilities determined? Overall, much more detail regarding the analytical methods should be provided along with the actual code used.
6. The clustering based on Haralick texture features shown in figure 2 and extended figure data 3 don't match. I appreciate the separate clustering based on mesh forming and non-

mesh forming morphologies in extended data, but it is surprising that claudins 1, 6, 9, and 11 clustered based on variance sums in figure 2 but claudin1 is in a separate group in extended data figure 3. This difference seems been driven by the sum average data which based on the color coding seems inconsistent in the 2 figures.

7. Claudin11 in fig 2 seems not to form strands. I also question whether intracellular or surface structures were analyzed in some claudins (see *cln22* extended data figure 3).

8. The analyses of strand intermixing, integration, induction, segregation, exclusion are entirely based on Pearson's correlation coefficients. In some cases, these are negative (figure 3 C, most channel forming claudins versus claudin-3), while others are close to zero (claudin-2 and claudin-3). The negative correlation should be interpreted as indicating segregation (repulsion) rather than simply a lower colocalization index, as stated in the text.

9. Furuse (1998) previously showed that claudins 1 and 2 copolymerize. What is different here?

10. In previous FRET studies of claudins differences between cis and trans interactions were assessed. It is not clear if these were assessed here, as the methods for the FRET studies are only superficially described in the extended methods. The authors should also explain the difference in Pearson values of claudin 3 with either claudin-2 or claudin-15 -- claudin-2 appears to have fret with claudin-3 while claudin-15 does not. This could make sense since claudin-2 had a Pearson's coefficient of approximately zero while the Pearson's coefficient for claudin 15 with claudin 3 seems to be negative. This is based only my review of the graphs. Proper statistical analysis should be applied in these questions and for other combinations. The same issues apply to claudin 2 and claudin 10a in figure 4.

11. Since claudin-1 and claudin 3 intermix, it would be good to ask if they behave similarly to one another when coexpressed with claudin 2 or claudin 15.

12. Figure 4e shows a mesh network of claudins 2 and 10a in MDCK C7 cells. The scale bar is indicated to represent 200 nm. I have a very hard time understanding this morphology. Were these polarized MDCK C7 monolayers or was this simply a cytoplasmic reticular distribution (for example, in the endoplasmic reticulum)? This would be simple to address using a cell impermeant snap tag probe

13. it's not clear what's been shown in figure 4g. It looks like staying at the apical aspect of epithelial cells lining a tubule with the lumen from the bottom left to top right (of the lower magnification image). If so, why only a single strip of claudin staining is seen instead of mesh pattern in the duodenum (figure 1). Proximal tubule and duodenum have similar numbers of strands, so these should be similar. The image also makes it seem that claudins 2 and 10a never colocalize, which should have led to a negative Pearson's coefficient but does not (even though it does in figure 4b). Finally, panel 4i results make it clear that analyzing only 3 tubules is insufficient.

14. As above, figure 5 is a nice start but adds little. It already known that ZO-1 binding interactions were not need for polymerization (but do dictate strand dynamics, Van Itallie et al, *Mol Biol Cell*, 2017). The observation that completely deleting the C-terminal tail may have had some effect is of interest and would be interesting to pursue (but is beyond the scope of this already data rich manuscript). However, simply excluding a key role for the C-terminal tail should not lead to the conclusion that the extracellular loops are responsible. As noted above, Nakamura et al (*Nat Comm*, 2019) have already reported a contribution of transmembrane domains to strand organization and Zhao et al (*Commun Biol*, 2018) have identified other interacting surfaces.

15. Figure 6 and the supporting data in extended figure 10 are the weakest part of this work and should be eliminated from the manuscript. First, the data do not match the conclusions. The test concludes that "expression of individual channel-forming CLdn2, CLdn10, or CLdn15 restored the relative and absolute permeability of TJ for the respective cations or anions." This is not true - extended data figure 10 shows that claudin-2 expression in quintuple knockout cells increases resistance and Na:Cl permeability ratio and figure 6 shows that this is because claudin2 reduced Cl permeability. However, claudin 2 is

a paracellular Na and water channel that does not change Cl permeability in routine epithelial monolayers. The changes induced by claudin 10a and claudin 15 are also incorrect with respect to previously defined behaviors of these claudins. A minor issue is that individual datapoints are not shown in 6d. It is hard to interpret further since bar graphs can make things that are not different look different.

16. It is also not clear how expression of any of these claudins reduced paracellular permeability of fluorescein which is anionic but has a hydrodynamic radius of only 5 Å which is smaller than defined pore sizes for claudins 2 and 15.

17. A model similar to that in 6e has been previously proposed based on mathematical modeling (Claude, *J Membr Biol*, 1978), structural modeling (Alberini, et al, *PLOS ONE*, 2017 and Samanta et al, *J Gen Physiol*, 2018), and patch clamping (Weber, *Ann N Y Acad Sci*, 2012). Together with functional and mutagenesis data from multiple groups, it is clear that claudin 2, and likely claudin 10a, form channels and that ions are unlikely to traverse breaks in tight junction strands. There are some data suggesting that strand breaks may explain macromolecule permeation but this is not studied here. Also, the MTSET experiment argues against the model of breaks since it targets the defined channel.

RESPONSE TO REVIEWER COMMENTS

We would like to thank all three reviewers for their positive comments on our study and their thoughtful and constructive remarks and questions that have helped us to further improve our manuscript and tailor it for the general readership of *Nature Communications*. Below we explain point-by-point how we have addressed the referees' suggestions.

Reviewer #1 (Remarks to the Author):

In this manuscript, the authors performed a systematic analysis of the assembly patterns of 26 different mammalian claudins including major splicing isoforms using STED superresolution microscopy to analyze them expressed by COS7 cells as a claudin null background. When individually expressed, half of them formed what they referred to as a tight junction like meshwork, the other half did not. They also expressed several combinations of claudins pairwise, predominantly with *cldn3*, to show that in many cases, one claudin influenced the assembly of another, they classified these effects as intermixing, integration, induction, segregation, and exclusion, which provides a valuable framework for understanding the fine structure of claudin organization in tight junctions. One notable finding was that channel forming claudins tended to segregate away from barrier forming claudins, which can influence their function. On the other hand, barrier forming claudins tended to fully intermix.

This was validated using claudins expressed in MDCK quintuple knockout cells. Determinants of strand architecture were attributed to the extracellular claudin domains as opposed to interactions with cholesterol or ZO1.

Strengths of the study include the scope of the claudins tested, validation of the COS7 model, unbiased analysis of meshwork architecture (Fig. 2), enabling further classification by principal component analysis and validation of results using bona fide epithelial cell models (duodenum, proximal tubule) and complementary techniques (e.g. FRET).

Although the basic principles of claudin segregation seemed to be conserved in the different models tested, there were some differences in strand architecture when comparing native tissue with cell lines (e.g. Fig 3 g vs 3 f and 3c). It would be useful for this to be further discussed in the main text.

Response: We thank the reviewer for his/her supportive statements and the thoughtful suggestion. In response, we have added the following explanatory statement on p.6 of the revised manuscript: "The apparent differences in the claudin strand architectures and colocalisation values between native tissue and cell lines (Fig. 3b,e,f,g; Fig. 4a,d,e,g) can be attributed to differences in

sample preparation (cryosectioned tissue vs. intact living or fixed cells), staining procedures (indirect immunofluorescence vs. genetically encoded markers), and/or the presence of other TJ proteins and additional cell adhesion structures in epithelial cells or native tissue.”

Also, as shown in Extended Data Fig 2, the claudins that are classified as not forming a meshwork seem to fall into multiple categories, such as those that are **uniformly distributed** (such as cldn4, cldn8, cldn12, etc.) some that **form puncta** (cldn13, cldn25, cldn26) and those that form another type of structure (cldn11b, cldn16, cldn27). Is there any significance to these differences in non-meshwork distributions?

Response: The difference in clustering of non-meshwork forming claudins is an interesting observation as pointed out by the reviewer. We have not analyzed any correlation further yet, since multiple non-classical claudins have not been assigned a clear function or tissue expression and no copolymerizing claudin has been reported.

We now state on p.4 of the revised manuscript: “We confirmed the mostly uniform and unstructured distribution of SNAP-Cldn4 or SNAP-Cldn8 by STED imaging of living COS-7 cells (Extended Data Fig. 5i). In contrast, all claudins previously attributed as “non-classic”^{24,38,39}, except for Cldn11 and Cldn20, did not form TJ strands on their own. Instead, they appeared either as uniformly distributed, small puncta, irregular clusters or small strands. Cldn22 and Cldn27 localized partly to ER (Cldn22 and Cldn27) (Extended Data Fig. 2 and 3).”

Since clustering could be influenced by chemical fixation and labeling and is incompletely characterized by Haralick texture features only (see points raised by reviewer #2 and #3) we also added the following statement to the discussion on p10: “Albeit a uniform distribution of Cldn4 and Cldn8 was observed here in fixed and living cells further analyses are required to determine the precise nanoscale architecture and organization of non-meshwork forming claudins in living cells. Classification of nanoscale intensity features of non-meshwork forming claudins in living cells could reveal new polymerization principles, the interplay of cis/trans interactions and general structure function relationships for these incompletely characterized claudin family members.”

The criteria for determining misclassification (Extended Figure 3) needs further explanation. For instance, was an independent stain for the endoplasmic reticulum used to determine cases for ER staining? How frequently did this occur?

Response: We appreciate this comment and now specify the reasons for misclassifications on p.4 of the revised manuscript: “In contrast, all claudins previously attributed as “non-classic”^{24,38,39}, except for Cldn11 and Cldn20, did not form TJ strands on their own. Instead, they appeared either as uniformly distributed, small puncta, irregular clusters or small strands. Cldn22 and Cldn27 localized partly to ER (Cldn22 and Cldn27) (Extended Data Fig. 2 and 3).”

In Ext. Data Fig. 3b,c we show enlarged overview images of several misclassified claudins and quantitative analysis of co-stainings of claudins with ER markers. We have added the following text to the figure legend: “Representative overview and magnification images of overlapping regions of misclassified Cldn3 and Cldn10b (both claudins form dense TJ-like meshworks but with a low signal intensity), Cldn12, mCldn18a, Cldn22 and mCldn27 (these claudins do not form TJ-like meshwork but the increased sometimes inhomogenous signal in the overlap and some ER localization (Cldn22 with 14/25 cells and Cldn27 with 15/19 cells with ER staining) led to a false interpretation by the algorithm) expressed in COS-7 fibroblasts in (a). All claudins were N-terminally tagged with YFP and boosted with α -GFP-NB-Atto647N. (c) Representative images of overlapping regions of COS-7 cells transfected with TJ-like meshwork forming SNAP-Cldn10a or non-TJ-like meshwork forming SNAP-Cldn22 (both magenta; BG-JF646) and immunostained for the ER marker Calreticulin (yellow; 2nd-AF594).”

deltaPDZ and delta CT mutants affect more than just ZO1 interactions, this should be emphasized in the text. How do these results reconcile with previous studies showing that CT domain mutations and tail swaps alter strand number and architecture as measured by FFEM?

Response: We thank reviewers #1 and #3 for the important notes and agree that our Δ PDZ and Δ CT mutants of Cldn2, 10a, 3 and 15 could disrupt additional interactions apart from ZO1.

To address this possibility, we have extended the conclusion on p.8 as follows: “Hence, claudin segregation does not appear to be a consequence of their association with extrinsic factors such as cholesterol, proteins that bind to phosphorylated residues within the C-terminal tail, or PDZ domain proteins such as ZO-1.”

In addition, we now discuss in detail our observations of impaired meshwork formation of deltaPDZ and delta CT and the possible roles of ECLs, C-term, PDZ motif, TMDs and other interfaces in claudin polymerization and nanoscale segregation on p.11 of the revised manuscript:

“Unexpectedly, we observed smaller, unstructured and less frequent meshworks formed by combinations of C-terminal or PDZ deletion mutants of Cldn2/10 in COS-7 cells. Conversely, FFEM and live imaging data of C-terminal or PDZ deletion mutants of Cldn1, Cldn2 or Cldn3 expressed in

non-polarized fibroblasts were reported to lead to unperturbed strand and meshwork morphologies, albeit showing higher meshwork dynamics^{27,31,66}. We therefore speculate that segregated meshworks could be more dynamic and unstable, especially at segregation break points with possibly imperfect interaction interfaces. Here ZO1 interactions could be required to stabilize segregated meshworks. Future work based on live-cell superresolution microscopy could characterize nanoscale strand dynamics, strand breaks and molecular factors effecting the barrier properties of claudin homo and heteropolymers in TJs.”

Why was FRET done using HEK cells? Would similar results be obtained with COS7 cells?

Response: FRET measurements to assess claudin *cis*-interactions are frequently performed in HEK cells, since these cells do not form TJs, show robust trafficking of overexpressed proteins to the plasma membrane and their more rounded shape allows the collection of more fluorescent signal from claudins enriched at contact regions rather than from intracellular organelles [REF 40 = Piontek et al. 2011]. We additionally performed FRET measurements for Cldn2/10a and Cldn3/15 in living COS-7 cells and found these claudins to segregate, akin to our observations in HEK cells. These results are included in Ext. Data Fig. 7c and referred to on p.6 of the revised manuscript:

“No FRET between barrier-forming Cldn3 and channel-forming Cldn15 was observed in HEK and COS-7 cells (Fig. 3d; Extended Data Fig. 7c)” and on p7: “FRET measurements in HEK and COS-7 cells confirmed the observed lack of interaction between Cldn2 and Cldn10a on a scale below 10 nm in living cells (Fig. 4c; Extended Data Fig. 7c).”

Reviewer #2 (Remarks to the Author):

In this manuscript, Gonschior and colleagues addressed the nanoscale organization of Claudin proteins at the tight junctions using super-resolution STED imaging. They showed that only a certain pool of claudins can form meshwork when alone, and multiple claudins forms various nanoscale behavior when together. For instance, some claudins are randomly distributed alone but forms various nanoscale structures with other claudins and these structural principles can be integration, intermixing, segregation, exclusion and induction.

Overall, the study has been done very carefully, it is full of interesting results and, I believe, it will be quite important for the field. Especially the nanoscale behavior of multiple Claudin complexes (integration, intermixing, segregation, exclusion and induction) is really nicely shown and will bring

more interesting questions in the future. Therefore, I recommend publication of the manuscript. I only have a few points which needs clarification with easy experiments or textual amendments.

Response: We thank the referee for his/ her enthusiastic comments and the support to publish our findings in *Nature Communications*.

Most of the imaging was done by nanobodies. Is accessibility of FP to nanobody the same for all Claudins? In my opinion, authors should unequivocally show that nanobody labelling does not bias the results on nano-organization (as there is growing evidence that nanobodies can select certain pools of the same protein PMID: 30466346; PMID: 33364580). This is especially for non-meshwork claudins. For instance, a live cell image of Cldn4 or Cldn8 (alone and together) would be great to prove that nanobody labelling does not interfere with nanoscale organization.

Response: We thank the reviewer for raising this important point and performed the suggested experiment for non-meshwork claudins alone or during integration and induction using liveSTED imaging of SNAP tagged claudins. SNAP-Cldn3 forms meshworks in living COS-7 cells, but SNAP-Cldn4 or SNAP-Cldn8 do not (Ext. Data Fig. 5i). Moreover, we observed the integration of SNAP-Cldn4 into YFP-Cldn3 meshworks and the induction of meshworks upon coexpression of YFP-Cldn4 and SNAP-Cldn8 (Ext. Data Fig. 5j).

We have thus added the following statement on p.4 of the revised manuscript: "We confirmed the mostly uniform and unstructured distribution of SNAP-Cldn4 or SNAP-Cldn8 by STED imaging of living COS-7 cells (Extended Data Fig. 5i)."

On p.5, we now state: "Integration, induction and segregation could further be observed by live cell STED imaging (Fig. 3e, Fig. 4d and Extended Data Fig. 5j), excluding the possibility that these nanoscale organization principles represent artefacts of chemical fixation and nanobody labelling."

Moreover, Atto647N is quite a hydrophobic dye which might interact with membranes easily. At least to show that the dye does not influence the results, nanobodies tagged with another dye (preferably not hydrophobic, such as Abberior Star 635P) should be used (only for selected claudins to prove the point)

Response: We agree that fluorophores on nanobodies or antibodies can affect labelling efficiencies. We have used Atto647N labelled nanobodies to enhance all GFP/YFP-tagged claudins in fixed COS-7 cells (Fig. 1-5) under identical conditions and have not noted unspecific binding of Atto647N labelled nanobodies to membranes of untransfected cells. As suggested by the reviewer

using a GFP nanobody coupled to more hydrophilic Atto594 as well as a different SNAP ligands we could confirm the five nanoscale organization principles of claudin copolymers, namely intermixing, integration, induction, segregation and exclusion in COS-7 cells. We now state on p.5: “These five different types of nanoscale organization of claudin copolymers were also observed in the mouse fibroblast cell line 3T3 and in COS-7 cells using a GFP-nanobody labelled with a more hydrophilic fluorophore (Extended Data Fig. 7f).”

Minor:

- Authors use many different ways of labelling, and it is hard to follow what the label is in given figures (FP vs SNAP vs nanobody). Therefore, it would be great to specify in the figures what fluorescent molecules they use. For instance, in figure 3f, instead of Cldn3 and Cldn1, one can write SNAP-Cldn3 and EYFP-Cldn1. This would make it easier to read the figures without going to legends for details.

Response: We agree that clear labelling of constructs and dyes is important. We added the genetic fluorescent tag (GFP, YFP, SNAP, Halo), immunolabeling (NB, 1st and 2nd Antibodies) as well as the fluorophores to all figure legends.

Size-dependent organization of Claudins (and how they affect other proteins' organization based on their size) was shown before, authors can discuss these results as well (PMID: 30209136). I believe it is in line with size dependence authors observe here.

Response: Size-dependent protein segregation at membrane interfaces was demonstrated (PMID: 27980602), also for claudins (PMID: 30209136). However, the segregation of certain claudins identified in the current study is unlikely to be driven by size since most of the claudins have a similar molecular weight and very similar size of the extracellular and transmembrane domains (PDB IDs 4P79, 5B2G, 6OV3, 3X29, 6AKE). The C-terminal intracellular domains differ in length, however there is no indication that this results in a significant size difference for the segregating claudins. As discussed in greater detail on p.10 we believe that ultimately the strength of *cis*- and *trans* interactions regulates polymerization, meshwork formation and nanoscale organization during intermixing, integration, induction, segregation and exclusion.

- Fig6, green/red combination is not ideal for color blind readers.

Response: We want to thank the reviewer for this important point. We replaced green/red color in Fig. 6d and e with magenta/cyan.

Reviewer #3 (Remarks to the Author):

Summary:

This is a remarkable study with an incredible amount of data. The decision to study all claudins is ambitious and provides a global view compared to previous studies focusing on fewer claudins. This is a great thing about this work.

The authors use superresolution microscopy (STED) and FRET to assess the nature of strands formed by claudins expressed in non-polarized epithelial cells (COS-7). They find that the networks formed by individual claudins can be subclassified based on differences between structural organization. The data also shows multiple claudins cannot forming strands. This is new.

Response: We thank the referee for his/ her enthusiastic comments and for highlighting the impact and novelty of our study.

Pairwise expression showed that 1. some strand-forming claudins colocalize, or intermix, within strands, 2. some claudins that could not form strands were incorporated, or integrated, into strands formed by other claudins, 3. some combinations of two claudins that could not form strands independently led to induction of co-polymerization, 4. some pairs of claudins concentrated within exclusive subregions of a strand mesh, or segregated, and 5. some claudins pairs formed completely independent strand networks and excluded one another (figure 3). Authors then examined claudins-3, 2, and 15 by FRET. There are some significant concerns regarding the cell type used, technical quality of the images, automated methods of analysis, and statistical interpretations.

But these data are mostly solid and make a significant contribution to the field.

Response: We thank the referee for these positive remarks and for the lucid summary of our data. We comment on the concerns regarding cell type and image quality below.

The authors first continue by asking whether some of these properties, particularly segregation can be observed in polarized epithelia using either Caco2 cells, mouse duodenum (figure 3), or mouse renal proximal tubules (figure 4). These data is not convincing and the numbers of images scored seem to be insufficient based on statistical results.

Response: We regret to hear that the reviewer finds our segregation data not convincing and would like to recapitulate briefly what we consider to be strong and supportive evidence for our claim. We use several cultured cell lines (COS-7, HEK, 3T3, Caco-2, MDCK) as well as two different mouse tissues (proximal tubules, duodenum) combined with multicolor STED microscopy, FRET, as well as extensive mutagenesis and image & statistical analysis to show that Cldn3 and Cldn15 segregate in cells and in vivo. In Fig. 3f we merely show a cropped image of segregated Cldn3 and Cldn15 for space reason. Segregation of Cldn3 and 15 in Caco-2 cells was fully quantified and statistically analyzed using proper positive controls for colocalisation in Ext. Data Fig. 5e: “Pearson correlation analysis of SNAP-Cldn3 co-expressed with YFP-Cldn3 as ctrl (yellow) and YFP-Cldn15 (magenta) in Caco-2 cells. Data represent the mean \pm SD. Every *N* represents the Pearson of one TJ-like meshwork. $N(\text{Cldn3+Cldn3})=15$; $N(\text{Cldn3+Cldn15})=16$; from 3 independent experiments; Mann-Whitney test, two-tailed; **** $p \leq 0.0001$.”

In Fig. 3g and Fig. 4g we show representative STED images of claudin segregation in mouse duodenum and proximal tubules. To better illustrate the TJ localization in complex tissues we added large overview images of mouse duodenum to Ext. Data Fig. 5h and of proximal tubules to Ext. Data Fig. 7d. Segregation of Cldn2 and Cldn10a was quantitatively and statistically analyzed as shown and described in the legend to Fig. 4h and i as Pearson correlation and strand length measurements on three independent tubules. We hope that the referee agrees that these collective and statistically significant data from multiple independent experiments including analysis of endogenous claudin localizations provide strong supportive evidence for subsets of claudins including Cldn3 and Cldn15. We further note that referees 1 and 2 found our data in support of Cldn3/ Cldn15 segregation to be compelling.

Authors deleted PDZ binding motifs from C-terminal claudin tails and seemed surprised to see that they could form strands even though this was originally shown by one of the authors (Furuse) in 1998 when he expressed claudins 1 and 2 that were FLAG-tagged at the C terminus (which would block PDZ binding). Since that time, at least two other groups have reproduced those data and one reported that PDZ domain is important for trafficking to the tight junction but not anchoring at the tight junction of in epithelial cells.

Response: We agree and have now expanded the discussion on molecular determinants significantly. We also speculate about the effects of ZO1 binding on segregated meshwork structure on p. 11 of our revised manuscript (see also specific point 14 below).

Finally, authors try to understand the implications for tight junction permeability and barrier functions. This work is not equal in quality to the other aspects of this work and might best be deleted from the manuscript entirely. It could serve as a starting point for a separate study. The work depends on MDCK cells in which 5 claudin isoforms have been knocked out. These may not have complete deletion of all claudins since previous work shows that MDCK cells express other claudins, including claudin-15 (Bagnat et al, Nat Cell Biol, 2007).

Response: We respectfully disagree with this assessment that may partly be owed to a misinterpretation of our data. We have made strong efforts to better describe the experimental setup and outcome that, as further explained below, is fully consistent with the published literature on TJ permeability and its regulation by specific claudin isoforms. We further note that this part of the work was not criticized by any of the two other reviewers of this manuscript (see above).

To our knowledge the best model to test the paracellular permeability of selected claudins and combinations of segregating claudins are the recently published claudin Quin KO (QKO) cells from Otani et al. 2019. Based on mRNA data Otani et al. 2019 knocked out five major claudins from MDCKII cells to obtain claudin QKO cells. These cells were carefully characterized as being free of TJ meshworks and show a similarly low TER as ZO1/2 DKO cells, indicating the absence of a functional paracellular ion barrier. We have confirmed the absence of Cldn1, 2, 3, 4 and 7 by IF (data not shown) and WB (Fig. 6d) and measured exactly the same value for TER and fluorescein flux as reported by Otani et al. 2019. We thus conclude that QKO MDCK cells are TJ-strand free.

The reviewer suggests Cldn15 to be present in MDCKII cells according to Bagnat et al., Nat. Cell. Biol., 2007 [PMID: 17632505]. Bagnat et al. uses immunofluorescence analysis to show that Cldn15 is not detected in control MDCKII cells, but in MDCKII cells that stably express Cldn15 (see Fig. 3b from Bagnat et al. 2007 below). In agreement, we also fail to detect any Cldn15 in QKO and MDCKII by WB in Fig. 6d, while the antibody robustly detects stably expressed Cldn15. We therefore conclude that QKO MDCKII cells do not express Cldn15, in agreement with published data (Bagnat et al., Nat. Cell. Biol., 2007 [PMID: 17632505]).

Figure 3 Cldn15 forms a paracellular ion pore. **a**, The epithelial barrier remains functional in *tcf2²¹⁶⁹* mutant larvae. Rhodamine-dextran (M_r 10K) and the biotinylation reagent sulpho-NHS-biotin (S-NHS-biotin) were injected into the yolk of 72 h.p.f. WT and *tcf2²¹⁶⁹* mutant larvae. After 2 h, larvae were fixed and the distribution of the tracers was examined by confocal microscopy. Green, S-NHS-biotin; red, rhodamine-dextran; blue, F-actin. Scale bar, 20 μ m. **b**, Cldn15 (green) colocalizes with ZO-1 (red) in cell lines stably expressing Cldn15. Scale bar, 10 μ m. **c**, Stable Cldn15

expression (filled bars) reduces the TER in LLC-PK1 cells but not in MDCKII cells ($n = 6$) compared with control cells (open bars), which were transfected with an empty vector. Similar results were obtained with two independent clones for each cell line (only one is shown for each). Error bars indicate s.d. **d**, Epithelial sheets retained barrier function. Diffusion of fluorescent tracers (rhodamine-dextran (M_r 10K) and FITC-inulin) was not affected by the expression of Cldn15. Cells incubated in buffer without Ca^{2+} , a treatment used to open the junctions, were used as a reference.

Also, the study in which these quintuple claudin KO cells were reported (Otani et al, J Cell Biol, 2019) retained junctional organization by immunostainings for ZO-1, ZO-2, ZO-3, and JAM-A and TEMs showed preservation of sites of close membrane apposition. So author's claim that these are TJ-free is not supported.

Response: We agree that the work of Otani et al., published in J. Cell. Biol. in 2019 represents a careful study of the role of claudins and JAM-A in epithelial polarity. Indeed Otani et al. reports that ZO protein and JAM-A localization and cell polarity are preserved in QKO cells. Importantly, however, QKO cells display a complete absence of TJ meshworks, absence of membrane kissing points and their ion barrier function is impaired to a similar degree as in ZO1/2 KO cells, in which TJ strands are also absent. Based on the data reported by Otani et al. and our data shown here and contained in our revised manuscript we conclude that QKO cells are free of TJ meshworks and do

not possess functional ion barriers. In our revised manuscript we have changed the term “TJ-free” for “TJ-strand free”. We hope the reviewer will agree with his assessment.

Second the data are not consistent with known functions of claudins 2 and 10a and just don't make sense.

Response: We disagree with this assessment and assume that there referee might have been misled by our description of the data. We kindly refer the referee to our detailed response to specific points 14-16.

Specifics:

1. COS-7 cells are African green monkey kidney-derived epithelial cells (Jensen et al, Proc Natl Acad Sci U S A 1964). They are not fibroblasts. They express keratins but it is not known if they express claudins or other transmembrane tight junction proteins. Key studies should be repeated in true nonepithelial cells as others have done (such as rat-1 or NIH3T3).

Response: We thank the reviewer for this thoughtful remark and the reference to the paper. Both Thermo and ATTC label COS-7 cells as fibroblast-like or of fibroblast morphology, respectively. We have defined now at p.3 “in COS-7 cells, e.g. non-polarized fibroblast-like cells isolated from African green monkey kidney (Fig. 1b).” and removed the term “fibroblast” or changed into “fibroblast-like” or “non-polarized cells” in the context of COS-7 cells. COS-7 cells were shown to form no TJ strands (Nunes et al. 2007 [PMID: 17130295]) and used previously to study strand morphology of newly introduced claudins (Zhao et al. 2018 [PMID: 30271933]).

Additional we show that the five different types of nanoscale organization of claudin copolymers can be identified in COS-7 cells and in 3T3 fibroblasts (Fig. 3b, Ext. Data Fig. 7f). With respect to these findings we now state on p.5: “These five different types of nanoscale organization of claudin copolymers were also observed in the mouse fibroblast cell line 3T3 and in COS-7 cells using a GFP-nanobody labelled with the more hydrophilic fluorophore (Extended Data Fig. 7f).”

2. Most of the images are inferior to recently published pictures using SIM and Airyscan (Van Itallie et al, Mol Biol Cell, 2017) or even widefield with deconvolution (Sasaki et al, Proc Natl Acad Sci USA, 2003). Does this relate to use of an undifferentiated epithelial cell type for the studies. Single to noise also seems poor.

Response: We agree that non-deconvolved STED images can give the impression of lower signal to noise ratio compared to deconvolved/filtered images from SIM, Airy Scan or deconvolution microscopy since the number of emitting fluorophores is effectively reduced by the donut-shaped depletion laser (Gonschior et al. 2020 [PMID: 31979366]). SIM/Airy Scan/WF with deconvolution techniques achieve lateral resolutions of 120/150/200 nm, depending on the fluorophore (Gonschior et al. 2020 [PMID: 31979366]), were used in earlier works to analyse TJ-like meshworks with large meshes and isolated single strands (Van Itallie et al. 2017 [PMID: 27974639], Sasaki et al. 2003 [PMID: 12651952]). STED microscopy has a resolution of 40-50 nm, that becomes especially important for resolving densely packed strands in TJ and TJ-like meshworks, as shown compared to confocal image in Fig. 1c,d and throughout our manuscript. We carefully estimated the thickness of claudin strands using STED microscopy to be 60 and 70 nm in fixed and living systems respectively (Fig. 1f). Using STED microscopy, we were able to resolve small clusters (e.g. Cldn4), large meshes (e.g. Cldn19) and small meshes (Cldn3) as shown in Ext. Data Fig. 3. We hope that the referee will agree that our data provide unprecedented insights into TJ strand organization at sub-50nm resolution.

3. Images of claudin-3 in mouse duodenum (figure 1) are striking. I have to ask if these are tight junction strands or tangential imaging artifacts -- Claudin-3 is present at lateral membranes. The differences between these and claudin 3 networks in COS-7 cells AND the thin lines seen by many other groups staining claudins in epithelial cells suggests that the results may be artifactual. STED-EM correlation microscopy is needed to validate the conclusions.

Response: We agree with the reviewer that the images of TJ in mouse duodenum in Fig. 1 and 3 are striking. Single and double labelled TJs in mouse duodenum in Fig. 1 and 3 show clearly defined meshes of apically localized TJ. When TJ are oriented parallel to the x-y imaging plane they extend over ~500nm consistent with FFEM on the same tissue (Furuse 2010 [PMID: 20182608]). We stated on p.3: "The intestinal TJ meshwork showed a thickness of ~500 nm and was composed of several individual TJ strands, consistent with FFEM observations³⁰."

For better understanding we also added larger confocal overview image of mouse duodenum to Ext. Data Fig. 5h, where multiple TJ in the same optical plane are visible.

We also noted differences in the staining pattern of claudins transfected cell lines and in tissues, where additional components are present that might conceivably modulate TJ architecture. In our revised manuscript we explain these differences on p.6: "The apparent differences in the claudin strand architectures and colocalisation values between native tissue and cell lines (Fig. 3b,e,f,g; Fig. 4a,d,e,g) can be attributed to differences in sample preparation (cryosectioned tissue vs. intact

living or fixed cells), staining procedures (indirect immunofluorescence vs. genetically encoded markers), and/or the presence of other TJ proteins and additional cell adhesion structures in epithelial cells or native tissue.”

4. The timelapse in figure 1e seems to be of very quickly moving cells or have some sort of movement artifact and are not helpful in shining light on claudin strand dynamics. Older Sasaki and Van Itallie articles above are much better.

Response: The timelapse in Fig. 1e illustrates claudin strand dynamics with a strand break that leads to the fusion of two smaller into a larger mesh. This liveSTED sequence illustrates similar TJ-like meshwork organization and similar strand thickness in living and fixed cells (Fig. 1d-f). To facilitate interpretation, we have added the full uncropped movie as Ext. Data Movie 1 (corresponding to the time lapse shown Fig. 1e).

5. The classifications based on network features shown in figure 2 are great but are also confusing. The data makes it look like claudin-2 is more closely related to claudin-15 than claudin-5 and claudin-14 but this is not how they are grouped. How many times was the analysis repeated? Were probabilities determined? Overall, much more detail regarding the analytical methods should be provided along with the actual code used.

Response: We agree that more detail of the analysis and code used is necessary. We therefore describe the analysis in more detail in the methods section and provided a Zenodo repository that provides a detailed description and examples on how the different clustering analysis have been performed (<https://doi.org/10.5281/zenodo.5744991>). We performed a principal component analysis that allows to address the question towards probabilities. We adjusted the text in the legends to Fig. 2, Ext. Data Fig. 3 and Fig. 4 based on the ClustVis analysis:

“Extended Data Fig. 4 | Principal component analysis of the claudin meshwork formers and their average mesh sizes. (a) Principal component analysis (PCA) data of the analyzed meshwork forming claudins (Class A (cyan), Class B (yellow), Class C (magenta)) (from Fig. 2). Based on a dataset of 15 claudins with 202 total images. Unit variance scaling was applied to features over samples. Single value decomposition with imputation was used for calculating principal components. X and Y axis show principal component 1 and principal component 2 that explain 67.3% and 27.9% of the total variance, respectively. Prediction ellipses are such that with probability 0.95, a new observation from the same group will fall inside the ellipse. $N=15$ data points.”

From the PCA one can note that the classes C and B are not perfectly separated, as also reflected in the hierarchical clustering results. This is likely owed to the fact that the features extracted from multiple input images per claudin display some degree of variability as seen in Ext.Data Fig. 4b for the average mesh size. That said, we found the PCA analysis to be relatively robust; e.g. when choosing different ROIs or data from a different annotator the clustering was largely reproduced.

It is important to state though that unsupervised machine learning can only produce hypothesis that needs to be combined with experimental data.

6. The clustering based on Haralick texture features shown in figure 2 and extended figure data 3 don't match. I appreciate the separate clustering based on mesh forming and non-mesh forming morphologies in extended data, but it is surprising that claudins 1, 6, 9, and 11 clustered based on variance sums in figure 2 but claudin1 is in a separate group in extended data figure 3. This difference seems been driven by the sum average data which based on the color coding seems inconsistent in the 2 figures.

Response: We agree with the reviewer that the color code in the heatmaps between the different clustering approaches (Fig. 2a and Fig. 3a) is different. This is expected since:

- A) the analysis is based on mean features from different images of multiple datasets
- B) the values are normalized using unit variance scaling

We also noted that in our experience the clustering using only Haralick texture features is less robust. Thus, we placed less emphasis on this result and use it only to separate meshwork from non-meshwork forming claudins.

We agree that based on the previous version of the manuscript it was not perfectly clear that both approaches are distinct and cannot be directly compared. We have therefore revised the method section and legends to Fig. 2 (see p.10) and Ext. Data Fig. 3 and 4 to make the distinction clear: "A different clustering analysis approach based on clustering of intensity texture as well as meshwork morphology features identified three distinct groups of meshwork forming claudins (Fig. 2)."

7. Claudin11 in fig 2 seems not to form strands.

Response: Based on the literature we expect Cldn11 to form bundles of parallel strands with 40-60 nm spacing in murine testis and myelin sheets (Gow et al. 1999 [PMID: 10612400], Morita et al.

1999 [PMID: 10225958]). When Cldn11 was expressed in COS-7 cells and visualized by STED microscopy after fixation & staining YFP-Cldn11 bundles of parallel strands were visible and probably had an inter-strand spacing of < 60nm. Careful examination of YFP-Cldn11 (Ext. Data Fig. 2) and new quantitative analysis of SNAP-Cldn11 reveal occasional linear staining with a width of 63 ± 11 nm, that is consistent with the width of single strands of SNAP-Cldn3 with 59 ± 11 nm (compare Ext. Data Fig. 3d and Fig. 1f), assuming 10 nm thick claudin strands visualized with ~50 nm STED resolution.

We have added this information on p.4: "On rare occasions, single strands of Cldn11 were observed and measured to have a similar width as Cldn3 strands (Extended Data Fig. 3d; Fig. 1f)."

I also question whether intracellular or surface structures were analyzed in some claudins (see cldn22 extended data figure 3).

Response: Most claudins formed TJ-like meshworks in overlapping regions of two transfected COS-7 cells with clear strands and meshworks visible. Largely only the overlapping regions were cropped and used for texture feature and segmentation based image analysis as shown in Fig. 2. and Ext. Data Fig. 3 and 4. Nevertheless, as noted by us and the reviewer some claudins localized to the ER, possibly as a result of their failure to undergo proper folding or assembly as a prerequisite for ER export. In Ext. Data Fig. 3b we show larger overview images of some claudins with clear ER localization. We also counted and report the number of cells with predominantly ER staining for Cldn22 and Cldn27 (Ext. Data Fig. 3b and legend).

We state this now clearly in the main text on p.4: "In contrast, all claudins previously attributed as "non-classic"^{24,38,39}, except for Cldn11 and Cldn20, did not form TJ strands on their own. Instead, they appeared either as uniformly distributed, small puncta, irregular clusters or small strands. Cldn22 and Cldn27 localized partly to ER (Cldn22 and Cldn27) (Extended Data Fig. 2 and 3)."

8. The analyses of strand intermixing, integration, induction, segregation, exclusion are entirely based on Pearson's correlation coefficients. In some cases, these are negative (figure 3 C, most channel forming claudins versus claudin-3), while others are close to zero (claudin-2 and claudin-3). The negative correlation should be interpreted as indicating segregation (repulsion) rather than simply a lower colocalization index, as stated in the text.

Response: As suggested by the reviewer we have further clarified the significance of Pearson correlation coefficients for claudin colocalisation on p.5: "The intermixing, segregation, and

exclusion behavior of the various claudins with respect to Cldn3 was confirmed by quantitative colocalisation analysis based on Pearson correlation of pixel intensity, where positive values indicate intermixing and negative values indicate segregation or exclusion (Fig. 3c; Extended Data Fig. 5b-d).”

As indicated by the reviewer we also noted a higher colocalisation of Cldn3 with Cldn2 compared to other channel claudins Cldn10a, 10b and 15, while Cldn1 showed similarly low colocalisation with Cldn2, 10a, 10b and Cldn15. This may reflect the formation of distinct small Cldn2 clusters in the context of different *trans* interactions of Cldn2 with Cldn1 or Cldn3 (based on Furuse et al. 1999 [PMID: 10562289]). Hence, we now state on p.11 of our revised manuscript: “Conversely very small clusters were found in FFEM images of reconstituted and endogenous Cldn2^{12,31} but their molecular identity was so far unknown. Cluster from Cldn2 FEEM images could reflect differential association of Cldn2 with the protoplasmic and extracellular FFEM surfaces⁶⁶ or alternatively a segregation pattern of Cldn2 from Cldn1 or Cldn3. *Trans* interactions reported between Cldn2 and Cldn3, but not between Cldn2 and Cldn1⁶⁶ are mirrored by higher STED colocalisation of Cldn2/3 compared to Cldn2/1. We speculate that stable *trans* interactions between Cldn2 and Cldn366 might lead to very short Cldn2 strands that are cannot be resolved by STED microscopy. Such short strands may underlie the high degree of colocalisation between claudins, Cldn3 and Cldn2 compared to that of Cldn3 with other channel-forming claudins, e.g. Cldn10a, Cldn10b or Cldn15.”

9. Furuse (1998) previously showed that claudins 1 and 2 copolymerize. What is different here?

Response: In the papers suggested by the reviewer Furuse et al 1998 [PMID: 9647647] and Furuse et al. 1998 [PMID: 9786950] only immunofluorescence and FFEM analysis of cells forming TJ-like meshworks from individually overexpressed Cldn1 or Cldn2 are presented. While next to this copolymers for Cldn1 and occludin are shown, no copolymers of Cldn1 and Cldn2 are assessed.

We found relevant data on co-polymerisation of co-overexpressed Cldn1 and Cldn2 in L fibroblasts in Fig. 6 c-d of Furuse et al. 1999 [PMID 10562289]. Here copolymerization is deduced from frequently found adjacent immunogold labels for Cldn1 and Cldn2 in FFEM. Since the immune gold labelling density is quite low and has a precision of ~15 nm due to the size of antibodies and gold particles it is possible that Cldn2 actually segregates into nanometer-sized clusters along Cldn1 strands. Clusters of this size were indeed identified in FFEM of L-cells coexpressing Cldn1/Cldn2 or Cldn3/2 in the same publication [PMID 10562289]. The low STED colocalisation values reported by us for Cldn1/2 (Ext. Data Fig. 5f) would be consistent with such small Cldn2 clusters.

10. In previous FRET studies of claudins differences between cis and trans interactions were assessed. It is not clear if these were assessed here, as the methods for the FRET studies are only superficially described in the extended methods. The authors should also explain the difference in Pearson values of claudin 3 with either claudin-2 or claudin-15 -- claudin-2 appears to have fret with claudin-3 while claudin-15 does not. This could make sense since claudin-2 had a Pearson's coefficient of approximately zero while the Pearson's coefficient for claudin 15 with claudin 3 seems to be negative. This is based only my review of the graphs. Proper statistical analysis should be applied in these questions and for other combinations. The same issues apply to claudin 2 and claudin 10a in figure 4.

Response: We agree that *cis* and *trans* interactions are important for claudin polymerization. The FRET measurements of N-terminally tagged claudins in living HEK cells shown in Fig. 3d and Fig. 4c reflect mostly *cis*-interactions. These analyses were performed to investigate if segregation occurs on a smaller scale of 10 nm, i.e. below the resolution of STED microscopy. *Trans* interactions of claudins would result in a distance of over >10 nm that is beyond the distance range covered by the Turquoise/Venus FRET pair (Hochreiter et al. 2015 [PMID: 26501285]).

To clarify this important point, we have substantially expanded the description of spectral FRET methods. Following the suggestion of reviewer #1 we have also performed additional FRET measurements for Cldn2/10a and Cldn3/15 in living COS-7 cells and reproduced the data from HEK cells. These new results are now shown in Ext. Data Fig. 7c and mentioned on p.6 of the revised manuscript: "No FRET between barrier-forming Cldn3 and channel-forming Cldn15 was observed in HEK and COS-7 cells (Fig. 3d; Extended Data Fig. 7c)." and further on p.7 "FRET measurements in HEK and COS-7 cells confirmed the observed lack of interaction between Cldn2 and Cldn10a on a scale below 10 nm in living cells (Fig. 4c; Extended Data Fig. 7c)."

All FRET data are carefully analyzed statistically. Possible reasons for different FRET and STED values of Cldn2/3 compared to other channel claudins are discussed on p.10 and are detailed under point 9.

11. Since claudin-1 and claudin 3 intermix, it would be good to ask if they behave similarly to one another when coexpressed with claudin 2 or claudin 15.

Response: We showed in the initial submission that Cldn3 and Cldn1 show segregation from Cldn2, Cldn10a, Cldn10b and Cldn15 and wrote on p.6: "To test whether these behaviors reflect a general organizational principle of TJ formation by claudins we analyzed the barrier-forming Cldn1. We found Cldn1 to intermix with Cldn3, but segregate from channel-forming Cldn2, Cldn10a, Cldn10b

and Cldn15 (Extended Data Fig. 5f).” The different colocalisation of Cldn1/2 or Cldn3/2 are now discussed on p.11 of our revised manuscript.

12. Figure 4e shows a mesh network of claudins 2 and 10a in MDCK C7 cells. The scale bar is indicated to represent 200 nm. I have a very hard time understanding this morphology. Were these polarized MDCK C7 monolayers or was this simply a cytoplasmic reticular distribution (for example, in the endoplasmic reticulum)? This would be simple to address using a cell impermeant snap tag probe

Response: We have noted that when non-polarized MDCK C7 are transfected at lower confluency they form similar bicellular overlaps with TJ-like meshworks as in COS-7 as indicated in Fig. 1b. These TJ-like meshworks are formed by claudin polymers as shown in Fig. 1-5 and are clearly different from the ER for most claudins analysed with the exception of Cldn22 and Cldn27 as shown in Ext. Data Fig. 3. All SNAP-tagged claudins possess an N-terminal, intracellular SNAP tag so they cannot be labeled by cell-impermeable SNAP ligands in their correct membrane topology. We expect that inserting a 27 kDa SNAP tag of ~4 nm size into the extracellular loops of claudins will adversely affect *trans* and *cis* interactions and, thereby, disturb strand formation.

The image in Fig. 4e represents a cropped region from such an overlap of two MDCKC7 cells. We now specify this in the revised legend to Fig. 4e: “Cropped STED image of segregation of SNAP-Cldn2 (yellow; BG-Atto590) and YFP-Cldn10a (magenta; α -GFP-NB-Atto647N) localized in an overlap region of two transfected MDCKC7 cells.”

13. it’s not clear what’s been shown in figure 4g. It looks like staying at the apical aspect of epithelial cells lining a tubule with the lumen from the bottom left to top right (of the lower magnification image). If so, why only a single strip of claudin staining is seen instead of mesh pattern in the duodenum (figure 1). Proximal tubule and duodenum have similar numbers of strands, so these should be similar. The image also makes it seem that claudins 2 and 10a never colocalize, which should have led to a negative Pearson’s coefficient but does not (even though it does in figure 4b). Finally, panel 4i results make it clear that analyzing only 3 tubules is insufficient.

Response: In Fig. 4g we show representative STED images of claudin segregation in mouse proximal tubules. To illustrate our selections, we have added large overview images of proximal tubules to Ext. Data Fig. 7d. In contrast to duodenum, proximal tubules were found to comprise TJs with a single strand as reported by Muto et al. 2010 [PMID: 20385797]. We now further show by STED microscopy of intact proximal tubules that this single strand is formed by segregated Cldn2 and Cldn10a immunoreactivity. We speculate that the large clusters that are apparent in FFEM of single

TJ strands of proximal tubules WT, but disappear upon Cldn2 KO as reported by Muto et al., actually reflect segregated Cldn2 strands. Segregation of Cldn2 and Cldn10a was quantitatively and statistically analyzed by Pearson correlation and strand length measurements on 3 independent tubules with 4 STED images per tubule. This is now clearly indicated in the legend to Fig. 4.

Finally, we have added the following explanation for the different values of colocalisation of Cldn2/10a in COS-7, MDCK C7 and tissue on p.6 of our revised manuscript: “The apparent differences in the claudin strand architectures and colocalisation values between native tissue and cell lines (Fig. 3b,e,f,g; Fig. 4a,d,e,g) can be attributed to differences in sample preparation (cryosectioned tissue vs. intact living or fixed cells), staining procedures (indirect immunofluorescence vs. genetically encoded markers), and/or the presence of other TJ proteins and additional cell adhesion structures in epithelial cells or native tissue.”

14. As above, figure 5 is a nice start but adds little. It already known that ZO-1 binding interactions were not need for polymerization (but do dictate strand dynamics, Van Itallie et al, Mol Biol Cell, 2017).

The observation that completely deleting the C-terminal tail may have had some effect is of interest and would be interesting to pursue (but is beyond the scope of this already data rich manuscript).

However, simply excluding a key role for the C-terminal tail should not lead to the conclusion that the extracellular loops are responsible. As noted above, Nakamura et al (Nat Comm, 2019) have already reported a contribution of transmembrane domains to strand organization and Zhao et al (Commun Biol, 2018) have identified other interacting surfaces.

Response: We appreciate the suggestion of the reviewer and agree that contributions of the transmembrane domains and additional interaction surfaces to claudin strand formation and segregation cannot be ruled out. In Fig. 5 we have collected data to understand the molecular determinants of segregation, namely claudin protein levels, plasma membrane cholesterol, PDZ and C-term deletion and determinants in the extracellular loops. Although we were able to alter the segregation behavior of Cldn2/10 by interchanging its ECLs, we cannot rule out contributions from claudin transmembrane domains and additional interaction surfaces to strand formation as noted by Nakamura et al. 2019 [PMID: 30778075] and Zhao et al. 2018 [PMID: 30778075].

Both the literature (Furuse et al. 1998 [PMID: 9786950], Furuse et al. 1999 [PMID: 10562289], Van Itallie et al. 2017 [PMID: 27974639]) as well as our own data on Cldn1 and Cldn3 (Ext. Data Fig. 1d) indicate that addition of a C-terminal tag, deletion of the PDZ motif or deletion of the entire C-terminus do not affect claudin polymerization and meshwork structure in non-polarized cells.

Nanoscale segregation as measured by colocalisation analysis of STED images of mutant Cldn2/10a or Cldn3/15 was preserved in the absence of ZO-1 binding or other interactions mediated by the C-terminus. We thus note on p.8 of our revised manuscript: “Deletion of the C-terminal PDZ binding motif (required for binding to ZO-1) or even almost the entire C-terminal cytoplasmic domain (Fig. 5g) in either Cldn2 and Cldn10a or Cldn3 and Cldn15 resulted in smaller and less frequent meshworks, but did not affect the ability of claudins to segregate along TJ strands (Fig. 5d,e; Extended Data Fig. 8e). “

We have now expanded the discussion on molecular determinants significantly and speculate about the effects of ZO-1 binding on segregated meshwork structure on p. 11: “Neither Cldn2 channel inactivation by MTSET, cholesterol depletion, nor the deletion of the entire claudin C-terminus or the PDZ motif required for phosphorylation or ZO1 binding affected the nanoscale segregation. Unexpectedly, we observed smaller, unstructured and less frequent meshworks formed by combinations of C-terminal or PDZ deletion mutants of Cldn2/10 in COS-7 cells. Conversely, FFEM and live imaging data of C-terminal or PDZ deletion mutants of Cldn1, Cldn2 or Cldn3 expressed in non-polarized fibroblasts were reported to lead to unperturbed strand and meshwork morphologies, albeit showing higher meshwork dynamics^{27,31,66}. We therefore speculate that segregated meshworks could be more dynamic and unstable, especially at segregation break points with possibly imperfect interaction interfaces. Here ZO1 interactions could be required to stabilize segregated meshworks. Future work based on live-cell superresolution microscopy could characterize nanoscale strand dynamics, strand breaks and molecular factors effecting the barrier properties of claudin homo and heteropolymers in TJs. The extracellular loops of channel claudins form not only the paracellular ion pore through structural organization, but together with transmembrane helix⁶⁷ and other interacting surfaces^{23,68} contribute to strand formation and segregation.”

15. Figure 6 and the supporting data in extended figure 10 are the weakest part of this work and should be eliminated from the manuscript. First, the data do not match the conclusions.

The test concludes that “expression of individual channel-forming CLdn2, CLdn10, or CLdn15 restored the relative and absolute permeability of TJ for the respective cations or anions.” This is not true - extended data figure 10 shows that claudin-2 expression in quintuple knockout cells increases resistance and Na:Cl permeability ratio and figure 6 shows that this is because claudin2 reduced Cl permeability.

However, claudin 2 is a paracellular Na and water channel that does not change Cl permeability in routine epithelial monolayers. The changes induced by claudin 10a and claudin 15 are also incorrect with respect to previously defined behaviors of these claudins.

Response: While we appreciate this critical comment that may have arisen from a misunderstanding of the data presented, we wish to highlight the importance of functional rescue studies using single claudins and claudin combinations. Let me first explain our results. So far, ion flux measurements were performed in primary tissues or cultured cells or in cultured cells after genetic manipulation, that is either knockdown/out or overexpression of selected claudins in epithelial cell lines that express many different claudins. Clonal variations and changes in expression levels or localization of background claudins may hide functional properties of the claudins under investigation or complicate interpretation. Most studies of channel claudins were based on overexpression in high resistance epithelial cell lines (Amasheh et al. 2002 [PMID: 12432083], Furuse et al. 2001 [PMID: 11309408], Colegio et al. 2002 [PMID: 12055082], Van Itallie et al. 2001 [PMID: 11375422], Günzel et al. 2009 [PMID: 19383724]) that lead to lower TER and selective Na⁺ or Cl⁻ permeability.

In order to avoid interference from other claudins we selected TJ-strand free Quin claudin KO (QKO) cells from Otani et al. 2019 [PMID: 31467165] (please see also our response to the general comments above). In agreement with Otani et al, 2019 we found decreased resistance and increased fluorescein flux in QKO cells compared to MDCKII cells (Ext. Data Fig. 10a,b,e,f,). Otani et al. showed similar low TER values for QKO and ZO1/2 dKO cells indicating that both cell lines have lost TJ strands and their paracellular ion barrier function.

Our ion specific measurements show a high Na⁺/Cl⁻ permeability for MDCKII cells as reported by (Tokuda et al, 2015 [PMID: 25781928]) while QKO cells show mostly unspecific and unrestricted ion flux with a Na/Cl ratio of about 1 (Ext. Data Fig. 10). We therefore decided to measure and calculate ion-specific paracellular fluxes as absolute permeabilities that take into account both ion flux and resistance [PMID: 25781928, PMID: 32174144]. MDCKII cells show absolute permeabilities for Na⁺ of $\sim 50 \times 10^{-6}$ cm/s and for Cl⁻ of $\sim 5 \times 10^{-6}$ cm/s. These values are very similar to previously published data [PMID: 16322055, PMID: 25781928]. The absolute permeabilities of QKO cells for Na⁺ and Cl⁻ are higher and indicate free paracellular ion fluxes consistent with their low TER (Otani et al. 2019 [PMID: 31467165]). We therefore are confident about our electrophysiological measurements and have therefore included MDCKII and QKO cells to access claudin expression levels by WB.

Using the QKO system we show that re-expression of Cldn2 or Cldn15 increases TER, indicative of a functional TJ barrier. Moreover, Cldn2 or Cldn15 expressing cells show high Na⁺/Cl⁻ selectivity.

These data are highly consistent with previously published works (Amasheh et al. 2002 [PMID: 12432083], Furuse et al. 2001 [PMID: 11309408], Colegio et al. 2002 [PMID: 12055082]).

Reexpression of Cldn10a also increases TER but shows higher Cl⁻ permeability consistent with (Van Itallie et al. 2001 [PMID: 11375422], Günzel et al. 2009 [PMID: 19383724]). Since in our new system (that lacks background claudin expression) only a single claudin is expressed we conclude that paracellular ion flux occurs through pores within channel claudin strands.

Furthermore, we demonstrate that combined re-expression of Cldn2 and Cldn10a restores specific Na⁺ and Cl⁻ permeabilities as reported recently by Curry et al. 2020 [PMID: 32174144]. Moreover, combined re-expression of Cldn15 with Cldn3 increases the ion specificity of Cldn15. We therefore conclude that reconstitution of individual or multiple claudins reveals important new insights into the regulation of paracellular ion fluxes.

Finally, we use the term “TJ-strand free” in the context of QKO cells and rewrote the paragraph in the results section of our revised manuscript that describes the dilution potential measurements (p.9-10).

A minor issue is that individual datapoints are not shown in 6d. It is hard to interpret further since bar graphs can make things that are not different look different.

Response: We thank with the reviewer for this suggestion and now show individual data points in Fig. 6d.

16. It is also not clear how expression of any of these claudins reduced paracellular permeability of fluorescein which is anionic but has a hydrodynamic radius of only 5 Å which is smaller than defined pore sizes for claudins 2 and 15.

Response: The hydrodynamic radius of fluorescein is ~5 Å bigger than the estimated radius of channels formed by claudin-2 (3.25 Å (Yu *et al.* 2009 [PMID: 19114638]) to 4 Å (Van Itallie et al. 2008 [PMID: 18198187]) and claudin-15 (3 Å (Rosenthal et al., 2020 [PMDID: 31188544]) to 4 Å [PMID: 29915162]) and claudin-10b (2.6 Å (Rosenthal et al., 2020 [PMDID: 31188544])). This together with the negative charges prevents permeation of fluorescein through channel claudins and is consistent with the data presented in Ext. Data Fig. 10.

17. A model similar to that in 6e has been previously proposed based on mathematical modeling (Claude, J Membr Biol, 1978), structural modeling (Alberini, et al, PLOS ONE, 2017 and Samanta et al, J Gen Physiol, 2018), and patch clamping (Weber, Ann N Y Acad Sci, 2012). Together with functional and mutagenesis data from multiple groups, it is clear that claudin 2, and likely claudin 10a, form channels and that ions are unlikely to traverse breaks in tight junction strands. There are some data suggesting that strand breaks may explain macromolecule permeation but this is not studied here. Also, the MTSET experiment argues against the model of breaks since it targets the defien channel.

Response: We agree that a model for paracellular ion conductance over TJs based on pore structures within claudin strands was proposed earlier, for example by Claude (J Membr Biol, 1978 [PMID: 641977]). As mentioned by the reviewer evidence for these pore structures come from electrophysiological measurement in tissue, cell culture and upon channel claudin overexpression, claudin mutagenesis, mathematical modelling and other experimental evidence. As the nanoscale TJ incorporation of these pore structures nevertheless remained unresolved, we investigated the nanoscale distribution of several channel and barrier forming claudins and found segregation in several cell lines and tissue. Our “ion maze” model conceptualizes these findings and postulates that the nanoscale segregation of claudins enables paracellular ion fluxes.

We now refer to these earlier studies in the introduction on p.2 of our revised manuscript, which reads as follows: “Early electrophysiological measurements, mathematical modelling²⁰, as well as structural and functional data on channel and barrier claudins²¹⁻²³ led to a model in which paracellular pores formed by select claudins integrate into TJ strands. How TJ meshworks composed of single or multiple claudins are organized at the nanoscale to integrate paracellular barrier and channel functions is, however, unknown.”

REVIEWER COMMENTS

Reviewer #1 (Remarks to the Author):

The authors have addressed the previous critique by adding additional controls and by significant editing and additions to the text. This has further strengthened the manuscript.

My only concern was that I could not find Supplementary Tables 1-5. Please double check to make sure they were uploaded with the rest of the documents.

Reviewer #2 (Remarks to the Author):

Authors addressed all my concerns, I recommend the publication of the manuscript.

Reviewer #3 (Remarks to the Author):

This is an improved revision that addresses many of my concerns. If the permeability data was removed and the interpretation of seeing strands in epithelial cells significantly tempered to explain potential artifacts that cannot be excluded I would support acceptance.

Major

1. Most importantly, I still believe the transport aspects of this work should be removed. Authors now suggest that channel claudins make barriers as well. That could explain the data if claudin2 makes a barrier to Cl and claudin-10a makes a barrier to Na. But no supporting data exist anywhere in the literature and the data here are insufficient to make this claim. Even papers from these authors refer to separate channel and barrier claudins. There are experiments that you could do to test the idea, but I would agree that they are beyond the scope of this paper. I advise authors to continue their excellent work using Figure 6 as the starting point of their next paper.

2. The "ionic maze" model remains a problem and should be removed from 3a and 6d. Authors convincingly show claudin segregation in COS7 and this part of 3a can remain (without showing ion permeability). They can also discuss implication of integrating this idea into existing TJ network models. First, what they have drawn is not a maze (google image it). More importantly, this idea was first proposed nearly 45 years ago based on a log relationship between number of strands and barrier properties of various epithelial. In 1978, Claude published a black and white drawing essentially identical to the "ion maze" (except for the segregation idea) and wrote "...model of the junction in which each junctional strand contains "pores" that can be open or closed to small ions, with a certain probability of each strand in each pathway having an open pore." (Claude. J. Membr. Biol. 1978) The open-close capability of channels, or pores, was more recently shown by Weber (eLife 2015) and quantitative data from that combined into a mathematical model (Weber et al. Ann. N. Y. Acad. Sci. 2017/ (Alberini, et al, PLOS ONE, 2017/Samanta et al, J Gen Physiol, 2018). Based on this, the ion maze model should not be presented as a new innovation. No functional implications of segregation (the new idea here) are presented. Authors should instead remove the ion maze model and cartoon and use the discussion to speculate about how segregation might affect permeability/barrier.

3. I continue to question authors claims to see claudin strands in epithelial cells by STED.

a. STED does not improve z-axis resolution over traditional confocal. Therefore the only way STED could show strands as claimed in 1c, 3fg, ext 5h is if the images were somehow tangential so that the z axis of the TJ was in the xy plane of the microscope. But this can't be for fig 1c, since the claimed strands are seen around the entire cell.

b. FFEM demonstrates that strand spacing is on the order of 20 to 60 nm. Authors claim pixel sizes of 18.9 nm in xy (not resolution, which is not stated). But with these pixel sizes, there is no way that the complex strand structures shown in 1c, 3fg and others can be seen with the imaging used (18.9 nm pixels, ~50 nm resolution). This is apparent in the image in the Furuse 2010 review and measured

precisely in other species small intestine (Okamoto, Ishimura. Arch Histol Jpn 1978/ Madara. J Cell Biol 1983). One correction that may help authors understand. DISTANCES BETWEEN strands are 20 to 60 nm. Strands THICKNESS (SIZE) is much less. See papers above and many others (Gonzalez-Mariscal. J. Membr. Biol. 1985;86:113-25 shows MDCK).

c. Besides, STED and PALM/STORM and also airyscan and structured illumination approaches have already been applied to TJ for over a decade. No one else has reported these structures despite many careful studies. Authors report using a standard pulsed-light Leica STED system. Why are they able to see strands when others have used identical and similar systems without seen strands?

d. Ultimately, some sort of fluorescence (STED) – EM correlative microscopy approach must be used to validate what is being claimed.

4. Authors explain "The image in Fig. 4e represents a cropped region from such an overlap of two MDCKC7 cells." Presumably this is an artifact of growth pattern and not a tight junction. This should be clarified in manuscript.

Minor

1. I remain concerned regarding the quality of images in COS7. Please even see Sasaki et al 2003 PNAS where it is stated that deconvolution was only used for timelapse deltatvision images and no deconvolution is reported for the standard confocal. Perhaps Dr. Furuse who was in the same group as Sasaki can elaborate. The images in the present manuscript are much noisier than those in Sasaki and many since. Is there a signal:noise problem in the STED system?

2. Regarding claudin-15 expression in MDCK II. First, I apologize for citing the wrong paper.** The correct citation is Shukla et al. BMC Genomics 2015;16:944. This was cited as the source of transcriptomic data in the paper that first reported the quintKO MDCK. I cannot read the column headings of the excel screenshot in the rebuttal, but I take author at their word that *cln15* is not expressed. However, *cln12* and *16* are, so the point that these should not be referred to as claudin-less cells is still true. This must be considered when interpreting permeability in the present manuscript. How do claudins 2 and 10a interact with 12 and 16 in COS7?

**Note: The incorrect reference I provided (Bagnat) used a high resistance MDCK I-like cell line C7, even though they wrote MDCK II. We know this because Bagnat stated "...we turned to MDCKC7 cells, which display a very high TER." Authors of present manuscript have also transfected C7 with claudin-15 (Rosenthal R, GUNZEL D, et al. Acta Physiol. 2020).

3. In rebuttal authors state "It is important to state though that unsupervised machine learning can only produce hypothesis that needs to be combined with experimental data." This is an important point that should be included in the manuscript.

4. I agree with the idea behind the added statement "The apparent differences in the claudin strand architectures and colocalisation values between native tissue and cell lines (Fig. 3b,e,f,g; Fig. 4a,d,e,g) can be attributed to differences in sample preparation (cryosectioned tissue vs. intact living or fixed cells), staining procedures (indirect immunofluorescence vs. genetically encoded markers), and/or the presence of other TJ proteins and additional cell adhesion structures in epithelial cells or native tissue." However, this must be stated as a hypothesis since authors provide no data showing that this is true.

Reviewer #1 (Remarks to the Author):

The authors have addressed the previous critique by adding additional controls and by significant editing and additions to the text. This has further strengthened the manuscript. My only concern was that I could not find Supplementary Tables 1-5. Please double check to make sure they were uploaded with the rest of the documents.

We want to thank the reviewer for his/ her support and for pointing out the missing supplementary tables 1-5. The tables have been uploaded as additional excel sheets named NatComm_Material_final.

Reviewer #2 (Remarks to the Author):

Authors addressed all my concerns, I recommend the publication of the manuscript.

We thank the reviewer for his/her support and the recommendation to publish our work in *Nat Commun*.

Reviewer #3 (Remarks to the Author):

This is an improved revision that addresses many of my concerns. If the permeability data was removed and the interpretation of seeing strands in epithelial cells significantly tempered to explain potential artifacts that cannot be excluded I would support acceptance.

We want to thank the reviewer for the comments and suggestions that we used to further improve our manuscript. We now show summary of all permeability data in Fig. 6 and further explain and discuss the permeability data. As suggested by the reviewer we verify the expression of more claudins in MDCKII QKO cells by Western Blot and IF and test potential strand incorporation by co-overexpression of claudins in COS-7 cells combined with STED microscopy analysis (Extended Data Fig. 9-10) following the reviewer's suggestion. We also extensively rephrased the description of the STED imaging data of TJ strands in epithelial tissue.

Major

1. Most importantly, I still believe the transport aspects of this work should be removed. Authors now suggest that channel claudins make barriers as well. That could explain the data if claudin2 makes a barrier to Cl and claudin-10a makes a barrier to Na. But no

supporting data exist anywhere in the literature and the data here are insufficient to make this claim. Even papers from these authors refer to separate channel and barrier claudins. There are experiments that you could do to test the idea, but I would agree that they are beyond the scope of this paper. I advise authors to continue their excellent work using Figure 6 as the starting point of their next paper.

While we politely disagree with the suggestion to remove the transport aspect of this work we have now tried to better and more explicitly explain our novel findings and discuss them in the context of published work.

We totally agree with the reviewer that our claudin polymerization screen (Fig. 3) in combination with the permeability data suggest that channel claudins Cldn2, Cldn10a, and Cldn15 alone can form barriers. These claudins form TJ strands, increased the TER when reconstituted in MDCKII QKO and restrict the paracellular passage of fluorescein and of non-permissive ions (Fig. 6d and Extended Data Fig. 12). These findings agree with widely accepted definitions of channel and barrier functions of claudins [p.3 and Ref 41 Günzel & Yu 2013 PMID: 23589827]:

„When considering barrier or channel properties of a certain claudin, it has to be kept in mind that all claudins that insert into the cell membrane and interact with claudins from neighboring cells contribute towards a barrier, if compared with the situation of cells without any tight junctions. In that respect, all claudins are barrier-forming proteins. However, if there is already a preexisting TJ, then barrier-forming claudins should cause a further increase in trans-epithelial resistance, whereas pore-forming claudins should cause a decrease when inserted into this TJ. Conversely, knock-down of a barrier-forming claudin should cause a decrease in transepithelial resistance, knock-down of a pore-forming claudin an increase.“

Our findings that channel claudins Cldn2, Cldn10a, and Cldn15 alone can form barriers are indeed novel and consistent with the published literature since Cldn2, Cldn10a and Cldn15 form strands when overexpressed individually (Fig. 2 and Extended Data Fig. 2, Furuse et al 1998 [PMID: 9786950]), localize to the TJ (Fig. 6, Amasheh et al. 2002 [PMID: 12432083], Colegio et al. 2002 [PMID: 12055082], Van Itallie et al. 2006 [PMID: 16804102], Sengoku et al. 2008 [PMID: 17989991], Curry et al. 2020 [PMID: 32174144] and in case of Cldn2/Cldn10a regulate the paracellular transport in MDCK cells [Curry et al. 2020 [PMID: 32174144]) and in the proximal tubules (Muto et al. 2010 [PMID: 20385797] and recently Breiderhoff et al 2022 [PMID: 35031570].

Further, we found that Flag-Cldn15 expressed in MDCKII QKO cells has a four-fold preference of Na⁺ over Cl⁻. This is consistent with the prediction of ion permeability from molecular modelling of Cldn15 homomeric channels by Samanta et al. 2018 [PMID: 29915162]. We found a similar ion-selectivity for the Na⁺ channel Cldn2 (Na/Cl ~5) that

agrees well with published studies in different epithelial cells (Amasheh et al. 2002 [PMID: 12432083] and Curry et al. 2020 [PMID: 32174144]).

Therefore the permeability data shows that single claudins expressed in MDCKII QKO cells can act as barriers, e.g. Cldn3 or ion-selective channels, e.g. Cldn2, Cldn10a and Cldn15. A combination of segregated channel claudins Cldn2 and Cldn10a enable parallel ion flux of Na⁺ and Cl⁻ consistent with Curry et al. 2020 [PMID: 32174144] whereas the combination of channel and barrier claudins increases the ion specific conductance of the expressed channel claudin.

Overall concerning this point we have made changes to Fig. 6 and in the main text on p. 10 and p.13-14 and we hope to convince the reviewer that the “permeability data” is valid and important for our conclusion that the “nanoscale segregation of channel and barrier claudins enables paracellular ion flux”.

2. The “ionic maze” model remains a problem and should be removed from 3a and 6d. Authors convincingly show claudin segregation in COS7 and this part of 3a can remain (without showing ion permeability). They can also discuss implication of integrating this idea into existing TJ network models. First, what they have drawn is not a maze (google image it). More importantly, this idea was first proposed nearly 45 years ago based on a log relationship between number of strands and barrier properties of various epithelial. In 1978, Claude published a black and white drawing essentially identical to the “ion maze” (except for the segregation idea) and wrote “...model of the junction in which each junctional strand contains "pores" that can be open or closed to small ions, with a certain probability of each strand in each pathway having an open pore.” (Claude. J. Membr. Biol. 1978) The open-close capability of channels, or pores, was more recently shown by Weber (eLife 2015) and quantitative data from that combined into a mathematical model (Weber et al. Ann. N. Y. Acad. Sci. 2017/ (Alberini, et al, PLOS ONE, 2017/Samanta et al, J Gen Physiol, 2018). Based on this, the ion maze model should not be presented as a new innovation. No functional implications of segregation (the new idea here) are presented. Authors should instead remove the ion maze model and cartoon and use the discussion to speculate about how segregation might affect permeability/barrier.

We agree with the reviewer and have integrated our findings of nanoscale segregation into previous models of paracellular ion transport proposed by Claude 1978 [PMID: 641977] and Weber et al. 2015 [PMID: 26568313]. We now have eliminated the term "ion maze" from the manuscript and removed the model from Fig 6e to Extended Data Fig. 13. Furthermore, we state and discuss that nanoscale segregation enables the paracellular ion transport extending the model proposed by Claude et al. 1978 and Weber et al. 2015. Our permeability data show that:

- a single claudin can form a strong paracellular barrier, e.g. Cldn3
- single claudins can form ion-selective channels, e.g. Cldn2, Cldn10a or Cldn15

- Of note a ~ four fold Na/Cl selectivity for Cldn15 was measured as predicted by molecular modelling (Samanta et al. 2018 [PMID: 29915162])
- Coexpression of segregating Cldn2 and Cldn10a produces balanced Na⁺/Cl⁻ permeability, relevant to ion absorption in the proximal tubules (Muto et al. 2010 [PMID: 20385797], and Breiderhoff et al. (2022) [PMID: 35031570]) and consistent with parallel Na⁺ and Cl⁻ ion flux observed in a coexpression model (Curry et al. 2020 [PMID: 32174144])
- Coexpression of segregating barrier and channel claudin, Cldn3 and Cldn15 produces more specific ion fluxes (Na⁺/Cl⁻ ~15) compared to Cldn15 alone (Na⁺/Cl⁻ ~4).

The permeability data therefore reveals novel functional consequences of segregation consistent with molecular modelling. The segregation principle found by STED imaging and functional measurements allows us to significantly enrich the paracellular ion transport model proposed by Claude 1978 [PMID: 641977] and Weber et al. 2015 [PMID: 26568313]. We now discuss the extension of the previous model more explicitly on p.13-14 of the manuscript.

3. I continue to question authors claims to see claudin strands in epithelial cells by STED.

We thank the reviewer for the careful assessment of our STED data in tissue and we want to answer his/her points below.

- a. STED does not improve z-axis resolution over traditional confocal. Therefore the only way STED could show strands as claimed in 1c, 3fg, ext 5h is if the images were somehow tangential so that the z axis of the TJ was in the xy plane of the microscope. But this cant be for fig 1c, since the claimed strands are seen around the entire cell.*

The reviewer is right that the STED images we show do not have an increased z resolution, since an increase in z-resolution leads to a decrease in x-y resolution in our Leica SP8 STED system. Of note, the exact orientation of all TJ in the highly folded tissue architecture of mouse duodenum is hard to deduce from our STED images since tissue cryo sections were first collected on coverslips and then postfixed in 100% EtOH prior to immunostaining. As pointed out by the reviewer the TJ-meshworks are seen around the entire cell and only a few TJs from mouse duodenum are clearly resolved because they are finally oriented parallel to the focal x-y plane of the microscope where STED reaches ~50nm resolution. Most TJs will have an intermediate orientation to the focal plane and will appear thinner when projected into the imaging plane. Nevertheless some well-resolved TJ meshworks

extend over ~400-500nm consistent with many observations based on FFEM of similar tissue (Furuse 2010 [PMID: 20182608]).

We modified the result part on p.3 :

“Super-resolution STED microscopy with ~50 nm lateral (XY) resolution enabled us to visualize endogenous Cldn3 within TJs of mouse duodenum labeled with specific antibodies (Fig. 1c). While single strands and close strand assemblies below 50 nm distance could not be resolved within the intestinal TJ meshwork large individual meshes and the overall TJ thickness of ~500 nm became apparent, consistent with earlier observations from FFEM³⁰.”

We also noted differences in the staining pattern of claudins transfected cell lines and in tissues, where additional components are present that might conceivably modulate TJ architecture. In our revised manuscript we explain these differences on p.6:

“We speculate that the apparent differences in the claudin meshwork architectures and colocalisation values between native tissue and cell lines (Fig. 3b,e,f,g; Fig. 4a,d,e,g) could be attributed to differences in sample preparation and orientation (cryosectioned tissue vs. intact living or fixed cells), postfixation by ethanol, staining procedures (indirect immunofluorescence vs. genetically encoded markers), and/or the presence of other TJ proteins and additional cell adhesion structures in epithelial cells or native tissue.”

b. FFEM demonstrates that strand spacing is on the order of 20 to 60 nm. Authors claim pixel sizes of 18.9 nm in xy (not resolution, which is not stated). But with these pixels sizes, there is no way that the complex strand structures shown in 1c, 3fg and others can be seen with the imaging used (18.9 nm pixels, ~50 nm resolution). This is apparent in the image in the Furuse 2010 review and measured precisely in other species small intestine (Okamoto, Ishimura. Arch Histol Jpn 1978/ Madara. J Cell Biol 1983). One correction that may help authors understand. DISTANCES BETWEEN strands are 20 to 60 nm. Strands THICKNESS (SIZE) is much less. See papers above and many others (Gonzalez-Mariscal. J. Membr. Biol. 1985;86:113-25 shows MDCK).

The reviewer statement is correct that TJ strands as close as 20-60 nm cannot be resolved by our STED microscope at ~50nm resolution sampled with 18.9 nm pixel size. It is clear that only FFEM can resolve the full meshwork architecture and closely spaced TJ strands in chemically fixed cells & tissues. When we carefully re-analysed FFEM images by Furuse 2010 [PMID: 20182608] and more references suggested by the reviewer we found the majority of strands to display spacings in the order of 60-200nm (see below & Freeze_Fracture_Collection_from_Literature document attached).

- Okamoto et al. Arch Histol Jpn 1978 [PMID: 718386] shows FFEM of duodenal epithelial cells in chicken embryos with meshwork height of 484-646nm and strand spacing of 120-151nm
- Madara et al. J Cell Biol 1983 [PMID: 6863387] shows FFEM of jejunal epithelial cells from guinea pigs with a meshwork height of 205 nm and strand spacing of 64-94nm.
- Tamura et al. Gastroenterology 2008 [PMID: 18242218] shows FFEM of cells in the murine small intestine with meshwork height of 278nm and strand spacing of mostly >60nm.

Therefore, we conclude that medium to large meshes in epithelial cells & tissue can be resolved by STED microscopy but not strands distances smaller than 50nm. We modified the results part on p.3 (see below):

“Super-resolution STED microscopy with ~50 nm lateral (XY) resolution enabled us to visualize endogenous Cldn3 within TJs of mouse duodenum labeled with specific antibodies (Fig. 1c). While single strands and close strand assemblies below 50 nm distance could not be resolved within the intestinal TJ meshwork large individual meshes and the overall thickness of ~500 nm became apparent, consistent with earlier observations from FFEM³⁰.”

In addition, in the discussion we noted on p.6:

“We speculate that the apparent differences in claudin meshwork architecture between native tissue and cell lines (Fig. 3b,e,f,g) might be attributable to differences in sample preparation and TJ orientation (cryosectioned tissue vs. intact living or fixed cells), post-fixation by ethanol, staining procedures (indirect immunofluorescence vs. genetically encoded markers), and/or the presence of other TJ proteins and additional cell adhesion structures in epithelial cells or native tissue.”

c. Besides, STED and PALM/STORM and also airyscan and structured illumination approaches have already been applied to TJ for over a decade. No one else has reported these structures despite many careful studies. Authors report using a standard pulsed-light Leica STED system. Why are they able to see strands when others have used identical and similar systems without seen strands?

We agree with the reviewer and cited in our manuscript several super-resolution microscopy studies of the nanoorganisation of Claudins overexpressed in cells (Kaufmann et al. 2012 [PMID: 22319608], Van Itallie et al. 2017[PMID: 27974639], Schlingmann 2016 [PMID: 27452368], reviewed in Gonschior et al. 2020 [PMID: 31979366]. Our data widely confirms previous findings, e.g. larger mesh sizes of Cldn5 than Cldn3 as seen in Kaufmann

et al. 2012 [PMID: 22319608]. We can only speculate why TJ architecture in native tissue was not resolved so far. We have shown here that fixation conditions and labelling protocols notably using antibodies needed to be carefully optimized (Extended Data Fig. 1d,e). Therefore we anticipate that our explicitly described material & methods section, the use of a widely available commercial STED System (Leica), sharing of reagents, code and our novel STED imaging data may enable other research groups to conduct detailed nanoscale analysis of TJs in the future.

We extended the discussion on super-resolution microscopy on p.3:

“The slightly larger value in living cells could be a due to strand mobility and/or a lower signal-to-noise ratio in cell culture medium compared to fixed cell mounting medium. Of note STED images generally show a lower signal-to-noise ratio than imaging techniques that were previously used to image claudin strands including wide field³³, confocal³³ and SIM²⁷. We note that single molecule localization microscopy of YFP-Cldn3 in fixed HEK cells²⁸ yielded similar strand widths.”

d. Ultimately, some sort of fluorescence (STED) – EM correlative microscopy approach must be used to validate what is being claimed.

We have used STED microscopy to resolve the TJ nanoarchitecture in living cells and after careful optimization of labelling and fixation in several different cell lines and tissues. Strand thickness analysis (Fig. 1f), clearly separated strands with very homogenous intensity distributions within meshworks for several different claudins, e.g. Cldn2, Cldn10a, Cldn15 and Cldn19a (Extended Data Fig. 3) and highly similar structures observed in STED TJ images and published FFEM work (Muto et al. 2010 [PMID: 20385797], Furuse et al. 2010 [PMID: 20182608], Kaufmann et al. 2012 [PMID: 22319608]) indicate that we are able to visualize individual claudin strands that are further than 50 nm apart.

To the best of our knowledge correlative STED-EM on individual claudin strands has never been performed and is technically well beyond the scope of this work for the following reasons:

- to the best of our knowledge only FFEM resolves all TJ strands, but requires strong fixation, freeze fracture and metal coating of the sample followed by transmission electron microscopy (TEM)
- Correlative STED-FFEM would require first STED imaging of fluorescently tagged claudins in selected regions of intact fixed COS-7 cells, followed by strong post fixation with glutaraldehyde, freeze-fracture and search and imaging of exactly the same region in the FFEM sample by TEM - since STED has medium throughput and FFEM has very low throughput a correlative

image of exactly the same area is nearly impossible to achieve in a reasonable timeframe

- Correlative FFEM-STED requires first fluorescent labelling that survives strong chemical fixation, freeze fracture and metal coating followed by transmission electron microscopy. The remaining fluorescent signal from SNAP/Halo-tagged claudins in metal coated FFEM sample should then be imaged by STED microscopy. Importantly FFEM metal coating would create strong reflections & absorption of the STED laser (several kW) leading to sample heating/melting and severe laser safety concerns.

We therefore conclude that correlative STED-FFEM imaging is at the moment well beyond the scope of this work and would require extensive technical optimizations.

4. Authors explain "The image in Fig. 4e represents a cropped region from such an overlap of two MDCK7 cells." Presumably this is an artifact of growth pattern and not a tight junction. This should be clarified in manuscript.

We agree with the reviewer that this pattern could rather represent TJ-like meshworks that were observed upon overexpression of claudins in non-polarized MDCK7 cells or COS-7 cells. We changed the text on p.7 accordingly into:

"Segregation of channel-forming Cldn2 and Cldn10a was also found in living COS-7 cells visualized by live STED imaging (Fig. 4d) and in chemically fixed not fully-polarized kidney epithelial cells (Fig. 4e,f)."

Minor

1. I remain concerned regarding the quality of images in COS7. Please even see Sasaki et al 2003 PNAS where it is stated that deconvolution was only used for timelapse deltatvision images and no deconvolution is reported for the standard confocal. Perhaps Dr. Furuse who was in the same group as Sasaki can elaborate. The images in the present manuscript are much noisier than those in Sasaki and many since. Is there a signal:noise problem in the STED system?

We agree with the reviewer's observation that STED images have a lower signal-to-noise than standard confocal images even without deconvolution. The lower signal-to-noise ratio is explained by the reduced number of emitting fluorophores in the center of the donut-shaped depletion laser, higher spatial sampling with smaller pixel sizes (~20nm to sample the ~50nm resolution adequately) and repeated excitation/depletion cycles leading to higher bleaching of fluorophores in STED imaging (Gonschior et al. 2020 [PMID:

31979366]). Overall the ~50nm STED resolution we achieve here is required to resolve dense meshworks and to classify the polymerisation properties of different claudins.

We state now on p3-4:

“Of note STED images generally show a lower signal-to-noise ratio than imaging techniques that were previously used to image claudin strands including wide field³³, confocal³³ and SIM²⁷. We note that single molecule localization microscopy of YFP-Cldn3 in fixed HEK cells²⁸ yielded similar strand widths.”

*2. Regarding claudin-15 expression in MDCK II. First, I apologize for citing the wrong paper. ***

The correct citation is Shukla et al. BMC Genomics 2015;16:944. This was cited as the source of transcriptomic data in the paper that first reported the quintKO MDCK. I cannot read the column headings of the excel screenshot in the rebuttal, but I take author at their word that cldn15 is not expressed. However, cldn12 and 16 are, so the point that these should not be referred to as claudin-less cells is still true. This must be considered when interpreting permeability in the present manuscript. How do claudins 2 and 10a interact with 12 and 16 in COS7?

***Note: The incorrect reference I provided (Bagnat) used a high resistance MDCK I-like cell line C7, even though they wrote MDCK II. We know this because Bagnat stated “...we turned to MDCKC7 cells, which display a very high TER.” Authors of present manuscript have also transfected C7 with claudin-15 (Rosenthal R, GUNZEL D, et al. Acta Physiol. 2020).*

We thank the reviewer for clarifying the reference and for the suggestion to conduct additional control experiments that we have gladly conducted. First, we collected antibodies to verify the transcriptomic data from Shukla et al. 2015 [PMID: 26572553] and carefully analyzed the protein levels of Cldn6, Cldn12, Cldn15 and Cldn16 in MDCKII and QuinKO cells by Western Blot. We notably used overexpression of untagged claudins in QuinKO cells as pos. controls and found that all antibodies seem to recognize their target protein (Extended Data Fig. 9a). While in WB Cldn6 and Cldn15 are not found to be expressed at detectable levels in MDCKII and QuinKO cells we found low intensity bands of Cldn12 and Cldn16 running at the expected size in both cell lines. Nevertheless, we have not found detectable amounts of Cldn12 or Cldn16 immunoreactivity at TJ in MDCKII and Quin KO cells labeled with ZO1 (Extended Data Fig. 9b-c).

As suggested by the reviewer we next analysed the copolymerisation of Cldn12 and Cldn16 with either Cldn2, Cldn3, Cldn10a or Cldn15 in COS-7 cells using STED microscopy (Extended Data Fig. 10). Both Cldn12 and Cldn16 formed no clear meshwork but small clusters (Extended Data Fig. 2) that notably showed no co-enrichment with strands and meshworks formed by Cldn2, Cldn3, Cldn10a or Cldn15 in COS-7 cells. We conclude that while MDCKII and QKO cells express Cldn12 and Cldn16 at low levels, they neither localize

to the TJ nor show strand enrichment with the claudins investigated in this study, namely Cldn2, Cldn3, Cldn10a and Cldn15.

The lack of strand formation, strand incorporation and TJ localisation of Cldn12 is consistent with Yamazaki et al. 2011 [PMID: 21372174] that found Cldn12 to be expressed in SF7 cells from mouse testis, yet to be free of TJ strands. Cldn12 notably lacks a PDZ motif and does not form strands upon overexpression in COS-7 (Extended Data Fig. 2 and Extended Data Fig. 10a) or HEK 293 cells (Piontek et al. 2011 [PMID: 21533891]).

While Cldn12 is proposed to act as a Ca²⁺ channel in the intestine and kidney (Fujita et al. 2008 [PMID: 18287530], Beggs et al. 2021 [PMID: 34810264]) it did not affect Na/Cl fluxes in earlier studies (Fujita et al. 2008 [PMID: 18287530]).

Cldn16 requires Cldn19 for TJ localisation and to act as a cation channel. It does not interact with Cldn10a (Hou et al. 2008 [PMID: 18188451], Hou et al. 2009 [PMID: 19706394], Milatz et al. 2017 [PMID: 28028216]). We confirmed the integration of Cldn16 into Cldn19 strands in COS-7 cells (Extended Data Fig. 7a,b), but no integration into Cldn2, Cldn3, Cldn10a and Cldn15 meshworks was observed (Extended Data Fig. 10b).

Altogether, we conclude that albeit Cldn12 and Cldn16 are expressed at very low levels in MDCKII and QKO cells these claudins do not form strands individually, do not copolymerize with Cldn2, Cldn3, Cldn10a and Cldn15 and do not localize at the TJ in MDCKII and QKO cells. Therefore low abundant Cldn12 and Cldn16 are unlikely to change polymerisation and ion flux behavior of the segregated claudins studied here. We have changed the abstract & main text and explicitly call MDCKII QKO cells now on p.9 “genome-edited claudin-deficient TJ-strand depleted quintuple claudin knock-out MDCKII cells (MDCKII QKO)⁶² in which the expression of major endogenous claudins found by Shukla et al. ⁶³ namely Cldn1, 2, 3, 4 and 7 are eliminated.”

We added these observation to the results on p.4:

“Claudins previously implicated in bivalent cation transport such as Cldn12^{39,40} or Cldn16¹⁸ were unable to form detectable strands and meshworks under these conditions (Extended Data Fig. 2 and 3).”

and results on p.9:

“To determine the ion permeabilities of single claudins and segregated claudin pairs we capitalized on genome-edited claudin-deficient TJ-strand depleted quintuple claudin knock-out MDCKII cells (MDCKII QKO)⁶² in which the expression of major endogenous claudins found by Shukla et al. ⁶³ namely Cldn1, 2, 3, 4 and 7 are eliminated. While we were able to detect expression of Cldn12 and Cldn16 but not Cldn6 and Cldn15 in MDCKII WT or Quin KO cells by immunoblot analysis, both claudins largely failed to colocalize with occludin or ZO-1 at the plasma membrane (Extended Data Fig. 9). Furthermore neither Cldn12 nor Cldn16

appeared to be capable of forming TJ strands when expressed on their own in the absence of other claudins (Extended Data Fig. 2) and do not copolymerized with strands and meshworks formed by Cldn2, 3, 10a or Cldn15 in COS-7 cells when analyzed by STED microscopy (Extended Data Fig. 10).”

and to discussion on p.13:

“Of note, very low levels of non-meshwork forming Cldn12 and Cldn16 were expressed in MDCKII QKO (Extended Data Fig. 9a), but these failed to colocalize with occludin or ZO-1 at the plasma membrane (Extended Data Fig. 9b,c) and did not copolymerize with Cldn2, Cldn3, Cldn10a, and Cldn15 (Extended Data Fig. 10). The apparent absence of both claudins from the TJ could be explained by their inability to form strands and/or the absence of copolymerizing claudins such as Cldn19, as previously observed for Cldn16^{19,75}.”

3. In rebuttal authors state “It is important to state though that unsupervised machine learning can only produce hypothesis that needs to be combined with experimental data.” This is an important point that should be included in the manuscript.

We agree with the reviewer on the importance to verify analysis and classifications based on imaging data by other molecular and functional readout and added this statement to the discussion on p.11 of the revised manuscript:

“However, it is important to note that unsupervised machine learning can only produce classifications and hypotheses that then need to be tested by further experimental analyses, e.g. by super-resolution imaging, genetic manipulations or paracellular flux analysis. ”

4. I agree with the idea behind the added statement “The apparent differences in the claudin strand architectures and colocalisation values between native tissue and cell lines (Fig. 3b,e,f,g; Fig. 4a,d,e,g) can be attributed to differences in sample preparation (cryosectioned tissue vs. intact living or fixed cells), staining procedures (indirect immunofluorescence vs. genetically encoded markers), and/or the presence of other TJ proteins and additional cell adhesion structures in epithelial cells or native tissue.” However, this must be stated as a hypothesis since authors provide no data showing that this is true.

We agree and have focussed this statement on strand architecture more cautiously on p.6:

“We speculate that the apparent differences in claudin meshwork architecture between native tissue and cell lines (Fig. 3b,e,f,g) might be attributable to differences in sample preparation and TJ orientation (cryosectioned tissue vs. intact living or fixed cells), post-fixation by ethanol, staining procedures (indirect immunofluorescence vs. genetically encoded markers), and/or the presence of other TJ proteins and additional cell adhesion structures in epithelial cells or native tissue.”

REVIEWER COMMENTS

Reviewer #3 (Remarks to the Author):

I like almost all aspects of the paper. Much of the new work included in this revision really elevates the paper.

However, I remain unconvinced regarding the conclusion that claudin 2 and other channel-forming claudins form barriers. The supporting data are really a single experiment with a very unexpected result.

In reading the claudin 2I66C experiments in the revision, I did have one thought as to how this could be resolved experimentally. It should be easy experiment given that authors have claudin 2I66C. Alternatively, if the experiment wasn't done and transport data was not removed, the result should be discussed in the context of the unexpected conclusion that channel forming claudins form barriers. If this is done, it would be important to determine if claudins 12 and 16 are recruited to tight junctions when claudins 2, 3, 10a, or 15 were expressed in the QKO cells.

MAJOR CONCERN

Authors write "...restored the relative (Fig. 6d; Extended Data Fig. 12) and absolute permeabilities of TJs for Na⁺ (Cldn2, Cldn15) or Cl⁻ (Cldn10a) compared to as MDCKII QKO cells (Fig. 6e)."

This is technically true but is misleading. Expression of claudin 2 or claudin 15 had no effect on the absolute permeability of Na⁺. Therefore it did not "restore" but instead reduced permeability of Cl⁻. This is the reason the relative permeability of Na and Cl⁻ was changed by claudins 2 or 15. The opposite holds true for claudin 10a. This suggests that these channel forming claudins create a barrier. There are no published data indicating that claudins 2, 15 or 10a form barriers. The quotation from the Günzel & Yu included in the rebuttal really doesn't apply here, as it does not speak to the situation where there are no claudins detected at the junction.

PROPOSED EXPERIMENTAL RESOLUTION

In reading this revision, I realized that there is one critical experiment that could resolve this issue. If claudin 2I66C were expressed in QKO cells, authors will be able to add MTSET, as done in Fig. 1f in COS7 cells. Claudin 2I66C expression should reduce only Cl⁻ conductance initially but then reduce both Na⁺ and Cl⁻ conductance after MTSET addition. This would be very exciting.

MINOR CONCERNS

Despite my comments below, I can accept the first 3 concerns below as modest overinterpretations. The fourth point, regarding references, should be corrected (Professors Günzel and Piontek are authors of the *Acta Physiol* paper).

The final point would be an interesting to know, but is not essential of the proposed experiment is done with expected results or if the transport data are removed.

1. I remain skeptical of the image interpretation is demonstrating nanosegregation. Nevertheless, even if one accepts that as fact, there are no data in the paper to suggest that it relates to function. Therefore, the sentence in the abstract stating "Electrophysiological analysis of claudins in epithelial cells suggests that nanoscale segregation of distinct channel-forming claudins enables barrier function combined with specific paracellular ion flux across TJs." should be rephrased or deleted.

2. The observations regarding Cldn10bN48K are fascinating and likely of great importance. These join the data in Fig. 5h, i and supporting the idea that "channel-forming claudins are spatially segregated from barrier-forming claudins via determinants mainly encoded in their extracellular domains also known to harbor mutations leading to human diseases." What is missing in Fig. 5h, i is the demonstration of FRET between Cldn2 and Cldn10aECL2. If authors are correct in their interpretation, they should restore a FRET signal. Without this key condition, we can only see that substitution of the claudin 10a ECL can disrupt FRET. It is always easier to break something than to repair it, so I would consider the evidence to be less strong than implied by the statement in the abstract. I would suggest that the authors do this experiment or temper the statement.

3. I am a bit confused by the author statement that "STED images generally show a lower signal-to-noise ratio than imaging techniques that were previously used to image claudin strands including wide field, confocal and SIM." If the signal-to-noise ratio is reduced, how are the STED images able to detect features that other techniques could not detect?

4. Authors write "Of note prior structural modelling of the Cldn15 ion channel predicted a ~4-fold Na⁺/Cl⁻ selectivity, in close agreement with our experimental data (Fig. 6d, Extended Data Fig. 12c,d)." However, modeling is not necessary, as the cation selectivity of claudin 15 has already been measured and reported (Fig. 2c - Rosenthal et al. *Acta Physiol.* 2020;228:e13334). This paper should be cited in addition to or in place of the modeling data.

5. Are claudins 12 and 16 recruited to tight junctions when claudins 2, 3, 10a, or 15 were expressed in the QKO cells? This might be expected if claudins 12 and 16 depend on other claudins to stabilize them at tight junctions. It might also provide further insight into the transport observations.

RESPONSE TO REVIEWER COMMENTS

Reviewer #3 (Remarks to the Author):

I like almost all aspects of the paper. Much of the new work included in this revision really elevates the paper.

However, I remain unconvinced regarding the conclusion that claudin 2 and other channel-forming claudins form barriers. The supporting data are really a single experiment with a very unexpected result. In reading the claudin 2I66C experiments in the revision, I did have one thought as to how this could be resolved experimentally. It should be easy experiment given that authors have claudin 2I66C.

Alternatively, if the experiment wasn't done and transport data was not removed, the result should be discussed in the context of the unexpected conclusion that channel-forming claudins form barriers. If this is done, it would be important to determine if claudins 12 and 16 are recruited to tight junctions when claudins 2, 3, 10a, or 15 were expressed in the QKO cells.

- **Multiple experiments support the conclusion that Cld2 and other channel claudins form barriers as they form strands (Fig. 2) localize to TJ (Fig 6a,c) in epithelial cells and reduce flux of fluorescein and ions (Fig 6d,e)**

- Claudins Cldn2 and Cldn10a are expressed in the proximal tubules of kidney where they form Na⁺ and Cl⁻ selective paracellular barriers as shown by Muto et al. 2010 [PMID: 20385797] and recently Breiderhoff et al 2022 [PMID: 35031570]
- The suggested experiment albeit feasible does not add significantly new information since Cldn2 I66C channel inactivation by MTSET is well established [Weber 2015, Li 2014, Rosenthal 2016]
- Claudin 12 and 16 are not recruited to TJ in MDCKII cells that also express Cldn1,2,3,4,7,12 and 16 (Ext. Data Figure 9b,c) and do not copolymerize with Cldn2,3,10a,15 in Cos7 cells in an experiment suggested by the reviewer ((Ext. Data Figure 10)

We had therefore already modified the main text during the 2nd revision:

“When Cldn2 or Cldn15 were reconstituted in QKO cells a paracellular barrier containing Na⁺ selective channels was formed with a high Na⁺/Cl⁻ permeability that closely match predictions from molecular models of Cldn15 homopolymers^{66,74}. Conversely reconstitution of Cldn10a created barrier containing a Cl⁻ selective channels consistent with the results from former studies^{13,65,76}. Albeit based on a limited set of claudins we noted that barrier claudins show significantly lower permeability to the small molecule fluorescein than all channel-forming claudins tested. This could be a consequence of more complex meshworks and/or fewer strand breaks occurring in restituted barrier claudins and will be subject of further investigation.”

MAJOR CONCERN

Authors write “...restored the relative (Fig. 6d; Extended Data Fig. 12) and absolute permeabilities of TJs for Na⁺ (Cldn2, Cldn15) or Cl⁻ (Cldn10a) compared to as MDCKII QKO cells (Fig. 6e).”

This is technically true but is misleading. Expression of claudin 2 or claudin 15 had no effect on the absolute permeability of Na⁺. Therefore it did not “restore” but instead reduced permeability of Cl⁻. This is the reason the relative permeability of Na and Cl⁻ was changed by claudins 2 or 15. The opposite holds true for claudin 10a. This suggests that these channel forming claudins create a barrier.

There are no published data indicating that claudins 2, 15 or 10a form barriers. The quotation from the Günzel & Yu included in the rebuttal really doesn't apply here, as it does not speak to the situation where there are no claudins detected at the junction.

PROPOSED EXPERIMENTAL RESOLUTION

In reading this revision, I realized that there is one critical experiment that could resolve this issue. If claudin 2166C were expressed in QKO cells, authors will be able to add MTSET, as done in Fig. 1f in COS7 cells. Claudin 2166C expression should reduce only Cl⁻ conductance initially but then reduce both both Na⁺ and Cl⁻ conductance after MTSET addition. This would be very exciting.

While we agree that this is an exciting experiment but feel that is well beyond the scope of our work. We find that Cldn2 166C channel inactivation by MTSET and other Cystein-reactive reagents is already extensively characterized and published [Weber 2015, Li 2014, Rosenthal 2016].

Furthermore multiple of our experimental data support the conclusion that Cld2 and other channel claudins form barriers as they form TJ like meshworks and strands (Fig. 2) localize to TJ in epithelial cells (Fig 6a,c) and reduce flux of fluorescein and ions when reconstituted in epithelial cells (Fig 6d,e). Our observations that channel claudins can form barriers are also consistent with a preserved barrier function of the proximal tubules of Cldn2 KO mice (Muto et al. 2010 [PMID: 20385797] and Cldn10 KO mice Breiderhoff et al 2022 [PMID: 35031570]).

MINOR CONCERNS

Despite my comments below, I can accept the first 3 concerns below as modest overinterpretations.

The fourth point, regarding references, should be corrected (Professors Günzel and Piontek are authors of the Acta Physiol paper).

The final point would be an interesting to know, but is not essential of the proposed experiment is done with expected results or if the transport data are removed.

1. I remain skeptical of the image interpretation is demonstrating nanosegregation. Nevertheless, even if one accepts that as fact, there are no data in the paper to suggest that it relates to function. Therefore, the sentence in the abstract stating “Electrophysiological analysis of claudins in epithelial cells suggests that nanoscale segregation of distinct channel-forming claudins enables barrier function combined with specific paracellular ion flux across TJs.” should be rephrased or deleted.

We do not share and understand the skeptical view of the nanosegregation data as we have observed segregation of channel claudins in multiple cell lines, epithelial cells and in two different tissues.

Additionally (as recognized by the reviewer below) we have identified the molecular determinants of nanosegregation to be localized in the ECLs of Cldn2 and Cldn10a. Furthermore the permeability data in Fig 6 and Ext. Data Fig. 12 indicates a functional role of segregation, as:

- a single claudin can form a strong paracellular barrier, e.g. Cldn3
- single claudins can form ion-selective channels, e.g. Cldn2, Cldn10a or Cldn15
- a ~ four fold Na/Cl selectivity for Cldn15 as predicted by molecular modelling (Samanta et al. 2018 [PMID: 29915162])
- Coexpression of segregating Cldn2 and Cldn10a produces balanced Na⁺/Cl⁻ permeability, relevant to ion absorption in the proximal tubules (Muto et al. 2010 [PMID: 20385797], and Breiderhoff et al. (2022) [PMID: 35031570]) and consistent with parallel Na⁺ and Cl⁻ ion flux observed in a coexpression model (Curry et al. 2020 [PMID: 32174144])
- Coexpression of segregating barrier and channel claudin, Cldn3 and Cldn15 produces more specific ion fluxes (Na⁺/Cl⁻ ~15) compared to Cldn15 alone (Na⁺/Cl⁻ ~4).

As suggested by the reviewer #3 we can rephrase the abstract: “Electrophysiological analysis of claudins in epithelial cells suggests that nanoscale segregation of distinct channel-forming claudins **could enable barrier function combined with specific paracellular ion flux across TJs.”**

2. The observations regarding Cldn10bN48K are fascinating and likely of great importance. These join the data in Fig. 5h, i and supporting the idea that “channel-forming claudins are spatially segregated from barrier-forming claudins via determinants mainly encoded in their extracellular domains also known to harbor mutations leading to human diseases.” What is missing in Fig. 5h, i is the demonstration of FRET between Cldn2 and Cldn10aECL2. If authors are correct in their interpretation, they should restore a FRET signal. Without this key condition, we can only see that substitution of the claudin 10a ECL can disrupt FRET. It is always easier to break something than to repair it, so I would consider the evidence to be less strong than implied by the statement in the abstract. I would suggest that the authors do this experiment or temper the statement.

We thank the reviewer for the appreciation of the molecular characterization of ECLs in segregation by analyzing a disease mutant Cldn10bN48K and ECL replacement mutants. Unfortunately, the suggested ECL mutant Cldn10a^{ECL2} failed to localize properly in COS-7 cells and was not included in this manuscript. As expected by the reviewer we could show increased FRET (Extended Data Fig. 8f) and the presence of trans-interaction of Cldn2^{ECL10a} with Cldn10 in a coculture experiment (Extended Data Fig. 8

g) confirming the increased Pearson colocalisation value of Cldn2^{ECL10a} with Cldn10a (Fig 5i). Overall multiple lines of evidence suggest that ECLs from Cldn10a and Cldn2 mediate segregation.

The FRET data was described & discussed on p9 of the 2nd revised manuscript: “Cldn2^{ECL10a} showed significantly less segregation from Cldn10a, while it efficiently segregated from Cldn2 (Fig. 5h,i). FRET and co-culture experiments confirmed these results. They further revealed that although the exchange of the extracellular loops that dominate claudin interactions reversed the segregation behavior, it had only had minor effects on the lateral association of Cldn2^{ECL10a} and normal Cldn2 in *cis* (Extended Data Fig. 8f,g). These results are consistent with the fact that claudin-claudin interactions are not only based on the ECLs but also on the association of their transmembrane domains^{60,61}.”

3. I am a bit confused by the author statement that “STED images generally show a lower signal-to-noise ratio than imaging techniques that were previously used to image claudin strands including wide field, confocal and SIM.” If the signal-to-noise ratio is reduced, how are the STED images able to detect features that other techniques could not detect?

STED super-resolution microscopy achieves a resolution of ~50nm as quantified in Fig. 1f for TJ strands so STED has 3-5x better resolution than widefield, confocal or SIM. Therefore STED even at a lower signal-to-noise ratio can detect novel nanoscale features of TJ and TJ-like meshworks that were not resolved using standard microscopy techniques. The statement “*STED images generally show a lower signal-to-noise ratio than imaging techniques that were previously used to image claudin strands including wide field, confocal and SIM.*” was notably added to answer repeated concerns of reviewer #3 with STED resolution & image quality (2nd round of review Minor Point 1).

4. Authors write “Of note prior structural modelling of the Cldn15 ion channel predicted a ~4-fold Na⁺/Cl⁻ selectivity, in close agreement with our experimental data (Fig. 6d, Extended Data Fig. 12c,d).” However, modeling is not necessary, as the cation selectivity of claudin 15 has already been measured and reported (Fig. 2c - Rosenthal et al. Acta Physiol. 2020;228:e13334). This paper should be cited in addition to or in place of the modeling data.

Cation selectivity of claudin 15 in an overexpression system in epithelial cells was indeed published before [Rosenthal 2016,]. Consistent with the literature we found that coexpression of segregating barrier and channel claudin, Cldn3 and Cldn15 produces more specific ion fluxes ($\text{Na}^+/\text{Cl}^- \sim 15$) compared to Cldn15 alone ($\text{Na}^+/\text{Cl}^- \sim 4$). Since in the literature there is no experimental data for ion fluxes through Cldn15 homooligomers we have cited rather the modelling data from Cldn15 (Samantha et al 2018) following a previous suggestion of reviewer #3.

5. Are claudins 12 and 16 recruited to tight junctions when claudins 2, 3, 10a, or 15 were expressed in the QKO cells? This might be expected if claudins 12 and 16 depend on other claudins to stabilize them at tight junctions. It might also provide further insight into the transport observations.

We have not observed TJ localization of Cldn12 and Cldn16 in MDCKII cells, that express also Cldn1, 2, 3, 4 and 7 (Extended Data Fig. 9b,c). Additionally a revision experiment based on a suggestion from reviewer #3 showed no copolymers and strand incorporation of Cldn12 and Cldn16 with either Cldn2, 3, 10a and 15 in COS-7 cells. We notably discussed and cited literature that Cldn16 requires Cldn19 for TJ localization. Cldn19 mRNA is not expressed in MDCKII cells [Shukla et al 2015]. Identifying claudins that eventually copolymerize with Cldn 12 is beyond the scope of this work and can be addressed in the future using the COS-7 and QKO reconstitution system described in our work.

We could describe this observation more explicitly on page 13 (additional changes in yellow):

“Of note, very low levels of non-meshwork forming Cldn12 and Cldn16 were expressed in MDCKII and MDCKII QKO (Extended Data Fig. 9a), but these failed to colocalize with occludin or ZO-1 at the plasma membrane of MDCKII and MDCKII QKO (Extended Data Fig. 9b,c) and did not copolymerize with Cldn2, Cldn3, Cldn10a, and Cldn15 in COS-7 cells (Extended Data Fig. 10). The apparent absence of both claudins from the TJ could be explained by their inability to form strands and/or the absence of copolymerizing claudins such as Cldn19, as previously observed for Cldn16^{19,75}.”

REVIEWERS' COMMENTS

Reviewer #1 (Remarks to the Author):

This manuscript fills an important knowledge gap in our understanding of claudin assembly into tight junction strands and how claudin heterogeneity could influence local paracellular permeability. The study has novel ramifications for the control of epithelial barrier function. It is a revised manuscript and the authors have addressed all the concerns raised in previous reviews.

Strengths of the manuscript include:

STED microscopy analysis of 26 different individually expressed mammalian claudins. The analysis also includes native claudins expressed by intestinal and kidney epithelium that validate results using transfected cells.

SNAP-tagged claudins enabled live cell super resolution microscopy to validate results obtained using fixed cells/tissues.

Unbiased machine learning to analyze strand morphology enabled clustering of claudins into different assembly groups that correlate with claudin structure and function.

Quantitative analysis of domain formation by different combinations of co-expressed claudins demonstrates functional segregation of claudins, supports a model where there are localized differences in tight junction permeability.

PDZ binding motif and C-terminal domain deletion mutants show that strand formation is an inherent property of claudin structure and does not require ZO-1 binding. This is underscored by the claudin-2/claudin-10 domain swap experiments (Fig 5h,i) demonstrating that residues associated with the extracellular loop domains regulate claudin intermixing. The ECL domain swap experiments are compatible with publications using computational structural models predicting that extracellular juxtamembrane motifs control claudin intermixing.

Transfection experiments using QuinkO MDCK cells clearly demonstrate that cldn2 and cldn10a reduce paracellular permeability to the flux tracer fluorescein while simultaneously altering PNa and PCI. This finding is most consistent with cldn2 and cldn10a forming tight junction strands. In addition, previous literature as cited by the authors shows that these claudins form tight junction strands.

RESPONSE TO REVIEWER COMMENTS

Reviewer #1 (Remarks to the Author):

This manuscript fills an important knowledge gap in our understanding of claudin assembly into tight junction strands and how claudin heterogeneity could influence local paracellular permeability. The study has novel ramifications for the control of epithelial barrier function. It is a revised manuscript and the authors have addressed all the concerns raised in previous reviews.

Strengths of the manuscript include:

STED microscopy analysis of 26 different individually expressed mammalian claudins. The analysis also includes native claudins expressed by intestinal and kidney epithelium that validate results using transfected cells.

SNAP-tagged claudins enabled live cell super resolution microscopy to validate results obtained using fixed cells/tissues.

Unbiased machine learning to analyze strand morphology enabled clustering of claudins into different assembly groups that correlate with claudin structure and function.

Quantitative analysis of domain formation by different combinations of co-expressed claudins demonstrates functional segregation of claudins, supports a model where there are localized differences in tight junction permeability.

PDZ binding motif and C-terminal domain deletion mutants show that strand formation is an inherent property of claudin structure and does not require ZO-1 binding. This is underscored by the claudin-2/claudin-10 domain swap experiments (Fig 5h,i) demonstrating that residues associated with the extracellular loop domains regulate claudin intermixing. The ECL domain swap experiments are compatible with publications using computational structural models predicting that extracellular juxtamembrane motifs control claudin intermixing.

Transfection experiments using QuinKO MDCK cells clearly demonstrate that cldn2 and cldn10a reduce paracellular permeability to the flux tracer fluorescein while simultaneously altering PNa and PCl. This finding is most consistent with cldn2 and cldn10a forming tight junction strands. In addition, previous literature as cited by the authors shows that these claudins form tight junction strands.

We thank the reviewer for his/her supportive statements and remarks.